# Population codes enable learning from few examples by shaping inductive bias

Blake Bordelon[1,2], Cengiz Pehlevan[1,2]*

[1]John A Paulson School of Engineering and Applied Sciences, Harvard University, Cambridge, United States; [2]Center for Brain Science, Harvard University, Cambridge, United States

**Abstract** Learning from a limited number of experiences requires suitable inductive biases. To identify how inductive biases are implemented in and shaped by neural codes, we analyze sample-efficient learning of arbitrary stimulus-response maps from arbitrary neural codes with biologically-plausible readouts. We develop an analytical theory that predicts the generalization error of the readout as a function of the number of observed examples. Our theory illustrates in a mathematically precise way how the structure of population codes shapes inductive bias, and how a match between the code and the task is crucial for sample-efficient learning. It elucidates a bias to explain observed data with simple stimulus-response maps. Using recordings from the mouse primary visual cortex, we demonstrate the existence of an efficiency bias towards low-frequency orientation discrimination tasks for grating stimuli and low spatial frequency reconstruction tasks for natural images. We reproduce the discrimination bias in a simple model of primary visual cortex, and further show how invariances in the code to certain stimulus variations alter learning performance. We extend our methods to time-dependent neural codes and predict the sample efficiency of readouts from recurrent networks. We observe that many different codes can support the same inductive bias. By analyzing recordings from the mouse primary visual cortex, we demonstrate that biological codes have lower total activity than other codes with identical bias. Finally, we discuss implications of our theory in the context of recent developments in neuroscience and artificial intelligence. Overall, our study provides a concrete method for elucidating inductive biases of the brain and promotes sample-efficient learning as a general normative coding principle.

*For correspondence:
cpehlevan@seas.harvard.edu

**Competing interest:** The authors declare that no competing interests exist.

## Editor's evaluation

This important study presents a theory of generalization in neural population codes and proposes sample efficiency as a new normative principle. The theory can be used to identify the set of 'easily learnable' stimulus-response mappings from neural data and makes strong behavioral predictions that can be evaluated experimentally. Overall, the new method for elucidating inductive biases of the brain is highly compelling and will be of interest to theoretical and experimental neuroscientists working towards understanding how the cortex works.

## Introduction

The ability to learn quickly is crucial for survival in a complex and an everchanging environment, and the brain effectively supports this capability. Often, only a few experiences are sufficient to learn a task, whether acquiring a new word (*Carey and Bartlett, 1978*) or recognizing a new face (*Peterson et al., 2009*). Despite the importance and ubiquity of sample efficient learning, our understanding of the brain's information encoding strategies that support this faculty remains poor (*Tenenbaum et al., 2011*; *Lake et al., 2017*; *Sinz et al., 2019*).

**Figure 1.** Learning tasks through linear readouts exploit representations of the population code to approximate a target response. (**A**) The readout weights from the population to a downstream neuron, shown in blue, are updated to fit target values $y$, using the local, biologically plausible delta rule. (**B**) Examples of tuning curves for two different population codes: Smooth tuning curves (Code 1) and rapidly varying tuning curves (Code 2). (**C**) (Left) A target function with low frequency content is approximated through the learning rule shown in A using these two codes. The readout from Code 1 (turquoise) fits the target function (black) almost perfectly with only $P = 12$ training examples, while readout from Code 2 (purple) does not accurately approximate the target function. (Right) However, when the number of training examples is sufficiently large ($P = 120$), the target function is estimated perfectly by both codes, indicating that both codes are equally expressive. (**D**) The same experiment is performed on a task with higher frequency content. (Left) Code 1 fails to perform well with $P = 12$ samples indicating mismatch between inductive bias and the task can prevent sample efficient learning while Code 2 accurately fits the target. (Right) Again, provided enough data $P = 120$, both models can accurately estimate the target function. Details of these simulations are given in Methods Generating example codes (Figure 1).

In particular, when learning and generalizing from past experiences, and especially from few experiences, the brain relies on implicit assumptions it carries about the world, or its inductive biases (***Wolpert, 1996***; ***Sinz et al., 2019***). Reliance on inductive bias is not a choice: inferring a general rule from finite observations is an ill-posed problem which requires prior assumptions since many hypotheses can explain the same observed experiences (***Hume, 1998***). Consider learning a rule that maps photoreceptor responses to a prediction of whether an observed object is a threat or is neutral. Given a limited number of visual experiences of objects and their threat status, many threat-detection rules are consistent with these experiences. By choosing one of these threat-detection rules, the nervous system reveals an inductive bias. Without the right biases that suit the task at hand, successful generalization is impossible (***Wolpert, 1996***; ***Sinz et al., 2019***). In order to understand why we can quickly learn to perform certain tasks accurately but not others, we must understand the brain's inductive biases (***Tenenbaum et al., 2011***; ***Lake et al., 2017***; ***Sinz et al., 2019***).

In this paper, we study sample efficient learning and inductive biases in a general neural circuit model which comprises of a population of sensory neurons and a readout neuron learning a stimulus-response map with a biologically-plausible learning rule (***Figure 1A***). For this circuit and learning rule, inductive bias arises from the nature of the neural code for sensory stimuli, specifically its similarity structure. While different population codes can encode the same stimulus variables and allow learning of the same output with perfect performance given infinitely many samples, learning performance can depend dramatically on the code when restricted to a small number of samples, where the reliance on and the effect of inductive bias are strong (***Figure 1B, C and D***). Given the same sensory examples and their associated response values, the readout neuron may make drastically different predictions depending on the inductive bias set by the nature of the code, leading to successful or failing generalizations (***Figure 1C and D***). We say that a code and a learning rule, together, have a good inductive bias for a task if the task can be learned from a small number of examples.

In order to understand how population codes shape inductive bias and allow fast learning of certain tasks over others with a biologically plausible learning rule, we develop an analytical theory of the readout neuron's learning performance as a function of the number of sampled examples, or sample size. We find that the readout's performance is completely determined by the code's kernel, a function which takes in pairs of population response vectors and outputs a representational similarity defined by the inner product of these vectors. We demonstrate that the spectral properties of the kernel introduce an inductive bias toward explaining sampled data with simple stimulus-response maps and determine compatibility of the population code with the learning task, and hence the sample-efficiency of learning. We apply this theory to data from the mouse primary visual cortex (V1) (*Stringer et al., 2021*; *Pachitariu et al., 2019*; *Stringer et al., 2018a*; *Stringer et al., 2018b*), and show that mouse V1 responses support sample-efficient learning of low frequency orientation discrimination and low spatial frequency reconstruction tasks over high frequency ones. We demonstrate the discrimination bias in a simple model of V1 and show how response nonlinearity, sparsity, and relative proportion of simple and complex cells influence the code's bias and performance on learning tasks, including ones that involve invariances. We extend our theory to temporal population codes, including codes generated by recurrent neural networks learning a delayed response task. We observe that many codes could support the same kernel function, however, by analyzing data from mouse primary visual cortex (V1) (*Stringer et al., 2021*; *Pachitariu et al., 2019*; *Stringer et al., 2018a*; *Stringer et al., 2018b*), we find that the biological code is metabolically more efficient than others.

Overall, our results demonstrate that for a fixed learning rule, the neural sensory representation imposes an inductive bias over the space of learning tasks, allowing some tasks to be learned by a downstream neuron more sample-efficiently than others. Our work provides a concrete method for elucidating inductive biases of populations of neurons and suggest sample-efficient learning as a novel functional role for population codes.

## Results

### Problem setup

We denote vectors with bold lower-case symbols $\mathbf{r}$ and matrices $\mathbf{K}$ with bold upper-case symbols. We denote an average of a function $g(\boldsymbol{\theta})$ over random variable $\boldsymbol{\theta}$ as $\langle g(\boldsymbol{\theta}) \rangle_{\boldsymbol{\theta}}$. Euclidean inner products between vectors are denoted either as $\mathbf{x} \cdot \mathbf{y}$ or $\mathbf{x}^{\top} \mathbf{y}$ and real Euclidean $n$-space is denoted $\mathbb{R}^n$. Sets of variables are represented with $\{\cdot\}$.

We consider a population of $N$ neurons whose responses, $\{r_1(\boldsymbol{\theta}), r_2(\boldsymbol{\theta}), ..., r_N(\boldsymbol{\theta})\}$, vary with the input stimuli, which is parameterized by a vector variable $\boldsymbol{\theta} \in \mathbb{R}^d$, such as the orientation and the phase of a grating (*Figure 1A*). These responses define the population code. Throughout this work, we will mostly assume that this population code is deterministic: that identical stimuli generate identical neural responses.

From the population responses, a readout neuron learns its weights $\mathbf{w}$ to approximate a stimulus-response map, or a target function $y(\boldsymbol{\theta})$, such as one that classifies stimuli as apetitive ($y = 1$) or aversive ($y = -1$), or a more smooth one that attaches intermediate values of valence. We emphasize that in our model only the readout neuron performs learning, and the population code is assumed to be static through learning. Our theory is general in its assumptions about the structure of the population code and the stimulus-response map considered (Methods Theory of generalization), and can apply to many scenarios.

The readout neuron learns from $P$ stimulus-response examples with the goal of generalizing to previously unseen ones. Example stimuli $\boldsymbol{\theta}^{\mu}$, ($\mu = 1, \ldots, P$) are sampled from a probability distribution describing stimulus statistics $p(\boldsymbol{\theta})$. This distribution can be natural or artificially created, for example, for a laboratory experiment (Appendix Discrete stimulus spaces: finding eigenfunctions with matrix eigendecomposition). From the set of learning examples, $\mathcal{D} = \{\boldsymbol{\theta}^{\mu}, y(\boldsymbol{\theta}^{\mu})\}_{\mu=1}^{P}$, the readout weights are learned with the local, biologically-plausible delta-rule, $\Delta w_j = \eta \sum_{\mu} r_j(\boldsymbol{\theta}^{\mu})(y(\boldsymbol{\theta}^{\mu}) - \mathbf{r}(\boldsymbol{\theta}^{\mu}) \cdot \mathbf{w})$, where $\eta$ is a learning rate (*Figure 1A*). Learning with weight decay, which privileges readouts with smaller norm, can also be accommodated in our theory as we discuss in (Appendix Weight decay and ridge regression). With or without weight decay, the learning rule converges to a unique set of weights $\mathbf{w}^{*}(\mathcal{D})$ (Appendix Convergence of the delta-rule without weight decay). Generalization error with these weights is given by

$$E_g(\mathcal{D}) = \int p(\boldsymbol{\theta})\,(\mathbf{w}^*(\mathcal{D}) \cdot \mathbf{r}(\boldsymbol{\theta}) - y(\boldsymbol{\theta}))^2 d\boldsymbol{\theta}, \tag{1}$$

which quantifies the expected error of the trained readout over the entire stimulus distribution $p(\boldsymbol{\theta})$. This quantity will depend on the population code $\mathbf{r}(\boldsymbol{\theta})$, the target function $y(\boldsymbol{\theta})$ and the set of training examples $\mathcal{D}$. Our theoretical analysis of this model provides insights into how populations of neurons encode information and allow sample-efficient learning.

## Kernel structure of population codes controls learning performance

First, we note that the generalization performance of the learned readout on a given task depends entirely on the inner product kernel, defined by $K(\boldsymbol{\theta}, \boldsymbol{\theta}') = \frac{1}{N}\sum_{i=1}^{N} r_i(\boldsymbol{\theta}) r_i(\boldsymbol{\theta}')$, which quantifies the similarity of population responses to two different stimuli $\boldsymbol{\theta}$ and $\boldsymbol{\theta}'$. The kernel, or similarity matrix, encodes the geometry of the neural responses. Concretely, distances (in neural space) between population vectors for stimuli $\boldsymbol{\theta}, \boldsymbol{\theta}'$ can be computed from the kernel $\frac{1}{N}\|\mathbf{r}(\boldsymbol{\theta}) - \mathbf{r}(\boldsymbol{\theta}')\|^2 = K(\boldsymbol{\theta}, \boldsymbol{\theta}) + K(\boldsymbol{\theta}', \boldsymbol{\theta}') - 2K(\boldsymbol{\theta}, \boldsymbol{\theta}')$ (*Edelman, 1998*; *Kriegeskorte et al., 2008*; *Laakso and Cottrell, 2000*; *Kornblith et al., 2019*; *Cadieu et al., 2014*; *Pehlevan et al., 2018*). The fact that the solution to the learning problem only depends on the kernel is due to the convergence of the learning rule to a unique solution $\mathbf{w}^*(\mathcal{D})$ for the training set $\mathcal{D}$ (*Neal, 1994*; *Girosi et al., 1995*). The dataset-dependent fixed point $\mathbf{w}^*(\mathcal{D})$ of the learning rule is a linear combination of the population vectors on the dataset $\mathbf{w}^*(\mathcal{D}) = \frac{1}{N}\sum_{\mu=1}^{P} \alpha^\mu \mathbf{r}(\boldsymbol{\theta}^\mu)$. Thus, the learned function computed by the readout neuron is

$$f(\boldsymbol{\theta}) = \mathbf{w}^*(\mathcal{D}) \cdot \mathbf{r}(\boldsymbol{\theta}) = \sum_{\mu=1}^{P} \alpha^\mu \left(\frac{1}{N}\mathbf{r}(\boldsymbol{\theta}^\mu) \cdot \mathbf{r}(\boldsymbol{\theta})\right) = \sum_{\mu=1}^{P} \alpha^\mu K(\boldsymbol{\theta}^\mu, \boldsymbol{\theta}), \tag{2}$$

where the coefficient vector satisfies $\boldsymbol{\alpha} = \mathbf{K}^+ \mathbf{y}$ (Appendix Convergence of the delta-rule without weight decay), and the matrix $\mathbf{K}$ has entries $K_{\mu\nu} = K(\boldsymbol{\theta}^\mu, \boldsymbol{\theta}^\nu)$ and $y_\mu = y(\boldsymbol{\theta}^\mu)$. The matrix $\mathbf{K}^+$ is the pseudo-inverse of $\mathbf{K}$. In these expressions the population code only appears through the kernel $K$, showing that the kernel alone controls the learned response pattern. This result applies also to

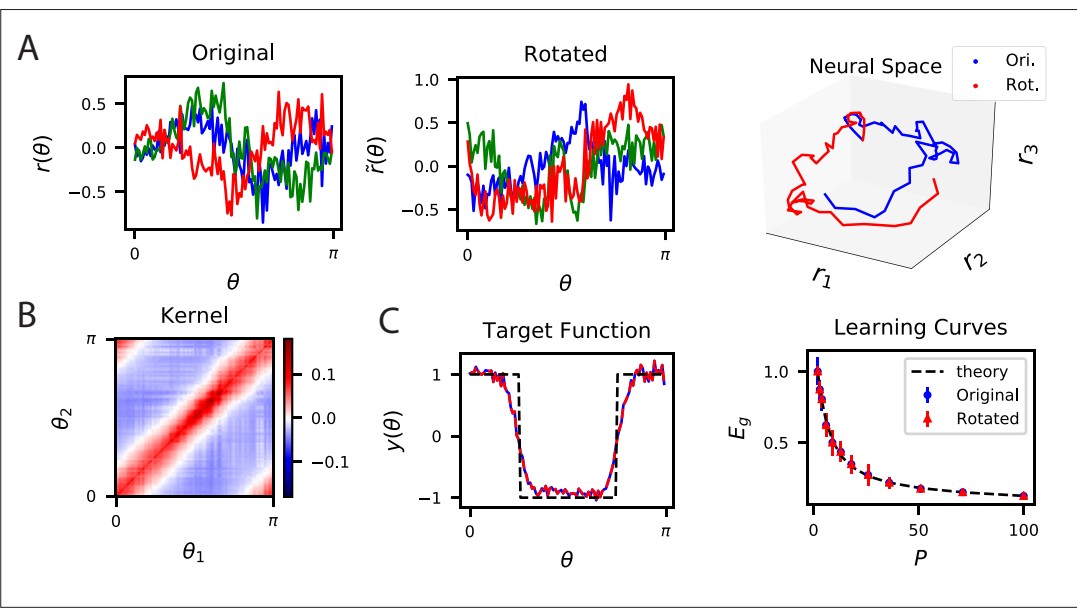

**Figure 2.** The inner product kernel controls the generalization performance of readouts. (**A**) Tuning curves $r(\theta)$ for three example recorded Mouse V1 neurons to varying static grating stimuli oriented at angle $\theta$ (*Stringer et al., 2021*; *Pachitariu et al., 2019*) (Left) are compared with a randomly rotated version (Middle) $\tilde{\mathbf{r}}(\theta)$ of the same population code. (Right) These two codes, original (Ori.) and rotated (Rot.) can be visualized as parametric trajectories in neural space. (**B**) The inner product kernel matrix has elements $K(\theta_1, \theta_2)$. The original V1 code and its rotated counterpart have identical kernels. (**C**) In a learning task involving uniformly sampled angles, readouts from the two codes perform identically, resulting in identical approximations of the target function (shown on the left as blue and red curves) and consequently identical generalization performance as a function of training set size $P$ (shown on right with blue and red points). The theory curve will be described in the main text.

nonlinear readouts (Appendix Convergence of Delta-rule for nonlinear readouts), showing that the kernel can control the learned solution in a variety of cases.

Since predictions only depend on the kernel, a large set of codes achieve identical desired performance on learning tasks. This is because the kernel is invariant with respect to rotation of the population code. An orthogonal transformation $\mathbf{Q}$ applied to a population code $\mathbf{r}(\boldsymbol{\theta})$ generates a new code $\tilde{\mathbf{r}}(\boldsymbol{\theta}) = \mathbf{Q}\mathbf{r}(\boldsymbol{\theta})$ with an identical kernel (Appendix Alternative neural codes with same kernel) since $\frac{1}{N}\tilde{\mathbf{r}}(\boldsymbol{\theta}) \cdot \tilde{\mathbf{r}}(\boldsymbol{\theta}') = \frac{1}{N}\mathbf{r}(\boldsymbol{\theta})^\top \mathbf{Q}^\top \mathbf{Q}\mathbf{r}(\boldsymbol{\theta}') = \frac{1}{N}\mathbf{r}(\boldsymbol{\theta}) \cdot \mathbf{r}(\boldsymbol{\theta}')$. Codes $\mathbf{r}(\boldsymbol{\theta})$ and $\tilde{\mathbf{r}}(\boldsymbol{\theta})$ will have identical readout performance on all possible learning tasks. We illustrate this degeneracy in *Figure 2* using a publicly available dataset which consists of activity recorded from ~20,000 neurons from the primary visual cortex of a mouse while shown static gratings (*Stringer et al., 2021*; *Pachitariu et al., 2019*). An original code $\mathbf{r}(\boldsymbol{\theta})$ is rotated to generate $\tilde{\mathbf{r}}(\boldsymbol{\theta})$ (*Figure 2A*) which have the same kernels (*Figure 2B*) and the same performance on a learning task (*Figure 2C*).

## Code-task alignment governs generalization

We next examine how the population code affects generalization performance of the readout. We calculated analytical expressions of the average generalization error in a task defined by the target response $y(\boldsymbol{\theta})$ after observing $P$ stimuli using methods from statistical physics (Methods Theory of generalization). Because the relevant quantity in learning performance is the kernel, we leveraged results from our previous work studying generalization in kernel regression (*Bordelon et al., 2020*; *Canatar et al., 2021*), and approximated the generalization error averaged over all possible realizations of the training dataset composed of $P$ stimuli, $E_g = \langle E_g(\mathcal{D}) \rangle_{\mathcal{D}}$. As $P$ increases, the variance in $E_g$ due to the composition of the dataset decreases, and our expressions become descriptive of the typical case. Our final analytical result is given in *Equation (11)* in Methods Theory of generalization. We provide details of our calculations in Methods Theory of generalization and Appendix Theory of generalization, and focus on their implications here.

One of our main observations is that given a population code $\mathbf{r}(\boldsymbol{\theta})$, the singular value decomposition of the code gives the appropriate basis to analyze the inductive biases of the readouts (*Figure 3A*). The tuning curves for individual neurons $r_i(\boldsymbol{\theta})$ form an $N$-by-$M$ matrix $\mathbf{R}$, where $M$, possibly infinite, is the number of all possible stimuli. We discuss the SVD for continuous stimulus spaces in Appendix Singular value decomposition of continuous population responses. The left-singular vectors (or principal axes) and singular values of this matrix have been used in neuroscience for describing lower dimensional structure in the neural activity and estimating its dimensionality, see e.g. (*Stopfer et al., 2003*; *Kato et al., 2015*; *Bathellier et al., 2008*; *Gallego et al., 2017*; *Sadtler et al., 2014*; *Stringer et al., 2018b*, *Stringer et al., 2021*; *Litwin-Kumar et al., 2017*; *Gao et al., 2017*; *Gao and Ganguli, 2015*). We found that the function approximation properties of the code are controlled by the singular values, or rather their squares $\{\lambda_k\}$ which give variances along principal axes, indexed in decreasing order, and the corresponding right singular vectors $\{\psi_k(\boldsymbol{\theta})\}$, which are also the kernel eigenfunctions (Methods Theory of generalization and Appendix Singular value decomposition of continuous population responses). This follows from the fact that learned response (*Equation (2)*) is only a function of the kernel $K$, and the eigenvalues $\lambda_k$ and orthonormal (uncorrelated) eigenfunctions $\psi_k(\boldsymbol{\theta})$ collectively define the code's inner-product kernel $K(\boldsymbol{\theta}, \boldsymbol{\theta}')$ through an eigendecomposition $K(\boldsymbol{\theta}, \boldsymbol{\theta}') = \frac{1}{N}\sum_{i=1}^N r_i(\boldsymbol{\theta})r_i(\boldsymbol{\theta}') = \sum_k \lambda_k \psi_k(\boldsymbol{\theta})\psi_k(\boldsymbol{\theta}')$ (*Mercer, 1909*) (Methods Theory of generalization and Appendix Theory of generalization).

Our analysis shows the existence of a bias in the readout towards learning certain target responses faster than others. The target response $y(\boldsymbol{\theta}) = \sum_k v_k \psi_k(\boldsymbol{\theta})$ and the learned readout response $f(\boldsymbol{\theta}) = \sum_k \hat{v}_k(\mathcal{D}) \psi_k(\boldsymbol{\theta})$ can be expressed in terms of these eigenfunctions $\psi_k$. Our theory shows that the readout's generalization is better if the target function $y(\boldsymbol{\theta})$ is aligned with the top eigenfunctions $\psi_k$, equivalent to $v_k^2$ decaying rapidly with $k$ (Appendix Spectral bias and code-task alignment). We formalize this notion by the following metric. Mathematically, generalization error $\langle E_g \rangle$ can be decomposed into normalized estimation errors $E_k$ for the coefficients of these eigenfunctions $\psi_k$, $\langle E_g \rangle_{\mathcal{D}} = \sum_k v_k^2 E_k$, where $E_k = \langle (\hat{v}_k(\mathcal{D}) - v_k)^2 \rangle_{\mathcal{D}} / v_k^2$. We found that the ordering of the eigenvalues $\lambda_k$ controls the rates at which these mode errors $E_k$ decrease as $P$ increases (Methods Theory of generalization, Appendix Spectral bias and code-task alignment), (*Bordelon et al., 2020*): $\lambda_k > \lambda_\ell \implies E_k < E_\ell$. Hence, larger eigenvalues mean lower generalization error for those normalized

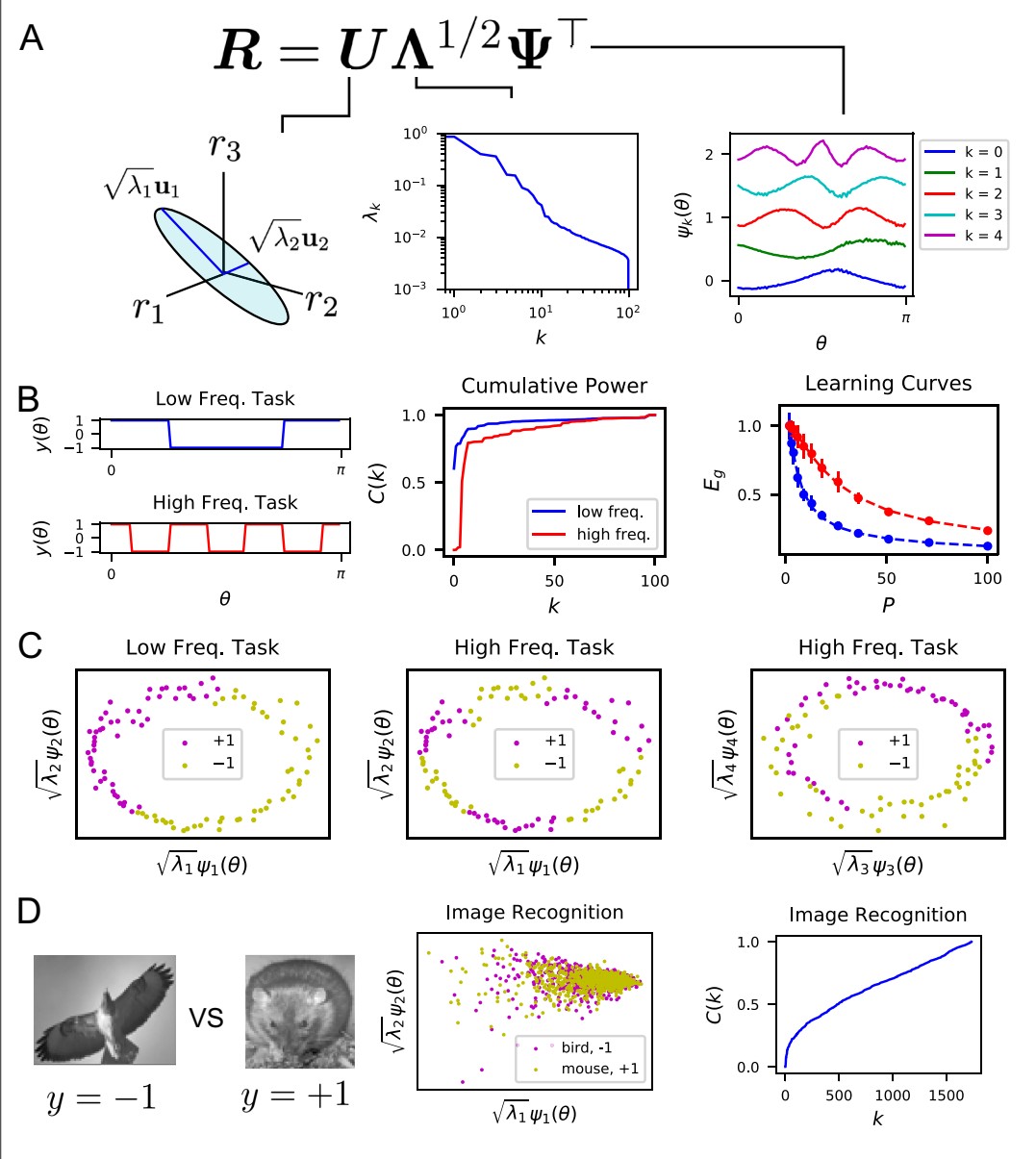

**Figure 3.** The singular value decomposition (SVD) of the population code reveals the structure and inductive bias of the code. (**A**) SVD of the response matrix $\mathbf{R}$ gives left singular vectors $\mathbf{u}_k$ (principal axes), kernel eigenvalues $\lambda_k$, and kernel eigenfunctions $\psi_k(\boldsymbol{\theta})$. The ordering of eigenvalues provides an ordering of which modes $\psi_k$ can be learned by the code from few training examples. The eigenfunctions were offset by 0.5 for visibility. (**B**) (Left) Two different learning tasks $y(\theta)$, a low frequency (blue) and high frequency (red) function, are shown. (Middle) The cumulative power distribution rises more rapidly for the low frequency task than the high frequency, indicating better alignment with top kernel eigenfunctions and consequently more sample-efficient learning as shown in the learning curves (right). Dashed lines show theoretical generalization error while dots and solid vertical lines are experimental average and standard deviation over 30 repeats. (**C**) The feature space representations of the low (left) and high (middle and right) frequency tasks. Each point represents the embedding of a stimulus response vector along the $k$-th principal axis $\mathbf{r}(\theta^\mu) \cdot \mathbf{u}_k = \sqrt{\lambda_k}\psi_k(\theta^\mu)$. The binary target value $\{\pm 1\}$ is indicated with the color of the point. The easy (left), low frequency task is well separated along the top two dimensions, while the hard, high frequency task is not linearly separable in two (middle) or even with four feature dimensions (right). (**D**) On an image discrimination task (recognizing birds vs mice), V1 has an entangled representation which does not allow good performance of linear readouts. This is evidenced by the top principal components (middle) and the slowly rising $C(k)$ curve (right).

The online version of this article includes the following figure supplement(s) for figure 3:

**Figure supplement 1.** Kernels, spectra and eigenfunctions are well preserved across a variety of bin counts.

mode errors $E_k$. We term this phenomenon the *spectral bias* of the readout. Based on this observation, we propose *code-task alignment* as a principle for good generalization. To quantify code-task alignment, we use a metric which was introduced in *Canatar et al., 2021* to measure the compatibility of a kernel with a learning task. This is the cumulative power distribution $C(k)$ which measures the total power of the target function in the top $k$ eigenmodes, normalized by the total power (*Canatar et al., 2021*):

$$C(k) = \frac{\sum_{\ell=1}^{k} v_{\ell}^2}{\sum_{\ell=1}^{\infty} v_{\ell}^2}. \tag{3}$$

Stimulus-response maps that have high alignment with the population code's kernel will have quickly rising cumulative power distributions $C(k)$, since a large proportion of power is placed in the top modes. Target responses with high $C(k)$ can be learned with fewer training samples than target responses with low $C(k)$ since the mode errors $E_k$ are ordered for all $P$ (Appendix Spectral bias and code-task alignment).

## Probing learning biases in neural data

Our theory can be used to probe the learning biases of neural populations. Here, we provide various examples of this using publicly available calcium imaging recordings from mouse primary visual cortex (V1). Our examples illustrate how our theory can be used to analyze neural data.

We first analyzed population responses to static grating stimuli oriented at an angle $\theta$ (*Stringer et al., 2021*; *Pachitariu et al., 2019*). We found that the kernel eigenfunctions have sinusoidal shape with differing frequency. The ordering of the eigenvalues and eigenfunctions in *Figure 3A* (and *Figure 3—figure supplement 1*) indicates a frequency bias: lower frequency functions of $\theta$ are easier to estimate at small sample sizes.

We tested this idea by constructing two different orientation discrimination tasks shown in *Figure 3B and C*, where we assign static grating orientations to positive or negative valence with different frequency square-wave functions of $\theta$. We trained the readout using a subset of the experimentally measured neural responses, and measured the readout's generalization performance. We found that the cumulative power distribution for the low frequency task has a more rapidly rising $C(k)$ (*Figure 3B*). Using our theory of generalization, we predicted learning curves for these two tasks, which express the generalization error as a function of the number of sampled stimuli $P$. The error for the low frequency task is lower at all sample sizes than the hard, high-frequency task. The theoretical predictions and numerical experiments show perfect agreement (*Figure 3B*). More intuition can be gained by visualizing the projection of the neural response along the top principal axes (*Figure 3C*). For the low-frequency task, the two target values are well separated along the top two axes. However, the high-frequency task is not well separated along even the top four axes (*Figure 3C*).

Using the same ideas, we can use our theory to get insight into tasks which the V1 population code is ill-suited to learn. For the task of identifying mice and birds (*Stringer et al., 2018b*, *Stringer et al., 2018a*) the linear rise in cumulative power indicates that there is roughly equal power along all kernel eigenfunctions, indicating that the representation is poorly aligned to this task (*Figure 3D*).

To illustrate how our approach can be used for different learning problems, we evaluate the ability of linear readouts to reconstruct natural images from neural responses to those images (*Figure 4*). The ability to reconstruct sensory stimuli from a neural code is an influential normative principle for primary visual cortex (*Olshausen and Field, 1997*). Here, we ask which aspects of the presented natural scene stimuli are easiest to learn to reconstruct. Since mouse V1 neurons tend to be selective towards low spatial frequency bands (*Niell and Stryker, 2008Bonin et al., 2011*; *Vreysen et al., 2012*), we consider reconstruction of band-pass filtered images with spatial frequency wave-vector $\mathbf{k} \in \mathbb{R}^2$ constrained to an annulus $|\mathbf{k}| \in \left[\sqrt{\max(s_{max}^2 - r^2, 0)}, s_{max}\right]$ for $r = 0.2$ (in units of $\text{pixels}^{-1}$) and plot the cumulative power $C(k)$ associated with each choice of the upper limit $s_{max}$ (*Figure 4C and D*). The frequency cutoffs were chosen in this way to preserve the volume in Fourier space to $V_{\mathbf{k}} = \pi r^2$ for $r < s_{max}$, which quantifies the dimension of the function space. We see that the lower frequency band-limited images are easier to reconstruct, as evidenced by their cumulative power $C(k)$ and learning curves $E_g$ (*Figure 4D and E*). This reflects the fact that the population preferentially encodes low spatial frequency content in the image (*Figure 4F*). Experiments with additional values of $r$ are

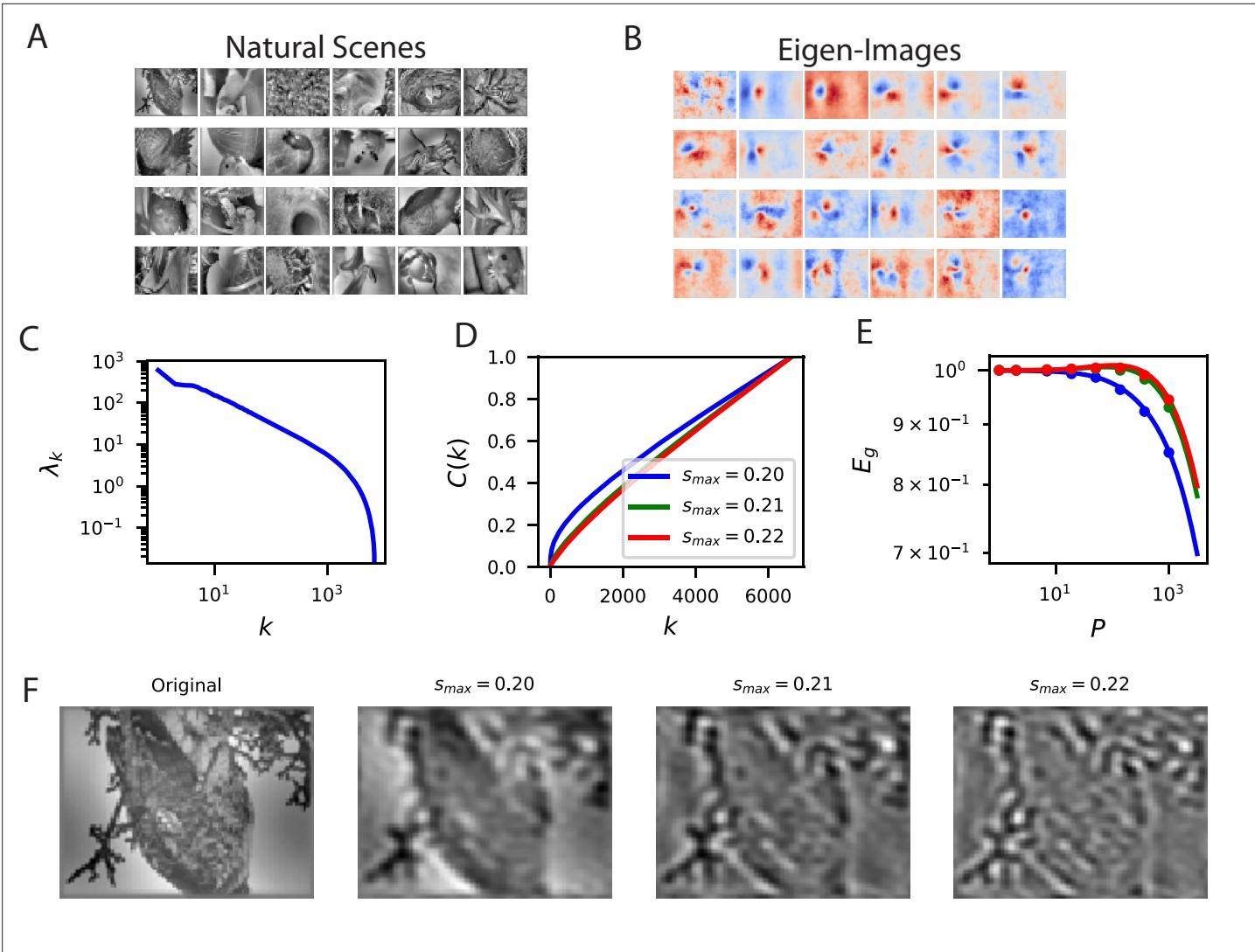

**Figure 4.** Reconstructing filtered natural images from V1 responses reveals preference for low spatial frequencies. (**A**) Natural scene stimuli $\boldsymbol{\theta}$ were presented to mice and V1 cells were recorded. (**B**) The images weighted by the top eigenfunctions $\mathbf{v}_k = \left\langle \psi_k(\boldsymbol{\theta})\boldsymbol{\theta} \right\rangle_{\boldsymbol{\theta}}$. These "eigenimages" collectively define the difficulty of reconstructing images through readout. (**C**) The kernel spectrum of the V1 code for natural images. (**D**) The cumulative power curves for reconstruction of band-pass filtered images. Filters preserve spatial frequencies in the range $|\mathbf{k}| \in \left[ \sqrt{\max(s_{max}^2 - 0.2^2, 0)}, s_{max} \right]$, chosen to preserve volume in Fourier space as $s_{max}$ is varied. (**E**) The learning curves obey the ordering of the cumulative power curves. The images filtered with the lowest band-pass cutoff are easiest to reconstruct from the neural responses. (**F**) Examples of a band-pass filtered image with different preserved frequency bands.

The online version of this article includes the following figure supplement(s) for figure 4:

**Figure supplement 1.** The inductive bias to reconstruct low spatial frequency components of natural scenes from the population responses holds over several band-pass filters of the form $|\mathbf{k}| \in \left[ \sqrt{\max(s_{max}^2 - r^2, 0)}, s_{max} \right]$.

provided in the *Figure 4—figure supplement 1* with additional details found in the Appendix Visual scene reconstruction task.

## Mechanisms of spectral bias and code-task alignment in a simple model of V1

How do population level inductive biases arise from properties of single neurons? To illustrate that a variety of mechanisms may be involved in a complex manner, we study a simple model of V1 to elucidate neural mechanisms that lead to the low frequency bias at the population level. In particular, we focus on neural nonlinearities and selectivity profiles.

We model responses of V1 neurons as photoreceptor inputs passed through Gabor filters and a subsequent experimentally motivated power-law nonlinearity (*Adelson and Bergen, 1985*; *Olshausen and Field, 1997*; *Rumyantsev et al., 2020*), modeling a population of orientation selective simple cells (*Figure 5A*) (see Appendix A simple feedforward model of V1). In this model, the kernel for static gratings with orientation $\theta \in [0, \pi]$ is of the form $K(\theta, \theta') = \kappa(|\theta - \theta'|)$, and, as a consequence, the eigenfunctions of the kernel in this setting are Fourier modes. The eigenvalues, and hence the strength of the spectral bias, are determined by the nonlinearity as we discuss in Appendix Gabor model spectral bias and fit to V1 data. We numerically fit the parameters of the nonlinearity to the V1 responses and use these parameters our investigations in *Figure 5—figure supplement 1*.

Next, to further illustrate the importance of code-task alignment, we study how invariances in the code to stimulus variations may affect the learning performance. We introduce complex cells in addition to simple cells in our model with proportion $s \in [0, 1]$ of simple cells (Appendix Gabor model spectral bias and fit to V1 data; *Figure 5A*), and allow phase, $\phi$, variations in static gratings. We use the energy model (*Adelson and Bergen, 1985*; *Simoncelli and Heeger, 1998*) to capture the phase invariant complex cell responses (Appendix Phase variation, complex cells and invariance, complex cell populations are phase invariant). We reason that in tasks that do not depend on phase information, complex cells should improve sample efficiency.

In this model, the kernel for the V1 population is a convex combination of the kernels for the simple and complex cell populations $K_{V1}(\theta, \theta', \phi, \phi') = sK_s(\theta, \theta', \phi, \phi') + (1 - s)K_c(\theta, \theta')$ where $K_s$ is the kernel for a pure simple cell population that depends on both orientation and phase, and $K_c$ is the kernel of a pure complex cell population that is invariant to phase (Appendix Complex cell populations are phase invariant). *Figure 5C* shows top kernel eigenfunctions for various values of $s$ elucidating inductive bias of the readout.

*Figure 5D and E* show generalization performance on tasks with varying levels of dependence on phase and orientation. On pure orientation discrimination tasks, increasing the proportion of complex cells by decreasing $s$ improves generalization. Increasing the sensitivity to the nuisance phase variable, $\phi$, only degrades performance. The cumulative power curve is also maximized at $s = 0$. However, on a task which only depends on the phase, a pure complex cell population cannot generalize, since variation in the target function due to changes in phase cannot be explained in the codes' responses. In this setting, a pure simple cell population attains optimal performance. The cumulative power curve is maximized at $s = 1$. Lastly, in a nontrivial hybrid task which requires utilization of both variables $\theta, \phi$, an optimal mixture $s$ exists for each sample budget $P$ which minimizes the generalization error. The cumulative power curve is maximized at different $s$ values depending on $k$, the component of the target function. This is consistent with an optimal heterogenous mix, because components of the target are learned successively with increasing sample size. V1 must code for a variety of possible tasks and we can expect a nontrivial optimal simple cell fraction $s$. We conclude that the degree of invariance required for the set of natural tasks, and the number of samples determine the optimal simple cell, complex cell mix. We also considered a more realistic model where the relative selectivity of each visual cortex neuron to phase $\phi$, measured with the F1/F0 ratio takes on a continuum of possible values with some cells more invariant to phase and some less invariant. In (Appendix Energy model with partially phase-selective cells, *Figure 5—figure supplement 3*) we discuss a simple adaptation of the energy model which can interpolate between a population of entirely simple cells and a population of entirely complex cells, giving diverse selectivity for the intermediate regime. We show that this model reproduces the inductive bias of *Figure 5*.

## Small and large sample size behaviors of generalization

Recently, *Stringer et al., 2018b* argued that the input-output differentiability of the code, governed by the asymptotic rate of spectral decay, may be enabling better generalization. Our results provide a more nuanced view of the relation between generalization and kernel spectra. First, generalization with low sample sizes crucially depend on the top eigenvalues and eigenfunctions of the code's kernel, not the tail. Second, generalization requires alignment of the code with the task of interest. Non-differentiable codes can generalize well if there is such an alignment. To illustrate these points, here, we provide examples where asymptotic conditions on the kernel spectrum are insufficient to describe generalization performance for small sample sizes (*Figure 6*, *Figure 6—figure supplement*

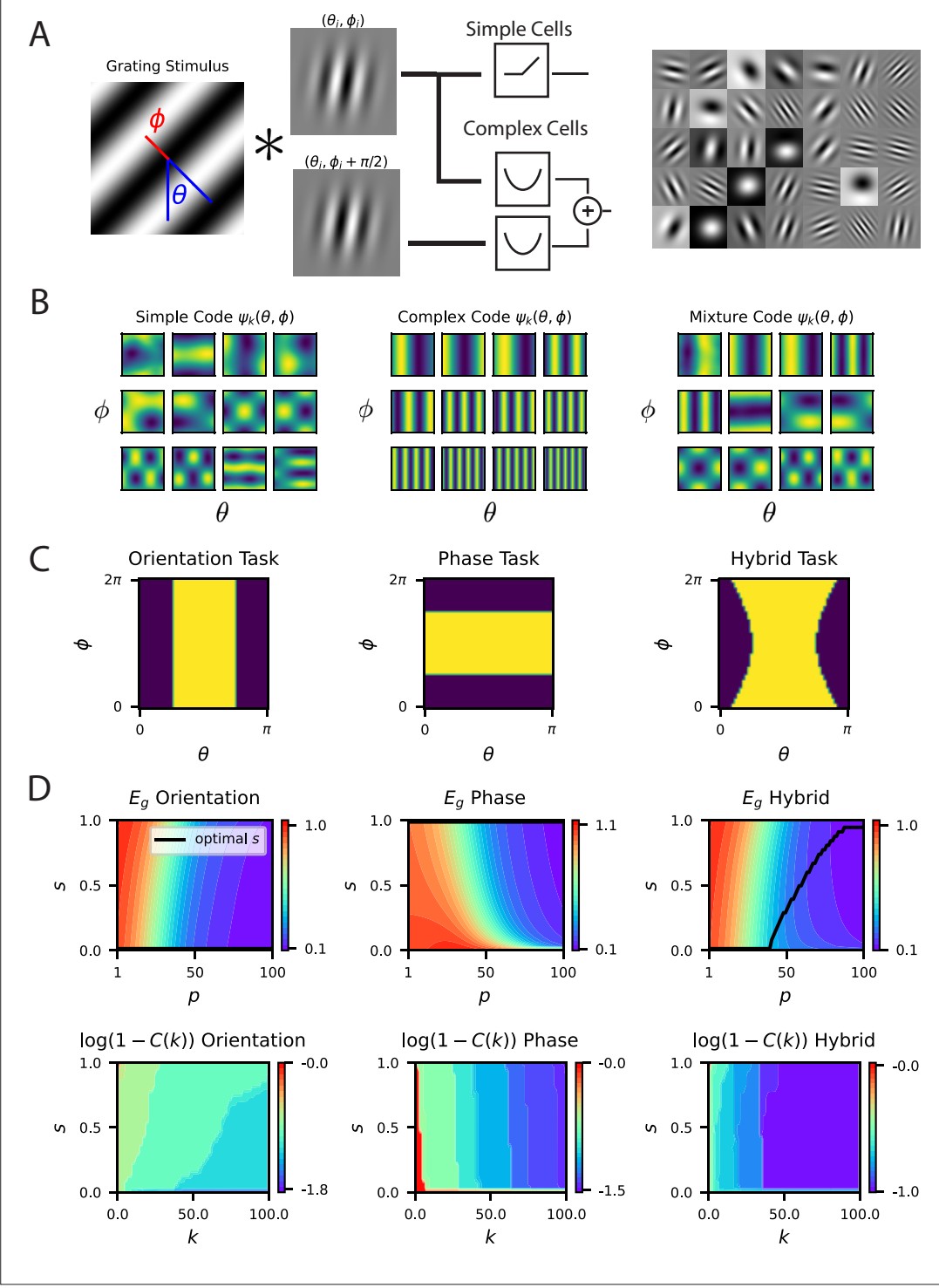

**Figure 5.** A model of V1 as a bank of Gabor filters recapitulates experimental inductive bias. (**A**) Gabor filtered inputs are mapped through nonlinearity. A grating stimulus (left) with orientation $\theta$ and phase $\phi$ is mapped through a circuit of simple and complex cells (middle). Some examples of randomly sampled Gabor filters (right) generate preferred orientation tuning of neurons in the population. (**B**) We plot the top 12 eigenfunctions $\psi_k(\theta, \phi)$ (modes) for pure simple cell population, pure complex cell population and a mixture population with half simple and half complex cells. The pure complex cell population has all eigenfunctions independent of phase $\phi$. A pure simple cell population $s = 1$ or mixture codes $0 < 1$ depend on both orientation phase in a nontrivial way. (**C**) Three tasks are visualized, where color indicates the binary target value $\pm 1$. The left task only depends on orientation

*Figure 5 continued on next page*

*Figure 5 continued*

stimulus variable $\theta$, the middle only depends on phase $\phi$, the hybrid task (right) depends on both. (**D**) (top) Generalization error and cumulative power distributions for the three tasks as a function of the simple-complex cell mixture parameter $s$.

The online version of this article includes the following figure supplement(s) for figure 5:

**Figure supplement 1.** The fit of our simple cell model to the mouse V1 code.

**Figure supplement 2.** Nonlinear Rectification and proportion of simple and complex cells influences the inductive bias of the population code.

**Figure supplement 3.** The modified energy based model with partial selectivity to phase $\phi$ reproduces the inductive bias of partial invariance.

---

*1* and Appendix Asymptotic power law scaling of learning curves), and where non-differentiable kernels generalize better than differentiable kernels (***Figure 6—figure supplement 2***).

Our example demonstrates how a code allowing good generalization for large sample sizes can be disadvantageous for small sizes. In ***Figure 6A***, we plot three different populations of neurons with smooth (infinitely differentiable) tuning curves that tile a periodic stimulus variable, such as the direction of a moving grating. The tuning width, $\sigma$, of the tuning curves strongly influences the structure of these codes: narrower widths have more high frequency content as we illustrate in a random 3D projection of the population code for $\theta \in [0, 2\pi]$ (***Figure 6A***). Visualization of the corresponding (von Mises) kernels and their spectra are provided in ***Figure 6B***. The width of the tuning curves control bandwidths of the kernel spectra ***Figure 6B***, with narrower curves having an later decay in the spectrum and higher high frequency eigenvalues. These codes can have dramatically different generalization performance, which we illustrate with a simple "bump" target response (***Figure 6C***). In this example, for illustration purposes, we let the network learn with a delta-rule with a weight decay, leading to a regularized kernel regression solution (Appendix Weight decay and ridge regression). For a sample size of $P = 10$, we observe that codes with too wide or too narrow tuning curves (and kernels) do not perform well, and there is a well-performing code with an optimal tuning curve width $\sigma$, which is compatible with the width of the target bump, $\sigma_T$. We found that optimal $\sigma$ is different for each $P$ (***Figure 6C***). In the large-$P$ regime, the ordering of the performance of the three codes are reversed (***Figure 6C***). In this regime generalization error scales in a power law (Appendix Asymptotic power law scaling of learning curves) and the narrow code, which performed worst for $P \sim 10$, performs the best. This example demonstrates that asymptotic conditions on the tail of the spectra are insufficient to understand generalization in the small sample size limit. The bulk of the kernel's spectrum needs to match the spectral structure of the task to generalize efficiently in the low-sample size regime. However, for large sample sizes, the tail of the eigenvalue spectrum becomes important. We repeat the same exercise and draw the same conclusions for a non-differentiable kernel (Laplace) (***Figure 6—figure supplement 1***) showing that these results are not an artifact of the infinite differentiability of von Mises kernels. We further provide examples where non-differentiable kernels generalizing better than differentiable kernels in ***Figure 6—figure supplement 2***.

## Time-dependent neural codes

Our framework can directly be extended to learning of arbitrary time-varying functions of time-varying inputs from an arbitrary spatiotemporal population code (Methods RNN experiment, Appendix Time dependent neural codes). In this setting, the population code $\mathbf{r}(\{\theta(t)\}, t)$ is a function of an input stimulus sequence $\theta(t)$ and possibly its entire history, and time $t$. A downstream linear readout $f(\{\theta\}, t) = \mathbf{w} \cdot \mathbf{r}(\{\theta\}, t)$ learns a target sequence $y(\{\theta\}, t)$ from a total of $\mathcal{P}$ examples that can come at any time during any sequence.

As a concrete example, we focus on readout from a temporal population code generated by a recurrent neural network in a task motivated by a delayed reach task (***Ames et al., 2019***; ***Figure 7A and B***). In this task, the network is presented for a short time an input cue sequence coding an angular variable which is drawn randomly from a distribution (***Figure 7C***). The recurrent neural network must remember this angle and reproduce an output sequence which is a simple step function whose height depends on the angle which begins after a time delay from the cessation of input stimulus and lasts for a short time (***Figure 7D***).

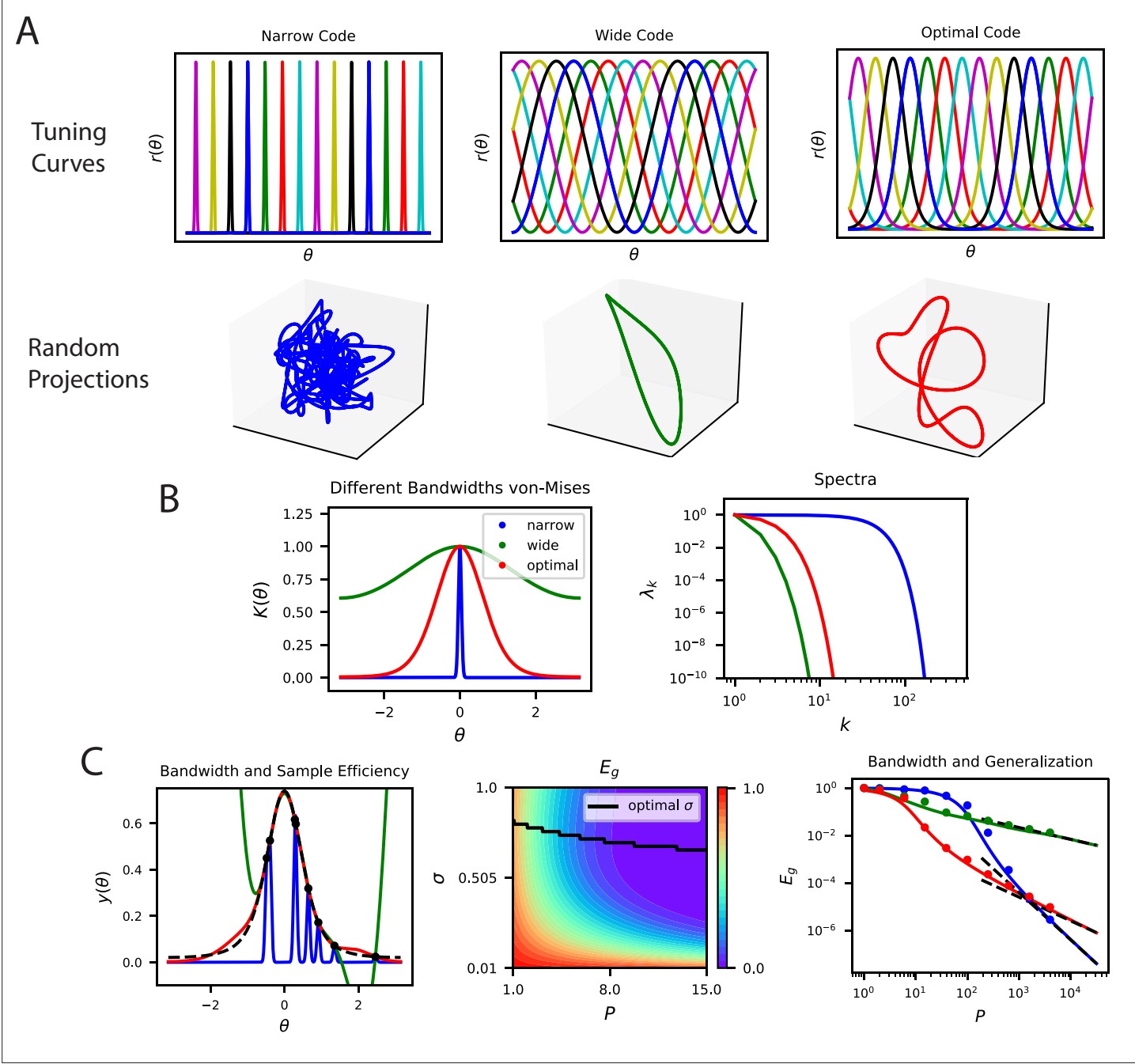

**Figure 6.** The top eigensystem of a code determines its low-$P$ generalization error. (**A**) A periodic variable is coded by a population of neurons with tuning curves of different widths (top). Narrow, wide and optimal refers to the example in C. These codes are all smooth (infinitely differentiable) but have very different feature space representations of the stimulus variable $\theta$, as random projections reveal (below). (**B**) (left) The population codes in the above figure induce von Mises kernels $K(\theta) \propto e^{\cos(\theta)/\sigma^2}$ with different bandwidths $\sigma$. (right) Eigenvalues of the three kernels. (**C**) (left) As an example learning task, we consider estimating a 'bump' target function. The optimal kernel (red, chosen as optimal bandwidth for $P = 10$) achieves a better generalization error than either the wide (green) or narrow (blue) kernels. (middle) A contour plot shows generalization error for varying bandwidth $\sigma$ and sample size $P$. (right) The large $P$ generalization error scales in a power law. Solid lines are theory, dots are simulations averaged over 15 repeats, dashed lines are asymptotic power law scalings described in main text. Same color code as B and C-left.

The online version of this article includes the following figure supplement(s) for figure 6:

**Figure supplement 1.** Bandwidth (spectral bulk) also governs generalization in non-smooth codes.

**Figure supplement 2.** Non-differentiable kernels can generalize better than infinitely differentiable kernels in a variety of contexts.

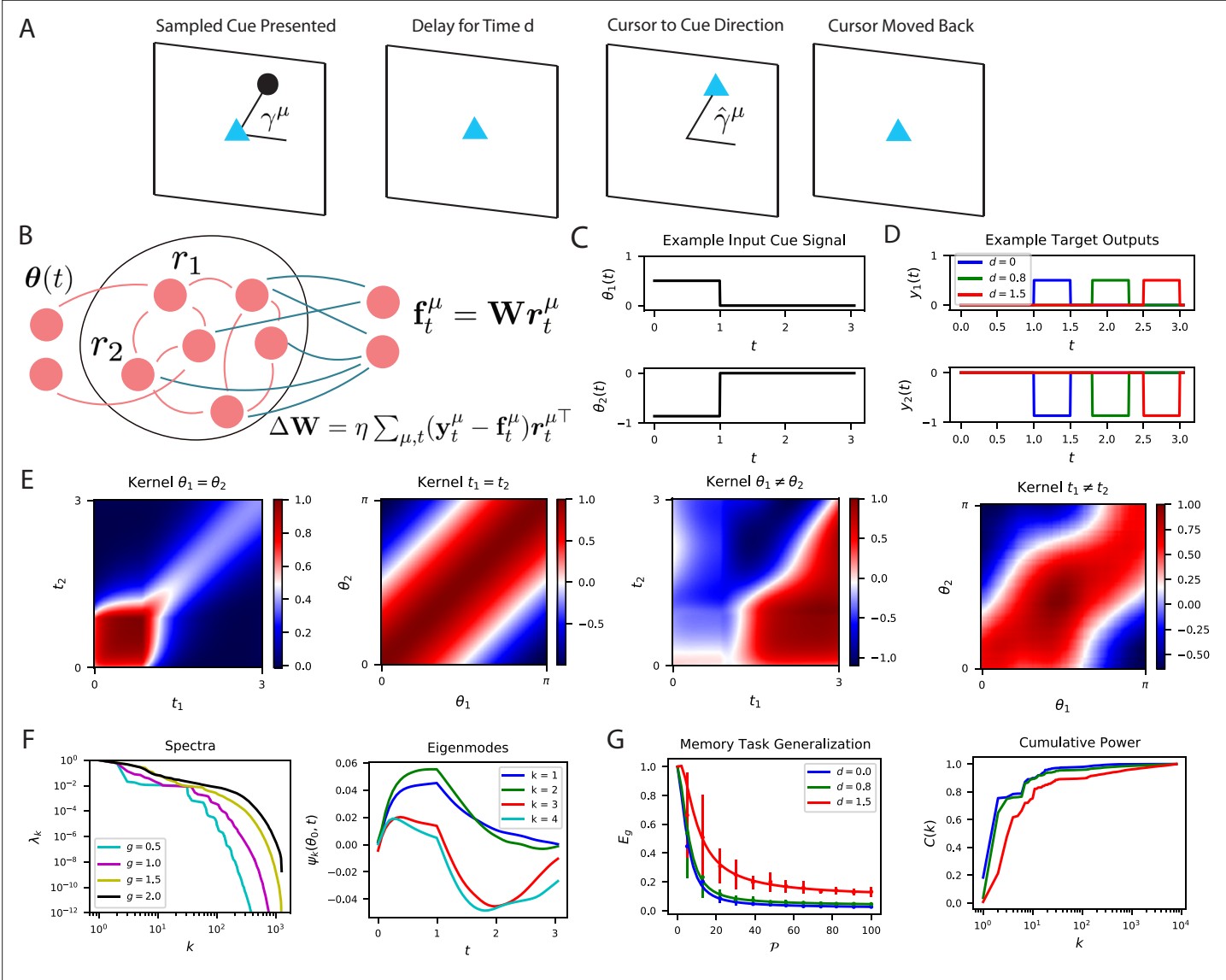

**Figure 7.** The performance of time-dependent codes when learning dynamical systems can be understood through spectral bias. (**A**) We study the performance of time dependent codes on a delayed response task which requires memory retrieval. A cue (black dot) is presented at an angle $\gamma^\mu$. After a delay time $d$, the cursor position (blue triangle) must be moved to the remembered cue position and then subsequently moved back to the origin after a short time. (**B**) The readout weights (blue) of a time dependent code can be learned through a modified delta rule. (**C**) Input is presented to the network as a time series which terminates at $t = 1$. The sequences are generated by drawing an angle $\gamma^\mu \sim \text{Uniform}[0, 2\pi]$ and using two step functions as input time-series that code for the cosine and the sine of the angle (Methods RNN experiment, Appendix Time dependent neural codes). We show an example of the one of the variables in a input sequence. (**D**) The target functions for the memory retrieval task are step functions delayed by a time $d$. (**E**) The kernel $K_{t,t'}^{\mu,\mu'}$ compares the code for two sequences at two distinct time points. We show the time dependent kernel for identical sequences (left) and the stimulus dependent kernel for equal time points (middle left) as well as for non-equal stimuli (middle right) and non-equal time (right). (**F**) The kernel can be diagonalized, and the eigenvalues $\lambda_k$ determine the spectral bias of the reservoir computer (left). We see that higher gain $g$ networks have higher dimensional representations. The 'eigensystems' $\psi_k(\theta^\mu, t)$ are functions of time and cue angle. We plot only $\mu = 0$ components of top systems $k = 1, 2, 3, 4$ (right). (**G**) The readout is trained to approximate a target function $y^\mu(t)$, which requires memory of the presented cue angle. (left) The theoretical (solid) and experimental (vertical errorbar, 100 trials) generalization error $E_g$ are plotted for the three delays $d$ against training sample size $\mathcal{P}$. (right) The ordering of $E_g$ matches the ordering of the $C(k)$ curves as expected.

The kernel induced by the spatiotemporal code is shown in *Figure 7E*. The high dimensional nature of the activity in the recurrent network introduces complex and rich spatiotemporal similarity structure. *Figure 7F* shows the kernel's eigensystem, which consists of stimulus dependent time-series $\psi_k(\{\boldsymbol{\theta}\}; t)$ for each eigenvalue $\lambda_k$. An interesting link can be made with this eigensystem and

linear low-dimensional manifold dynamics observed in several cortical areas (*Stopfer et al., 2003*; *Kato et al., 2015*; *Gallego et al., 2017*; *Cunningham and Yu, 2014*; *Sadtler et al., 2014*; *Gao and Ganguli, 2015*; *Gallego et al., 2018*; *Chapin and Nicolelis, 1999*; *Bathellier et al., 2008*). The kernel eigenfunctions also define the latent variables obtained through a singular value decomposition of the neural activity $\mathbf{r}(\{\boldsymbol{\theta}\}; t) = \sum_k \sqrt{\lambda_k} \mathbf{u}_k \psi_k(\{\boldsymbol{\theta}\}; t)$ (*Gallego et al., 2017*). With enough samples, the readout neuron can learn to output the desired angle with high fidelity (*Figure 7G*). Unsurprisingly, tasks involving long time delays are more difficult and exhibit lower cumulative power curves. Consequently, the generalization error for small delay tasks drops much more quickly with increasing samples $\mathcal{P}$.

## Biological codes are metabolically more efficient and more selective than other codes with identical kernels

Although, the performance of linear readouts may be invariant to rotations that preserve kernels (*Figure 2*), metabolic efficiency may favor certain codes over others (*Barlow, 1961*; *Atick and Redlich, 1992*; *Attneave, 1954*; *Olshausen and Field, 1997*; *Simoncelli and Olshausen, 2001*), reducing degeneracy in the space of codes with identical kernels. To formalize this idea, we define $\boldsymbol{\delta}$ to be the vector of spontaneous firing rates of a population of neurons, and $\mathbf{s}^\mu = \mathbf{r}(\boldsymbol{\theta}^\mu) + \boldsymbol{\delta}$ be the spiking rate vector in response to a stimulus $\boldsymbol{\theta}^\mu$. The vector $\boldsymbol{\delta}$ ensures that neural responses are non-negative. The modulation with respect to the spontaneous activity, $\mathbf{r}(\boldsymbol{\theta}^\mu)$, gives the population code and defines the kernel, $K(\boldsymbol{\theta}^\mu, \boldsymbol{\theta}^\mu) = \frac{1}{N}\mathbf{r}(\boldsymbol{\theta}^\mu) \cdot \mathbf{r}(\boldsymbol{\theta}^\nu)$. To avoid confusion with $\mathbf{r}(\boldsymbol{\theta}^\mu)$, we will refer to $\mathbf{s}^\mu$ as total spiking activity. We propose that population codes prefer smaller spiking activity subject to a fixed kernel. In other words, because the kernel is invariant to any change of the spontaneous firing rates and left rotations of $\mathbf{r}(\boldsymbol{\theta})$, the orientation and shift of the population code $\mathbf{r}(\boldsymbol{\theta})$ should be chosen such that the resulting total spike count $\sum_{i=1}^{N} \sum_{\mu=1}^{P} s_i^\mu$ is small.

We tested whether biological codes exhibit lower total spiking activity than others exhibiting the same kernel on mouse V1 recordings, using deconvolved calcium activity as a proxy for spiking events (*Stringer et al., 2021*; *Pachitariu et al., 2019*; *Pachitariu et al., 2018*) (Methods Data analysis; *Figure 8*). To compare the experimental total spiking activity to other codes with identical kernels, we computed random rotations of the neural responses around spontaneous activity, $\tilde{\mathbf{r}}(\boldsymbol{\theta}^\mu) = \mathbf{Q}\mathbf{r}(\boldsymbol{\theta}^\mu)$, and added the $\boldsymbol{\delta}$ that minimizes total spiking activity and maintains its nonnegativity (Methods Generating RROS codes). We refer to such an operation as RROS (random rotation and optimal shift), and a code generated by an RROS operation as an RROS code. The matrix $\mathbf{Q}$ is a randomly sampled orthogonal matrix (*Anderson et al., 1987*). In other words, we compare the true code to the most metabolically efficient realizations of its random rotations. This procedure may result in an increased or decreased total spike count in the code, and is illustrated in a synthetic dataset in *Figure 8A*. We conducted this procedure on subsets of various sizes of mouse V1 neuron populations, as our proposal should hold for any subset of neurons (Methods Generating RROS codes), and found that the true V1 code is much more metabolically efficient than randomly rotated versions of the code (*Figure 8B and C*). This finding holds for both responses to static gratings and to natural images as we show in *Figure 8B and C* respectively.

To further explore metabolic efficiency, we posed an optimization problem which identifies the most efficient code with the same kernel as the biological V1 code. This problem searches over rotation matrices $\mathbf{Q}$ and finds the $\mathbf{Q}$ matrix and off-set vector $\boldsymbol{\delta}$ which gives the lowest cost $\sum_{i\mu} s_i^\mu$ (Methods Comparing sparsity of population codes) (*Figure 8*). Although the local optimum identified with the algorithm is lower in cost than the biological code, both the optimal and biological codes are significantly displaced from the distribution of random codes with same kernel. Our findings do not change when data is preprocessed with an alternative strategy, an upper bound on neural responses is imposed on rotated codes, or subsets of stimuli are considered (*Figure 8—figure supplement 1*). We further verified these results on electrophysiological recordings of mouse visual cortex from the Allen Institute Brain Observatory (*de Vries et al., 2020*), (*Figure 8—figure supplement 2*). Overall, the large disparity in total spiking activity between the true and randomly generated codes with identical kernels suggests that metabolic constraints may favor the biological code over others that realize the same kernel.

The disparity between the true biological code and the RROS code is not only manifested in terms of total activity level, but also in terms of single neuron and single stimulus sparseness measures,

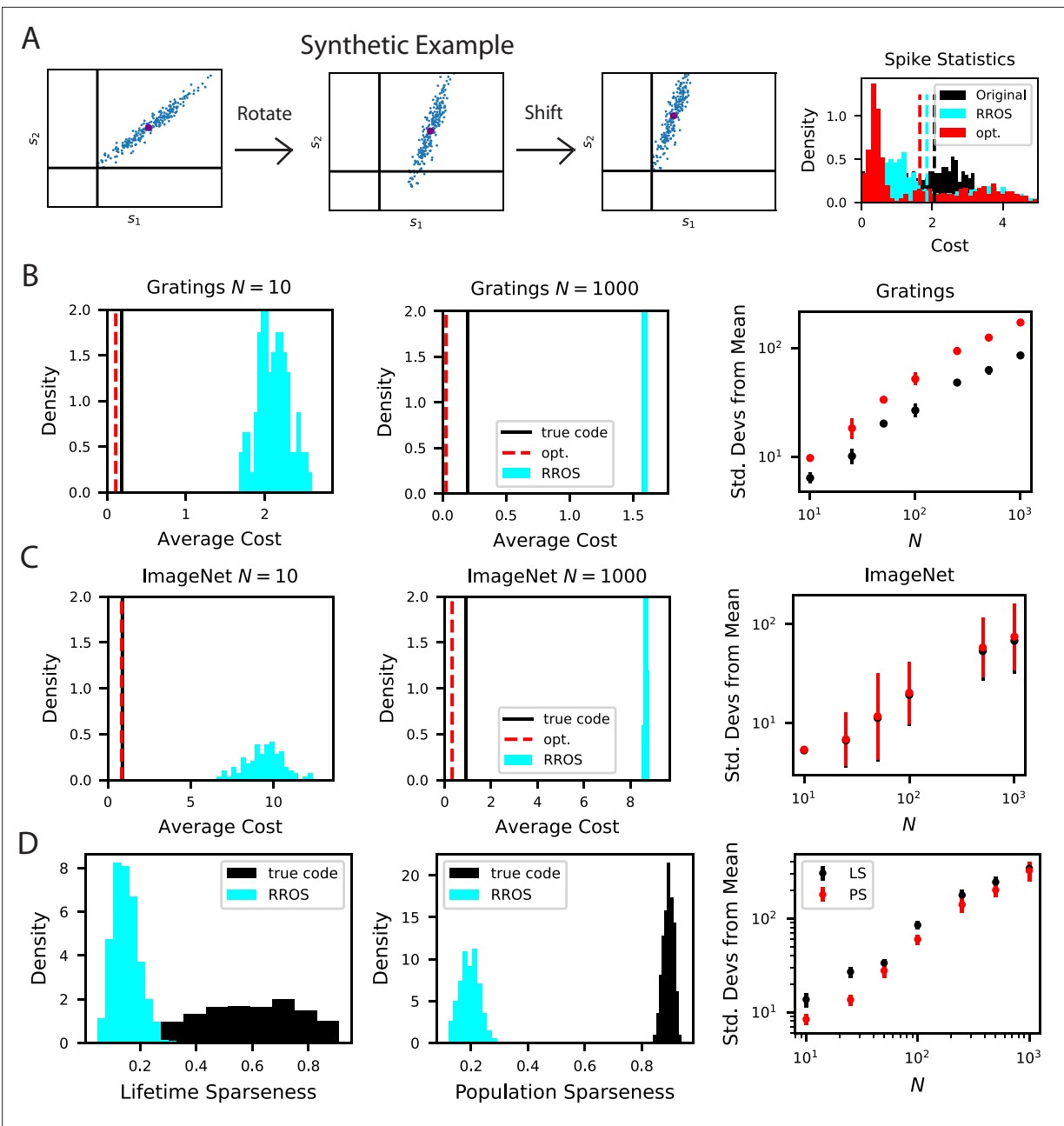

**Figure 8.** The biological code is more metabolically efficient than random codes with same inductive biases. (**A**) We illustrate our procedure in a synthetic example. A non-negative population code (left) can be randomly rotated about its spontaneous firing rate (middle), illustrated as a purple dot, and optimally shifted to a new non-negative population code (right). If the kernel is measured about the spontaneous firing rate, these transformations leave the inductive bias of the code invariant but can change the total spiking activity of the neural responses. We refer to such an operation as random rotation + optimal shift (RROS). We also perform gradient descent over rotations and shifts, generating an optimized code (opt). (**B**) Performing RROS on $N$ neuron subsamples of experimental Mouse V1 recordings (**Stringer et al., 2021**; **Pachitariu et al., 2019**), shows that the true code has much lower average cost $\frac{1}{NP} \sum_{i\mu} s_i^\mu$ compared to random rotations of the code. The set of possible RROS transformations (Methods Generating RROS codes, and Methods Comparing sparsity of population codes) generates a distribution over average cost, which has higher mean than the true code. We also optimize metabolic cost over the space of RROS transformations, which resulted in the red dashed lines. We plot the distance (in units of standard deviations) between the cost of the true and optimal codes and the cost of randomly rotated codes for different neuron subsample sizes $N$. (**C**) The same experiment performed on Mouse V1 responses to ImageNet images from 10 relevant classes (**Stringer et al., 2018a**; **Stringer et al., 2018b**). (**D**) The *lifetime* (LS) and *population sparseness* (PS) levels (Methods Lifetime and population sparseness) are higher for the Mouse V1 code than for a RROS code. The distance between average LS and PS of true code and RROS codes increases with $N$.

The online version of this article includes the following figure supplement(s) for figure 8:

*Figure 8 continued on next page*

*Figure 8 continued*

**Figure supplement 1.** Our metabolic efficiency finding is robust to different pre-processing techniques and upper bounds on neural firing.

**Figure supplement 2.** The observation that randomly oriented codes with the same kernel require higher spike counts than the original code is reproduced from electrophysiological recordings of Mouse visual cortex (VISp and VISal) from the AIBO.

specifically lifetime and population sparseness distributions (Methods Lifetime and population sparseness) (*Willmore and Tolhurst, 2001*; *Lehky et al., 2005*; *Treves and Rolls, 1991*; *Pehlevan and Sompolinsky, 2014*). In *Figure 8D*, we compare the lifetime and population sparseness distributions of the true biological code with a RROS version of the same code, revealing biological neurons have significantly higher lifetime sparseness. In Appendix Necessary conditions for optimally sparse codes, we provide analytical arguments which suggest that tuning curves of optimally sparse non-negative codes with full-rank kernels will have selective tuning.

## Discussion

Elucidating inductive biases of the brain is fundamentally important for understanding natural intelligence (*Tenenbaum et al., 2011*; *Lake et al., 2017*; *Sinz et al., 2019*; *Zador, 2019*). These biases are coded into the brain by the dynamics of its neurons, the architecture of its networks, its representations and plasticity rules. Finding ways to extract the inductive biases from neuroscience datasets requires a deeper theoretical understanding of how all these factors shape the biases, and is an open problem. In this work, we attempted to take a step towards filling this gap by focusing on how the structure of static neural population codes shape inductive biases for learning of a linear readout neuron under a biologically plausible learning rule. If the readout neuron's output is correlated with behavior, and that correlation is known, then our theory could possibly be modified to predict what behavioral tasks can be learned faster.

Under the delta rule, the generalization performance of the readout is entirely dependent on the code's inner product kernel; the kernel is a determinant of inductive bias. In its finite dimensional form, the kernel is an example of a representational similarity matrix and is a commonly used tool to study neural representations (*Edelman, 1998*; *Kriegeskorte et al., 2008*; *Laakso and Cottrell, 2000*; *Kornblith et al., 2019*; *Cadieu et al., 2014*; *Pehlevan et al., 2018*). Our work elucidates a concrete link between this experimentally measurable mathematical object, and sample-efficient learning.

We derived an analytical expression for the generalization error as a function of sample-size under very general conditions, for an arbitrary stimulus distribution, arbitrary population code and an arbitrary target stimulus-response map. We used our findings in both theoretical and experimental analysis of primary visual cortex, and temporal codes in a delayed reach task. This generality of our theory is a particular strength.

Our analysis elucidated two principles that define the inductive bias. The first one is spectral bias: kernel eigenfunctions with large eigenvalues can be estimated using a smaller number of samples. The second principle is the code-task alignment: target functions with most of their power in top kernel eigenfunctions can be estimated efficiently and are compatible with the code. The cumulative power distribution, $C(k)$ (*Canatar et al., 2021*), provides a measure of this alignment. These findings define a notion of 'simplicity' bias in learning from examples, and provides a solution to the question of what stimulus-response maps are easier to learn. A similar simplicity bias has been also observed in training deep neural networks (*Rahaman et al., 2019*; *Xu et al., 2019*; *Kalimeris et al., 2019*). Due to a correspondence between gradient-descent trained neural networks in the infinite-width limit and kernel machines (*Jacot et al., 2018*), results on the spectral bias of kernel machines may shed light onto these findings (*Bordelon et al., 2020*; *Canatar et al., 2021*). Though our present analysis focused on learning a single layer weight vector with the biologically plausible delta-rule, future work could explore the learning curves of other learning rules for deep networks (*Bordelon and Pehlevan, 2022a*), such as feedback alignment (*Lillicrap et al., 2016*) or perturbation methods (*Jabri and Flower, 1992*). Such analysis could explore how inductive bias is also shaped by choice of learning rule, as well as the structure of the initial population code.

We applied our findings in both theoretical and experimental analysis of mouse primary visual cortex. We demonstrated a bias of neural populations towards low frequency orientation discrimination and low spatial frequency reconstruction tasks. The latter finding is consistent with the finding

that mouse visual cortex neurons are selective for low spatial frequency (*Niell and Stryker, 2008*; *Vreysen et al., 2012*). The toy model of the visual cortex as a mixture of simple and complex cells demonstrated how invariances, specifically the phase invariance of the complex cells, in the population code can facilitate learning some tasks involving phase invariant responses at the expense of performance on others. The role of invariances in learning with kernel methods and deep networks have recently been investigated in machine learning literature, showing that invariant representations can improve capacity (*Farrell et al., 2021*) and sample efficiency for invariant learning problems (*Mei et al., 2021*; *Li et al., 2019*; *Xiao and Pennington, 2022*).

A recent proposal considered the possibility that the brain acts as an overparameterized interpolator (*Hasson et al., 2020*). Suitable inductive biases are crucial to prevent overfitting and generalize well in such a regime (*Belkin et al., 2019*). Our theory can explain these inductive biases since, when the kernel is full-rank, which typically is the case when there are more neurons in the population than the number of learning examples, the delta rule without weight decay converges to an interpolator of the learning examples. Modern deep learning architectures also operate in an overparameterized regime, but generalize well (*Zhang et al., 2016*; *Belkin et al., 2019*), and an inductive bias towards simple functions has been proposed as an explanation (*Bordelon et al., 2020*; *Canatar et al., 2021*; *Kalimeris et al., 2019*; *Valle-Perez et al., 2018*). However, we also showed that interpolation can be harmful to prediction accuracy when the target function has some variance unexplained by the neural code or if the neural responses are significantly noisy, motivating use of explicit regularization.

Our work promotes sample efficiency as a general coding principle for neural populations, relating neural representations to the kinds of problems they are well suited to solve. These codes may be shaped through evolution or themselves be learned through prior experience (*Zador, 2019*). Prior related work in this area demonstrated the dependence of sample-efficient learning of a two-angle estimation task on the width of the individual neural tuning curves (*Meier et al., 2020*) and on additive function approximation properties of sparsely connected random networks (*Harris, 2019*).

A sample efficiency approach to population coding differs from the classical efficient coding theories (*Attneave, 1954*; *Barlow, 1961*; *Atick and Redlich, 1992*; *Srinivasan et al., 1982*; *van Hateren, 1992*; *Rao and Ballard, 1999*; *Olshausen and Field, 1997*; *Chalk et al., 2018*), which postulate that populations of neurons optimize information content of their code subject to metabolic constraints or noise. While these theories emphasize different aspects of the code's information content (such as reduced redundancy, predictive power, or sparsity), they do not address sample efficiency demands on learning. Further, recent studies demonstrated hallmarks of redundancy and correlation in population responses (*Chapin and Nicolelis, 1999*; *Bathellier et al., 2008*; *Pitkow and Meister, 2012*; *Gao and Ganguli, 2015*; *Abbasi-Asl et al., 2016*; *Gallego et al., 2018*; *Stringer et al., 2018b*), violating a generic prediction of efficient coding theories that responses of different neurons should be uncorrelated across input stimuli in high signal-to-noise regimes to reduce redundancy in the code and maximize information content (*Barlow, 1961*; *Atick and Redlich, 1992*; *Srinivasan et al., 1982*; *van Hateren, 1992*; *Haft and van Hemmen, 1998*; *Huang and Rao, 2011*). In our theory, the structured correlations of neural responses correspond to the decay in the spectrum of the kernel, and play a key role in biasing learned readouts towards simple functions.

In recent related studies, the asymptotic decay rate of the kernel's eigenspectrum was argued to be important for generalization (*Stringer et al., 2018b*) and robustness (*Nassar et al., 2020*). The spectral decay rate in the mouse V1 was found to be consistent with a high dimensional (power law) but smooth (differentiable) code, and smoothness was argued to be an enabler of generalization (*Stringer et al., 2018b*). While we also identify power law spectral decays, we show that sample-efficient learning requires more than smoothness conditions in the form of asymptotic decay rates on the kernel's spectrum. The interplay between the stimulus distribution, target response and the code gives rise to sample efficient learning. Because of spectral bias, the top eigenvalues govern the small sample size behavior. The tail of the spectrum becomes important at large sample sizes.

Though the kernel is degenerate with respect to rotations of the code in the neural activity space, we demonstrated that the true V1 code has much lower average activity than random codes with the same kernel, suggesting that evolution and learning may be selecting neural codes with low average spike rates which preserve sample-efficiency demands for downstream learning tasks. We predict that metabolic efficiency may be a determinant in the orientation and placement of the ubiquitously observed low-dimensional coding manifolds in neural activity space in other parts of the brain (*Gallego*

*et al., 2018*). The demand of metabolic efficiency is consistent with prior sparse coding theories (*Niven and Laughlin, 2008*; *Olshausen and Field, 1997*; *Simoncelli and Olshausen, 2001*; *Hromádka et al., 2008*), however, our theory emphasizes sample-efficient learning as the primary normative objective for the code. As a note of caution, while our analysis holds under the assumption that the neural code is deterministic, real neurons exhibit variability in their responses to repeated stimuli. Such noisy population codes do not generally achieve identical generalization performance under RROS transformations. For example, if each neuron is constrained to produce i.i.d. Poisson noise, then simple shifts of the baseline firing rate reduce the information content of the code. However, if the neural noise is Gaussian (even with stimulus dependent noise covariance), then the generalization error is conserved under RROS operations (Appendix Effect of noise on RROS symmetry). Further studies could focus on revealing the space of codes with equivalent inductive biases under realistic noise models.

Our work constitutes a first step towards understanding inductive biases in neuronal circuits. To achieve this, we focused on a linear, delta-rule readout of a static population code. More work is need to study other factors that affect inductive bias. Importantly, sensory neuron tuning curves can adapt during perceptual learning tasks (*Gilbert, 1994*; *Goltstein et al., 2021*; *Ghose et al., 2002*; *Schoups et al., 2001*) with the strength of adaptation dependent on brain area (*Yang and Maunsell, 2004*; *Adab et al., 2014*; *Op de Beeck et al., 2007*). In many experiments, these changes to tuning in sensory areas are small (*Schoups et al., 2001*; *Ghose et al., 2002*), satisfying the assumptions of our theory. For example monkeys trained on noisy visual motion detection exhibit changes in sensory-motor (LIP) but not sensory areas (MT), consistent with a model of readout from a static sensory population code (*Law and Gold, 2008*; *Shadlen and Newsome, 2001*). However, other perceptual learning tasks and other brain areas can exhibit significant changes in neural tuning (*Recanzone et al., 1993*; *Pleger et al., 2003*; *Furmanski et al., 2004*). This diversity of results motivates more general analysis of learning in multi-layer networks, where representations in each layer can adapt flexibly to task structure (*Shan and Sompolinsky, 2021*; *Mastrogiuseppe et al., 2022*; *Bordelon and Pehlevan, 2022b*; *Ahissar and Hochstein, 2004*). Alternatively, our current analysis of inductive bias can still be consistent with multi-layer learning if the network is sufficiently overparameterized and tuning curves change very little (*Jacot et al., 2018*; *Lee et al., 2018*; *Shan and Sompolinsky, 2021*). In this case, network training is equivalent to kernel learning with a kernel that depends on the learning rule and architecture (*Bordelon and Pehlevan, 2022a*). However, in the regime of neural network training where tuning curves change significantly, more sophisticated analytical tools are needed to predict generalization (*Flesch et al., 2021*; *Yang and Hu, 2021*; *Bordelon and Pehlevan, 2022b*). Although our work focused on linear readouts, arbitrary nonlinear readouts which generate convex learning objectives have been recently studied in the high dimensional limit, giving qualitatively similar learning curves which depend on kernel eigenvalues and task model alignment (*Loureiro et al., 2021b*; *Cui et al., 2022*) (see Appendix Typical case analysis of nonlinear readouts).

Our work focused on how signal correlations influence inductive bias (*Averbeck et al., 2006*; *Cohen and Kohn, 2011*). However, since real neurons do exhibit variability in their responses to identical stimuli, one should consider the effect of neural noise and noise correlations in learning. We provide a preliminary analysis of learning with neural noise in Appendix Impact of neural noise and unlearnable targets on learning, where we show that neural noise can lead to irreducible asymptotic error which depends on the geometry of the signal and noise correlations. Further, if the target function is not fully expressible as linear combinations of neural responses, overfitting peaks in the learning curves are possible, but can be mitigated with regularization implemented by a weight decay in the learning rule (see *Appendix 1—figure 1*). Future work could extend our analysis to study how signal and noise correlations interact to shape inductive bias and generalization performance in the case where the noise correlation matrices are non-isotropic, including the role of differential correlations (*Moreno-Bote et al., 2014*). Overall, future work could build on the present analysis to incorporate a greater degree of realism in a theory of inductive bias.

Finally, we discuss possible applications of our work to experimental neuroscience. Our theory has potential implications for experimental studies of task learning. First, in cases where the population selective to stimuli can be measured directly, an experimenter could design easy or difficult tasks for an animal to learn from few examples, under a hypothesis that the behavioral output is a linear function of the observed neurons. Second, in cases where it is unclear which neural population contributes to learning, one could utilize our theory to solve the inverse problem of inferring the relevant kernel from

observed learning curves on different tasks (*Wilson et al., 2015*). From these tasks, the experimenter could compare the inferred kernel to those of different recorded populations. For instance, one could compare the kernels from separate populations to the inferred kernel obtained from learning curves on certain visual learning tasks. This could provide new ways to test theories of perceptual learning (*Gilbert, 1994*). Lastly, extensions of our framework could quantify the role of neural variability on task learning and the limitation it imposes on accuracy and sample efficiency.

## Methods

### Generating example codes (Figure 1)

The two codes in *Figure 1* were constructed to produce two different kernels for $\theta \in S^1$:

$$K_1(\theta, \theta') = \exp(0.25 \cos(\theta - \theta')) \,, \; K_2(\theta, \theta') = \sum_{k=1}^{20} \cos(k(\theta - \theta')). \tag{4}$$

An infinite number of codes could generate either of these kernels. After diagonalizing the kernel into its eigenfunctions on a grid of 120 points, $\mathbf{K}_1 = \boldsymbol{\Psi}_1 \boldsymbol{\Lambda}_1 \boldsymbol{\Psi}_1^\top, \mathbf{K}_2 = \boldsymbol{\Psi}_2 \boldsymbol{\Lambda}_2 \boldsymbol{\Psi}_2^\top$, we used a random rotation matrix $\mathbf{Q} \in \mathbb{R}^{N \times N}$ (which satisfies $\mathbf{Q}\mathbf{Q}^\top = \mathbf{Q}^\top \mathbf{Q} = \mathbf{I}$) to generate a valid code

$$\mathbf{R}_1 = \mathbf{Q} \boldsymbol{\Lambda}_1^{1/2} \boldsymbol{\Psi}_1^\top \,, \; \mathbf{R}_2 = \mathbf{Q} \boldsymbol{\Lambda}_2^{1/2} \boldsymbol{\Psi}_2^\top. \tag{5}$$

This construction guarantees that $\mathbf{R}_1^\top \mathbf{R}_1 = \mathbf{K}_1$ and $\mathbf{R}_2^\top \mathbf{R}_2 = \mathbf{K}_2$. We plot the tuning curves for the first three neurons. The target function in the first experiment is $y = \cos(\theta) - 0.6 \cos(4\theta)$, while second experiment used $y = \cos(6\theta) - \cos(8\theta)$.

### Theory of generalization

Recent work has established analytic results that predict the average case generalization error for kernel regression

$$E_g = \left\langle E_g(\mathcal{D}) \right\rangle_{\mathcal{D}} = \left\langle \left[ f(\boldsymbol{\theta}, \mathcal{D}) - y(\boldsymbol{\theta}) \right]^2 \right\rangle_{\boldsymbol{\theta}, \mathcal{D}} \tag{6}$$

where $E_g(\mathcal{D}) = \left\langle [f(\boldsymbol{\theta}, \mathcal{D}) - y(\boldsymbol{\theta})]^2 \right\rangle_{\boldsymbol{\theta}}$ is the generalization error for a certain sample $\mathcal{D}$ of size $P$ and $f(\boldsymbol{\theta}, \mathcal{D}) = \mathbf{w} \cdot \mathbf{r}(\boldsymbol{\theta})$ is the kernel regression solution for $\mathcal{D}$ (Appendix Convergence of the delta-rule without weight decay) (*Bordelon et al., 2020*; *Canatar et al., 2021*). The typical or average case error $E_g$ is obtained by averaging over all possible datasets of size $P$. This average case generalization error is determined solely by the decomposition of the target function $y(\mathbf{x})$ along the eigenbasis of the kernel and the eigenspectrum of the kernel. This continuous diagonalization again takes the form (Appendix Singular value decomposition of continuous population responses) (*Rasmussen and Williams, 2005*)

$$\int p(\boldsymbol{\theta}) K(\boldsymbol{\theta}, \boldsymbol{\theta}') \psi_k(\boldsymbol{\theta}) d\boldsymbol{\theta} = \lambda_k \psi_k(\boldsymbol{\theta}'). \tag{7}$$

Our theory is also applicable to discrete stimuli if $p(\boldsymbol{\theta})$ is a Dirac measure as we describe in (Appendix Discrete stimulus spaces: finding eigenfunctions with matrix eigendecomposition). Since the eigenfunctions form a complete set of square integrable functions (*Rasmussen and Williams, 2005*), we expand both the target function $y(\boldsymbol{\theta})$ and the learned function $f(\boldsymbol{\theta}, \mathcal{D})$ in this basis $y(\boldsymbol{\theta}) = \sum_k v_k \psi_k(\boldsymbol{\theta}) \,, \; f(\boldsymbol{\theta}, \mathcal{D}) = \sum_k \hat{v}_k \psi_k(\boldsymbol{\theta})$, where $\hat{v}_k$ are understood to be functions of the dataset $\mathcal{D}$. The eigenfunctions are orthonormal $\int d\boldsymbol{\theta} p(\boldsymbol{\theta}) \psi_k(\boldsymbol{\theta}) \psi_\ell(\boldsymbol{\theta}) = \delta_{k,\ell}$, which implies that the generalization error for any set of coefficients $\hat{\mathbf{v}}$ is

$$E_g(\mathcal{D}) = \left\langle (y(\boldsymbol{\theta}) - f(\boldsymbol{\theta}, \mathcal{D}))^2 \right\rangle_{\boldsymbol{\theta}} = \sum_{k,\ell} (\hat{v}_k - v_k)(\hat{v}_\ell - v_\ell) \left\langle \psi_k(\boldsymbol{\theta}) \psi_\ell(\boldsymbol{\theta}) \right\rangle_{\boldsymbol{\theta}} = \|\hat{\mathbf{v}} - \mathbf{v}\|^2 \tag{8}$$

We now introduce the equivalent training error, or empirical loss, written directly in terms of eigenfunction coefficients $\hat{\mathbf{v}}$, which depends on the random dataset $\mathcal{D} = \{(\boldsymbol{\theta}^\mu, y^\mu)\}_{\mu=1}^P$

$$H(\hat{\mathbf{v}}, \mathcal{D}) = \sum_\mu [(\hat{\mathbf{v}} - \mathbf{v}) \cdot \boldsymbol{\psi}(\boldsymbol{\theta}^\mu)]^2 + \lambda \sum_k \frac{\hat{v}_k^2}{\lambda_k} \tag{9}$$

This loss function is minimized by delta rule updates with weight decay constant $\lambda$. It is straightforward to verify that the $H$-minimizing coefficients are $\hat{\mathbf{v}}^* = (\boldsymbol{\Psi}\boldsymbol{\Psi}^\top + \lambda\boldsymbol{\Lambda}^{-1})^{-1}\boldsymbol{\Psi}\boldsymbol{\Psi}^\top\mathbf{v}$, giving the learned function $f(\boldsymbol{\theta}, \mathcal{D}) = \hat{\mathbf{v}}^* \cdot \boldsymbol{\psi}(\boldsymbol{\theta}) = \mathbf{k}(\boldsymbol{\theta})^\top(\mathbf{K} + \lambda\mathbf{I})^{-1}\mathbf{y}$ where the vectors $\mathbf{k}$ and $\mathbf{y}$ have entries $[\mathbf{k}(\boldsymbol{\theta})]_\mu = K(\boldsymbol{\theta}, \boldsymbol{\theta}^\mu)$ and $[\mathbf{y}]_\mu = y(\boldsymbol{\theta}^\mu)$ for each training stimulus $\boldsymbol{\theta}^\mu \in \mathcal{D}$. The $P \times P$ kernel gram matrix $\mathbf{K}$ has entries $[\mathbf{K}]_{\mu\nu} = K(\boldsymbol{\theta}^\mu, \boldsymbol{\theta}^\nu)$. The $\lambda \to 0$ limit of the minimizer of $H$ coincides with kernel interpolation. This allows us to characterize generalization without reference to learned readout weights $\mathbf{w}$. The generalization error for this optimal function is

$$
\begin{aligned}
E_g(\mathcal{D}) &= \|\hat{\mathbf{v}}^* - \mathbf{v}\|^2 = \mathbf{v}^\top\boldsymbol{\Lambda}^{-1}\mathbf{G}(\mathcal{D})^2\boldsymbol{\Lambda}^{-1}\mathbf{v} \\
\mathbf{G}(\mathcal{D}) &= \left(\tfrac{1}{\lambda}\boldsymbol{\Psi}\boldsymbol{\Psi}^\top + \boldsymbol{\Lambda}^{-1}\right)^{-1}.
\end{aligned}
\tag{10}
$$

We note that the dependence on the randomly sampled dataset $\mathcal{D}$ only appears through the matrix $\mathbf{G}(\mathcal{D})$. Thus to compute the *typical* generalization error we need to average this matrix over realizations of datasets, *i.e.*$\langle\mathbf{G}(\mathcal{D})\rangle_\mathcal{D}$. There are multiple strategies to perform such an average and we will study one here based on a partial differential equation which was introduced in *Sollich, 1998*; *Sollich, 2002* and studied further in *Bordelon et al., 2020*. We describe in detail one method for performing such an average in Appendix Computation of learning curves. After this computation, we find that the generalization error can be approximated at large $P$ as

$$
E_g = \tfrac{\kappa^2}{1-\gamma}\sum_k \tfrac{v_k^2}{(\lambda_k P + \kappa)^2} \,, \quad \kappa = \lambda + \kappa\sum_k \tfrac{\lambda_k}{\lambda_k P + \kappa},
\tag{11}
$$

where $\gamma = P\sum_k \tfrac{\lambda_k^2}{(\lambda_k p + \kappa)^2}$, giving the desired result. We note that (11) defines the function $\kappa$ implicitly in terms of the sample size $P$. Taking $\lambda \to 0$ gives the generalization error of the minimum norm interpolant, which desribes the generalization error of the solution. This result was recently reproduced using the replica method from statistical mechanics in an asymptotic limit where the number of neurons and samples are large (*Bordelon et al., 2020*; *Canatar et al., 2021*). Other recent works have verified our theoretical expressions on a variety of kernels and datasets (*Loureiro et al., 2021b*; *Simon et al., 2021*).

Additional intuition for the spectral bias phenomenon can be gained from the expected learned function $\langle f(\boldsymbol{\theta}, \mathcal{D})\rangle_\mathcal{D} = \sum_k \tfrac{\lambda_k P}{\lambda_k P + \kappa}v_k\psi_k(\boldsymbol{\theta})$, which is the average readout prediction over possible datasets $\mathcal{D}$. The function $\kappa(P)$ is defined implicitly as $\kappa = \lambda + \kappa\sum_k \tfrac{\lambda_k}{\lambda_k P + \kappa}$ and decreases with $P$ from $\kappa(0) = \lambda + \sum_k \lambda_k$ to its asymptotic value $\lim_{P\to\infty}\kappa(P) = \lambda$. The coefficient for the $k$-th eigenfunction $\tfrac{\lambda_k P}{\lambda_k P + \kappa}v_k$ approaches the true coefficient $v_k$ as $P \to \infty$. The $k$-th eigenfunction $\psi_k$ is effectively learned when $P \gg \tfrac{\kappa}{\lambda_k}$. For large eigenvalues $\lambda_k$, fewer samples $P$ are needed to satisfy this condition, while small eigenvalue modes will require more samples.

## RNN experiment

For the simulations in *Figure 7*, we integrated a rate-based recurrent network model with $N = 6000$ neurons, time constant $\tau = 0.05$ and gain $g = 1.5$. Each of the $P = 80$ randomly chosen angles $\gamma^\mu$ generates a trajectory over $T = 100$ equally spaced points in $t \in [0, 3]$. The two dimensional input sequence is simply $\boldsymbol{\theta}(t) = H(t)H(1 - t)\left[\cos(\gamma^\mu), \sin(\gamma^\mu)\right]^\top \in \mathbb{R}^2$. Target function for a delay $d$ is $\mathbf{y}(\theta^\mu, t) = H(1.5 + d - t)H(t - d - 1)[\cos(\gamma^\mu), \sin(\gamma^\mu)]^\top$ which is nonzero for times $t \in [1 + d, 1.5 + d]$. In each simulation, the activity in the network is initialized to $\mathbf{u}(0) = \mathbf{0}$. The kernel gram matrix $\mathbf{K} \in \mathbb{R}^{PT \times PT}$ is computed by taking inner products of the time varying code at for different inputs $\gamma^\mu$ and at different times. Learning curves represent the generalization error obtained by randomly sampling $\mathcal{P}$ time points from the $PT$ total time points generated in the simulation process and training readout weights $\mathbf{w}$ to convergence with gradient descent.

## Data analysis
### Data source and processing

Mouse V1 neuron responses to orientation gratings were obtained from a publicly available dataset (*Stringer et al., 2021*; *Pachitariu et al., 2019*). Two-photon calcium microscopy fluorescence traces were deconvolved into spike trains and spikes were counted for each stimulus, as described in *Stringer et al., 2021*. The presented grating angles were distributed uniformly over $[0, 2\pi]$ radians. Data pre-processing, which included z-scoring against the mean and standard deviation of null stimulus

responses, utilized the provided code for this experiment, which also publicly available at https://github.com/MouseLand/stringer-et-al-2019 (**Stringer, 2019**). This preprocessing technique was used in all Figures in the paper. To reduce corruption of the estimated kernel from neural noise (trial-to-trial variability), we first trial average responses, binning the grating stimuli oriented at different angles $\theta$ into a collection of 100 bins over the interval from $[0, 2\pi]$ and averaging over all of the available responses from each bin. Since grating angles were sampled uniformly, there is a roughly even distribution of about 45 responses in each bin. After trial averaging, SVD was performed on the response matrix $\mathbf{R}$, generating the eigenspectrum and kernel eigenfunctions as illustrated in **Figure 3**. **Figures 2, 3 and 8**, all used this data anytime responses to grating stimuli were mentioned.

In **Figures 3D, 4 and 8C**, the responses of mouse V1 neurons to ImageNet images (**Deng et al., 2009**) were obtained from a different publicly available dataset (**Stringer et al., 2018a**). The images were taken from 15 different classes from the Imagenet dataset with ethological relevance to mice (birds, cats, flowers, hamsters, holes, insects, mice, mushrooms, nests, pellets, snakes, wildcats, other animals, other natural, other man made). In the experiment in **Figure 3D** we use all images from the mice and birds category for which responses were recorded. The preprocessing code and image category information were obtained from the publicly available code base at https://github.com/MouseLand/stringer-pachitariu-et-al-2018b (**Stringer, 2018c**). Again, spike counts were obtained from deconvolved and z-scored calcium fluorescence traces. In the reconstruction experiment shown in **Figure 4** we use the entire set of images for which neural responses were recorded.

## Generating RROS codes

In **Figure 8**, the randomly rotated codes are generated by sampling a matrix $\mathbf{Q}$ from the Haar measure on the set of $N$-by-$N$ orthogonal matrices (**Anderson et al., 1987**), and chosing a $\delta$ by solving the following optimization problem:

$$\min_{\boldsymbol{\delta} \in \mathbb{R}^N} \sum_{i=1}^N \sum_{\mu=1}^P s_i^\mu \ , \ \text{s.t.} \ \mathbf{s}^\mu = \mathbf{Qr}(\theta^\mu) + \boldsymbol{\delta} \ , \ s_i^\mu \geq 0 \ , \ i \in [N] \ , \ \mu \in [P], \tag{12}$$

which minimizes the total spike count subject to the kernel and nonnegativity of firing rates. The solution to this problem is given by $\delta_i^* = -\min_{\mu=1,\dots,P}[\mathbf{Qr}(\theta^\mu)]_i$.

## Comparing sparsity of population codes

To explore the metabolic cost among the set of codes with the same inductive biases, we estimate the distribution of average spike counts of codes with the same inner product kernel as the biological code. These codes are generated in the form $\mathbf{s}^\mu = \mathbf{Qr}^\mu + \boldsymbol{\delta}$ where $\boldsymbol{\delta}$ solves the optimization problem

$$\min_{\boldsymbol{\delta} \in \mathbb{R}^N} \sum_{i,\mu} s_i^\mu \ , \ s.t. \ \mathbf{s}^\mu = \mathbf{Qr}^\mu + \boldsymbol{\delta} \ , \ s_i^\mu \geq 0 \tag{13}$$

To quantify the distribution of such codes, we randomly sample $\mathbf{Q}$ from the invariant (Haar) measure for $N \times N$ orthogonal matrices and compute the optimal $\boldsymbol{\delta}$ as described above. This generates the aqua colored distribution in **Figure 8B and C**.

We also attempt to characterize the most efficient code with the same inner product kernel

$$\min_{\mathbf{Q}, \boldsymbol{\delta}} \sum_{i,\mu} s_i^\mu \ , \ s.t. \ \mathbf{s}^\mu = \mathbf{Qr}^\mu + \boldsymbol{\delta} \ , \ s_i^\mu \geq 0. \tag{14}$$

Since this optimization problem is non-convex in $\mathbf{Q}$, there is no theoretical guarantee that minima are unique. Nonetheless, we attempt to optimize the code by starting $\mathbf{Q}$ at the identity matrix and conduct gradient descent over orthogonal matrices (**Plumbley, 2004**). Such updates take the form

$$\mathbf{Q}_{t+1} = \exp(-\eta \nabla \mathcal{L}) \mathbf{Q}_t \ , \ \nabla \mathcal{L} = \frac{\partial \mathcal{L}}{\partial \mathbf{Q}} \mathbf{Q}^\top - \mathbf{Q} \frac{\partial \mathcal{L}}{\partial \mathbf{Q}}^\top \tag{15}$$

where $\exp(\cdot)$ is the matrix exponential. To make the loss function differentiable, we incorporate the non-negativity constraint with a soft-minimum:

$$\begin{aligned} \mathcal{L} \ &= \sum_{i\mu} \left( \mathbf{q}_i^\top \mathbf{r}^\mu - \text{softmin}_\nu(\mathbf{q}_i^\top \mathbf{r}^\nu, \beta) \right) \\ \text{softmin}(a^1, a^2, &..., a^P; \beta) = \frac{1}{Z} \sum_{\mu=1}^P a^\mu \exp(-\beta a^\mu), \end{aligned} \tag{16}$$

where $Z = \sum_{\nu} \exp(-\beta a^{\nu})$ is a normalizing constant and $\mathbf{Q} = [\mathbf{q}_1, ... \mathbf{q}_N]$. In the $\beta \to \infty$ limit, this cost function converges to the exact optimization problem with non-negativity constraint. Finite $\beta$, however, allows learning with gradient descent. Gradients are computed with automatic differentiation in JAX (**Bradbury et al., 2018**). This optimization routine is run until convergence and the optimal value is plotted as dashed red lines labeled 'opt'. in **Figure 8**.

We show that our result is robust to different pre-processing techniques and to imposing bounds on neural firing rates in the **Figure 8—figure supplement 1**. To demonstrate that our result is not an artifact of z-scoring the deconvolved signals against the spontaneous baseline activity level, we also conduct the random rotation experiment on the raw deconvolved signals. In addition, we show that imposing realistic constraints on the upper bound of the each neuron's responses does not change our findings. We used a subset of $N = 100$ neurons and computed random rotations. However, we only accepted a code as valid if it's maximum value was less than some upper bound $u_b$. Subsets of $N = 100$ neurons in the biological code achieve maxima in the range between 3.2 and 4.7. We performed this experiment for $u_b \in \{3, 4, 5\}$ so that the artificial codes would have maxima that lie in the same range as the biological code.

## Lifetime and population sparseness

We compute two more refined measures of sparseness in a population code. For each neuron $i$ we compute the lifetime sparseness $LS_i$ (also known as selectivity) and for each stimulus $\boldsymbol{\theta}$ we compute the population sparseness $PS_{\boldsymbol{\theta}}$ which are defined as the following two ratios (**Willmore and Tolhurst, 2001**; **Lehky et al., 2005**; **Treves and Rolls, 1991**; **Pehlevan and Sompolinsky, 2014**)

$$LS_i = \frac{1}{1 - \frac{1}{P}} \frac{\mathrm{Var}_{\boldsymbol{\theta}} \, r_i(\boldsymbol{\theta})}{\langle r_i(\boldsymbol{\theta})^2 \rangle_{\boldsymbol{\theta}}} \, , \quad PS_{\boldsymbol{\theta}} = \frac{1}{1 - \frac{1}{N}} \frac{\mathrm{Var}_i \, r_i(\boldsymbol{\theta})}{\langle r_i(\boldsymbol{\theta})^2 \rangle_i} \tag{17}$$

The normalization factors ensure that these quantities lie in the interval between $(0, 1)$. Intuitively, lifetime sparseness quantifies the variability of each neuron's responses over the full set of stimuli, whereas population sparseness quantifies the variability of responses in the code for a given stimulus $\boldsymbol{\theta}$.

## Fitting a Gabor model to mouse V1 kernel

Under the assumption of translation symmetry in the kernel $K(\theta, \theta')$, we averaged the elements of the over rows of the empirical mouse V1 kernel (**Pachitariu et al., 2019**)

$$K(\Delta) = \frac{1}{P} \sum_{\mu=1}^{P} K(\theta^{\mu}, \theta^{\mu} + \Delta) \tag{18}$$

where angular addition is taken mod $\pi$. This generates the black dots in **Figure 5B**. We aimed to fit a threshold-power law nonlinearity of the form $g_{q,a}(z) = \max\{0, z - a\}^q$ to the kernel. Based on the Gabor model discussed above, we parameterized tuning curves as

$$r_{\sigma^2, q, a}(\theta, \theta_i) = g_{q,a}\left( \frac{\cosh(\sigma^{-2} \cos(\theta - \theta_i))}{\cosh(\sigma^{-2})} \right), \tag{19}$$

where $\theta_i$ is the preferred angle of the $i$-th neuron's tuning curve. Rather than attempting to perform a fit of $\sigma^2, a, q, \{\theta_i\}_{i=1}^{N}$ of this form to the responses of each of the $\sim$ 20-k neurons, we instead simply attempt to fit to the population kernel by optimizing over $(\sigma^2, a, q)$ under the assumption of uniform tiling of $\theta_i$. However, we noticed that two of these variables $\sigma^2, a$ are constrained by the sparsity level of the code. If each neuron, on average, fires for only a fraction $f$ of the uniformly sampled angles $\theta$, then the following relationship holds between $\sigma^2$ and

$$a = \frac{\cosh\left(\sigma^{-2} \cos\left(\frac{\pi}{2} f\right)\right)}{\cosh(\sigma^{-2})}. \tag{20}$$

Calculation of the coding level $f$ for the recorded responses allowed us to infer $a$ from $\sigma^2$ during optimization. This reduced the free parameter set to $(\sigma^2, q)$. We then solve the following optimization problem

$$\min_{\sigma^2, q} \left\langle \left( \hat{K}_{\sigma^2, q}(\theta) - K(\theta) \right)^2 \right\rangle_{\theta} \quad \hat{K}_{\sigma^2, q}(\theta) = \left\langle r_{\sigma^2, q}(\theta, \theta') r_{\sigma^2, q}(0, \theta') \right\rangle_{\theta'}, \tag{21}$$

where integration over $\theta_i$ is performed numerically. Using the Scipy Trust-Region constrained optimizer, we found $(q, \sigma^{-2}, a) = (1.7, 5.0, 0.2)$ which we use as the fit parameters in *Figure 5*.

## Lead contact

Requests for information should be directed to the lead contact, Cengiz Pehlevan (cpehlevan@seas.harvard.edu).

## Data and code availability

Mouse V1 neuron responses to orientation gratings and preprocessing code were obtained from a publicly available dataset: https://github.com/MouseLand/stringer-et-al-2019, (*Stringer et al., 2021*; *Pachitariu et al., 2019*).

Responses to ImageNet images and preprocessing code were obtained from another publicly available dataset, https://github.com/MouseLand/stringer-pachitariu-et-al-2018b (*Stringer et al., 2018a*).

The code generated by the authors for this paper is also available https://github.com/Pehlevan-Group/sample_efficient_pop_codes (*Pehlevan-Group, 2022*).

## Acknowledgements

We thank Jacob Zavatone-Veth and Abdulkadir Canatar for useful comments and discussions about this manuscript. BB acknowledges the support of the NSF-Simons Center for Mathematical and Statistical Analysis of Biology at Harvard (award #1764269) and the Harvard Q-Bio Initiative. CP and BB were also supported by NSF grant DMS-2134157.

## Additional information

### Funding

| Funder | Grant reference number | Author |
| --- | --- | --- |
| National Science Foundation | DMS-2134157 | Blake Bordelon Cengiz Pehlevan |

The funders had no role in study design, data collection and interpretation, or the decision to submit the work for publication.

### Author contributions

Blake Bordelon, Conceptualization, Software, Formal analysis, Investigation, Visualization, Methodology, Writing - original draft, Writing - review and editing; Cengiz Pehlevan, Conceptualization, Supervision, Funding acquisition, Investigation, Methodology, Writing - original draft, Project administration, Writing - review and editing

### Author ORCIDs

Blake Bordelon http://orcid.org/0000-0003-0455-9445
Cengiz Pehlevan http://orcid.org/0000-0001-9767-6063

### Decision letter and Author response

Decision letter https://doi.org/10.7554/eLife.78606.sa1
Author response https://doi.org/10.7554/eLife.78606.sa2

## Additional files

### Supplementary files

• MDAR checklist

### Data availability

Mouse V1 neuron responses to orientation gratings and preprocessing code were obtained from a publicly available dataset: https://github.com/MouseLand/stringer-et-al-2019. Responses to

ImageNet images and preprocessing code were obtained from another publicly available dataset, https://github.com/MouseLand/stringer-pachitariu-et-al-2018b. The code generated by the authors for this paper is also available https://github.com/Pehlevan-Group/sample_efficient_pop_codes, (copy archived at swh:1:rev:6cd4f0fe7043ae214dd682a9dc035a497ffa2d61).

The following previously published datasets were used:

| Author(s) | Year | Dataset title | Dataset URL | Database and Identifier |
|---|---|---|---|---|
| Carsen S, Marius P, Nicholas S, Matteo C, Kenneth DH | 2018 | Recordings of ten thousand neurons in visual cortex in response to 2,800 natural images | https://doi.org/10.25378/janelia.6845348.v4 | Figshare, 10.25378/janelia.6845348.v4 |
| Marius P, Michalis M, Carsen S | 2019 | Recordings of ~20,000 neurons from V1 in response to oriented stimuli | http://doi.org/10.25378/janelia.8279387.v3 | Figshare, 10.25378/janelia.8279387.v3 |

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

# Appendix 1

## Singular value decomposition of continuous population responses

SVD of population responses is usually evaluated with respect to a discrete and finite set of stimuli. In the main paper, we implicitly assumed that a generalization of SVD to a continuum of stimuli. In this section we provide an explicit construction of this generalized SVD using techniques from functional analysis. Our construction is an example of the quasimatrix SVD defined in *Townsend and Trefethen, 2015* and justifies our use of SVD in the main text.

For our construction, we note that Mercer's theorem guarantees the existence of an eigendecomposition of any inner product kernel $K(\boldsymbol{\theta}, \boldsymbol{\theta}')$ in terms of a complete orthonormal set of functions $\{\psi_k\}_{k=1}^{\infty}$ (*Rasmussen and Williams, 2005*). In particular, there exist a non-negative (but possibly zero) summable eigenvalues $\{\lambda_k\}_{k=1}^{\infty}$ and a corresponding set of orthonormal eigenfunctions such that

$$K(\boldsymbol{\theta}, \boldsymbol{\theta}') = \sum_{k=1}^{\infty} \lambda_k \psi_k(\boldsymbol{\theta}) \psi_k(\boldsymbol{\theta}'). \tag{22}$$

For a stimulus distribution $p(\boldsymbol{\theta})$, the set of functions $\{\psi_k\}_{k=1}^{\infty}$ are orthonormal and form a complete basis for square integrable functions $L_2$ which means

$$\begin{aligned} \langle \psi_k(\boldsymbol{\theta}) \psi_\ell(\boldsymbol{\theta}) \rangle_{\boldsymbol{\theta}} &= \int p(\boldsymbol{\theta}) \psi_k(\boldsymbol{\theta}) \psi_\ell(\boldsymbol{\theta}) d\boldsymbol{\theta} = \delta_{k\ell}, \\ f(\boldsymbol{\theta}) &= \sum_k \langle f(\boldsymbol{\theta}') \psi_k(\boldsymbol{\theta}') \rangle_{\boldsymbol{\theta}'} \psi_k(\boldsymbol{\theta}), \ \forall f \in L_2. \end{aligned} \tag{23}$$

Next, we use this basis to construct the SVD. Each of the tuning curves $r_i \in L_2$ (assumed to be square integrable) can be expressed in this basis with the top $N$ of the functions in the set

$$r_i(\boldsymbol{\theta}) = \sum_{k=1}^{N} A_{ik} \psi_k(\boldsymbol{\theta}), \tag{24}$$

where we introduced a matrix $\mathbf{A} \in \mathbb{R}^{N \times N}$ of expansion coefficients. Note that $\mathrm{rank}(\mathbf{A}) \leq N$. We compute the singular value decomposition of the finite matrix $\mathbf{A}$

$$\mathbf{A} = \sqrt{N} \sum_{k=1}^{\mathrm{rank}(\mathbf{A})} \sqrt{\lambda_k} \mathbf{u}_k \mathbf{v}_k^\top. \tag{25}$$

We note that the signal correlation matrix for this population code can be computed in closed form

$$\boldsymbol{\Sigma}_s = \frac{1}{N} \mathbf{A} \left\langle \boldsymbol{\psi}(\boldsymbol{\theta}) \boldsymbol{\psi}(\boldsymbol{\theta})^\top \right\rangle_{\boldsymbol{\theta}} \mathbf{A}^\top = \frac{1}{N} \mathbf{A} \mathbf{A}^\top = \sum_{k=1}^{\mathrm{rank}(\mathbf{A})} \lambda_k \mathbf{u}_k \mathbf{u}_k^\top, \tag{26}$$

due to the orthonormality of $\{\psi_k\}$. Thus the principal axes $\mathbf{u}_k$ of the neural correlations are the left singular vectors of $\mathbf{A}$. We may similarly express the inner product kernel in terms of the eigenfunctions

$$K(\boldsymbol{\theta}, \boldsymbol{\theta}') = \frac{1}{N} \mathbf{r}(\boldsymbol{\theta}) \cdot \mathbf{r}(\boldsymbol{\theta}') = \frac{1}{N} \boldsymbol{\psi}(\boldsymbol{\theta})^\top \mathbf{A}^\top \mathbf{A} \boldsymbol{\psi}(\boldsymbol{\theta}'). \tag{27}$$

The kernel eigenvalue problem demands (*Rasmussen and Williams, 2005*)

$$\begin{aligned} &\int p(\boldsymbol{\theta}) K(\boldsymbol{\theta}, \boldsymbol{\theta}') \boldsymbol{\psi}(\boldsymbol{\theta}) d\boldsymbol{\theta} = \frac{1}{N} \mathbf{A}^\top \mathbf{A} \boldsymbol{\psi}(\boldsymbol{\theta}') = \boldsymbol{\Lambda} \boldsymbol{\psi}(\boldsymbol{\theta}') \implies \frac{1}{N} \mathbf{A}^\top \mathbf{A} = \boldsymbol{\Lambda} \\ &\implies \sum_{k=1}^{\mathrm{rank}(\mathbf{A})} \lambda_k \mathbf{v}_k \mathbf{v}_k^\top = \sum_{k=1}^{\mathrm{rank}(\mathbf{A})} \lambda_k \mathbf{e}_k \mathbf{e}_k^\top. \end{aligned} \tag{28}$$

The $\mathbf{v}_k$ vectors must be identical to $\pm \mathbf{e}_k$, the Cartesian unit vectors, if the eigenvalues are non-degenerate. From this exercise, we find that the SVD for $\mathbf{A}$ has the form $\mathbf{A} = \sqrt{N} \sum_{k=1}^{\mathrm{rank}(\mathbf{A})} \sqrt{\lambda_k} \mathbf{u}_k \mathbf{e}_k^\top$. With this choice, the population code admits a singular value decomposition

$$\mathbf{r}(\boldsymbol{\theta}) = \mathbf{A} \boldsymbol{\psi}(\boldsymbol{\theta}) = \sqrt{N} \sum_{k=1}^{\mathrm{rank}(\mathbf{A})} \sqrt{\lambda_k} \mathbf{u}_k \psi_k(\boldsymbol{\theta}). \tag{29}$$

This singular value decomposition demonstrates the connection between neural manifold structure (principal axes $\mathbf{u}_k$) and function approximation (kernel eigenfunctions $\psi_k$). This singular value decomposition can be verified by computing the inner product kernel and the correlation matrix, utilizing the orthonormality of $\{\mathbf{u}_k\}$ and $\{\psi_k\}$. This exercise has important consequences for the space of learnable functions, which is at most $\mathrm{rank}(\mathbf{A})$ dimensional since linear readouts lie in $\mathrm{span}\{r_i(\boldsymbol{\theta})\}_{i=1}^{N}$.

## Discrete stimulus spaces: finding eigenfunctions with matrix eigendecomposition

In our discussion so far, our notation suggested that $\theta$ take a continuum of values. Here we want to point that our theory still applies if $\theta$ take a discrete set of values. In this case, we can think of a Dirac measure $p(\theta) = \sum_{i=1}^{\tilde{P}} p_i \delta(\theta - \theta^i)$, where $i$ indexes all the $\tilde{P}$ values $\theta$ can take. With this choice

$$\int p(\theta) K(\theta, \theta') \psi_k(\theta) d\theta = \sum_{i=1}^{\tilde{P}} p_i K(\theta^i, \theta') \psi_k(\theta^i) = \lambda_k \psi_k(\theta'). \tag{30}$$

Demanding this equality for $\theta' = \theta^i$, $i = 1, ..., \tilde{P}$ generates a matrix eigenvalue problem

$$\mathbf{KB\Psi} = \mathbf{\Psi\Lambda}, \tag{31}$$

where $\mathbf{B}_{ij} = \delta_{ij} p_i$. The eigenfunctions over the stimuli are identified as the columns of $\mathbf{\Psi}$ while the eigenvalues are the diagonal elements of $\mathbf{\Lambda}_{k\ell} = \lambda_k \delta_{k\ell}$.

### Experimental considerations

In an experimental setting, a finite number of stimuli are presented and the SVD is calculated over this finite set regardless of the support of $p(\theta)$. This raises the question of the interpretation of this SVD and its relation to the inductive bias theory we presented. Here we provide two interpretations.

In the first interpretation, we think of the empirical SVD as providing an estimate of the SVD over the full distribution $p(\theta)$. To formalize this notion, we can introduce a Monte-Carlo estimate of the integral eigenvalue problem

$$\int p(\theta) K(\theta, \theta') \psi_k(\theta) d\theta \approx \frac{1}{\tilde{P}} \sum_{\mu=1}^{\tilde{P}} K(\theta^\mu, \theta') \psi_k(\theta^\mu) = \lambda_k \psi_k(\theta'). \tag{32}$$

For this interpretation to work, the experimenter must sample the stimuli from $p(\theta)$, which could be the natural stimulus distribution. Measuring responses to a larger number of stimuli gives a more accurate approximation of the integral above, which will provide a better estimate of generalization performance on the true distribution $p(\theta)$.

In the second interpretation, we construct an empirical measure on $\tilde{P}$ experimental stimulus values $\hat{p}(\theta) = \frac{1}{\tilde{P}} \sum_{\mu=1}^{\tilde{P}} \delta(\theta - \theta^\mu)$, and consider learning and generalization over this distribution. This allows the application of our theory to an experimental setting where $\hat{p}(\theta)$ is designed by an experimenter. For example, the experimenter could procure a complicated set of $\tilde{P}$ videos, to which an associated function $y(\theta)$ must be learned. After showing these videos to the animal and measuring neural responses, the experimenter could compute, with our theory, generalization error for a uniform distribution over this full set of $\tilde{P}$ videos. Our theory would predict generalization over this distribution after providing supervisory feedback for only a strict subset of $P < \tilde{P}$ videos. Under this interpretation, the relationship between the integral eigenvalue problem and matrix eigenvalue problem is exact rather than approximate

$$\int \hat{p}(\theta) K(\theta, \theta') \psi_k(\theta) d\theta = \frac{1}{\tilde{P}} \sum_{\mu=1}^{\tilde{P}} K(\theta^\mu, \theta') \psi_k(\theta^\mu) = \lambda_k \psi_k(\theta'). \tag{33}$$

Demanding either of (32) or (33) equalities for $\theta' = \theta^\nu$, $\nu = 1, ..., P$ generates a matrix eigenvalue problem

$$\mathbf{K\Psi} = P\mathbf{\Psi\Lambda}. \tag{34}$$

The eigenfunctions restricted to $\{\theta^\mu\}$ are identified as the columns of $\mathbf{\Psi}$ while the eigenvalues are the diagonal elements of $\mathbf{\Lambda}_{k\ell} = \lambda_k \delta_{k\ell}$. For the case where $N$ and $P$ are finite, the spectrum obtained through eigendecomposition of the kernel $\mathbf{K}$ is the same as would be obtained through the finite $N$ signal correlation matrix $\mathbf{\Sigma}_s$, since they are inner and outer products of trial averaged population response matrices $\mathbf{R}$.

## Translation invariant kernels

For the special case where the data distribution $p(\boldsymbol{\theta}) = \frac{1}{V}$ is uniform over volume $V$ and the kernel is translation invariant $K(\boldsymbol{\theta}, \boldsymbol{\theta}') = \kappa(\boldsymbol{\theta} - \boldsymbol{\theta}')$, the kernel can be diagonalized in the basis of plane waves

$$\int p(\boldsymbol{\theta}) K(\boldsymbol{\theta}, \boldsymbol{\theta}') \psi_{\mathbf{k}}(\boldsymbol{\theta}) d\boldsymbol{\theta} = \frac{1}{V} \int \kappa(\boldsymbol{\theta} - \boldsymbol{\theta}') e^{i\mathbf{k}\cdot\boldsymbol{\theta}} d\boldsymbol{\theta} = \frac{1}{V} \hat{\kappa}(\mathbf{k}) e^{i\mathbf{k}\cdot\boldsymbol{\theta}'} \tag{35}$$

The eigenvalues are the Fourier components of the Kernel $\lambda_{\mathbf{k}} = \frac{1}{V}\hat{\kappa}(\mathbf{k}) = \frac{1}{V}\int d\boldsymbol{\theta} e^{i\mathbf{k}\cdot\boldsymbol{\theta}} \kappa(\boldsymbol{\theta})$ while the eigenfunctions are plane waves $\psi_{\mathbf{k}}(\boldsymbol{\theta}) = e^{i\mathbf{k}\cdot\boldsymbol{\theta}}$. The set of admissible momenta $\mathcal{S}_{\mathbf{k}} = \{\mathbf{k}_0, \pm\mathbf{k}_1, \pm\mathbf{k}_2, ...\}$ are determined by the boundary conditions. The diagonalized representation of the kernel is therefore

$$K(\boldsymbol{\theta}, \boldsymbol{\theta}') = \sum_{\mathbf{k}\in\mathcal{S}_{\mathbf{k}}} \lambda_{\mathbf{k}} e^{i\mathbf{k}\cdot(\boldsymbol{\theta}-\boldsymbol{\theta}')} \tag{36}$$

For example, if the space is the torus $\mathbb{T}^n = S^1 \times S^1 \times ... \times S^1$, then the space of admissible momenta are the points on the integer lattice $\mathcal{S}_{\mathbf{k}} = \mathbb{Z}^n = \{\mathbf{k} \in \mathbb{R}^n | k_i \in \mathbb{Z} \; \forall i = 1, ..., n\}$. Reality and symmetry of the kernel demand that $\text{Im}(\lambda_{\mathbf{k}}) = 0$ and $\lambda_{-\mathbf{k}} = \lambda_{\mathbf{k}} \geq 0$. Most of the models in this paper consider $\theta \sim \text{Unif}\left(S^1\right)$, where the kernel has the following Fourier/Mercer decomposition

$$\begin{aligned} K(\theta - \theta') &= \sum_{k=-\infty}^{\infty} \lambda_k e^{ik(\theta-\theta')} = 2\sum_{k=0}^{\infty} \lambda_k \cos(k(\theta - \theta')) \\ &= \sum_{k=0}^{\infty} \lambda_k \left[ \sqrt{2}\cos(k\theta)\sqrt{2}\cos(k\theta') + \sqrt{2}\sin(k\theta)\sqrt{2}\sin(k\theta') \right] \end{aligned} \tag{37}$$

where we invoked the simple trigonometric identity $\cos(a - b) = \cos(a)\cos(b) + \sin(a)\sin(b)$. By recognizing that $\{\sqrt{2}\cos(k\theta), \sqrt{2}\sin(k\theta)\}_{k=0}^{\infty}$ form a complete orthonormal set of functions with respect to $\text{Unif}\left(S^1\right)$, we have identified this as the collection of kernel eigenfunctions.

## Invariant kernels possess invariant eigenfunctions

Suppose the kernel $K(\boldsymbol{\theta}, \boldsymbol{\theta}')$ is invariant to some set of transformations $t \in \mathcal{T}$, by which we mean that

$$K(t\boldsymbol{\theta}, \boldsymbol{\theta}') = K(\boldsymbol{\theta}, t\boldsymbol{\theta}') = K(\boldsymbol{\theta}, \boldsymbol{\theta}'), \; \forall t \in \mathcal{T} \tag{38}$$

We will now show that any eigenfunction of such a kernel with nonzero eigenvalue must be an invariant function. Let $\psi_k(\boldsymbol{\theta})$ be an eigenfunction with eigenvalue $\lambda_k > 0$, then

$$\psi_k(t\boldsymbol{\theta}) = \frac{1}{\lambda_k}\int p(\boldsymbol{\theta}')K(\boldsymbol{\theta}', t\boldsymbol{\theta})d\boldsymbol{\theta}' = \frac{1}{\lambda_k}\int p(\boldsymbol{\theta}')K(\boldsymbol{\theta}', \boldsymbol{\theta})d\boldsymbol{\theta}' = \psi_k(\boldsymbol{\theta}) \tag{39}$$

This establishes that all functions which depend on $\mathcal{T}$ transformations must necessarily lie in the null-space of $K$.

## Theory of generalization

### Convergence of the delta-rule without weight decay

In this section, we discuss the delta-rule convergence when weight decay parameter is set to $\lambda = 0$. The next section considers the simpler case where $\lambda > 0$. Gradient descent training of readout weights $\mathbf{w}$ on a finite sample of size $P$ converges to the kernel regression solution (**Bartlett et al., 2020**; **Hastie et al., 2020**). Let $\mathcal{D} = \{\boldsymbol{\theta}^\mu, y^\mu\}_{\mu=1}^{P}$ be the dataset with samples $\boldsymbol{\theta}^\mu$ and target values $y^\mu$. We introduce a shorthand $\mathbf{r}^\mu = \mathbf{r}(\boldsymbol{\theta}^\mu)$ for convenience. The empirical loss we aim to minimize is a sum of the squared losses of each data point in the training set

$$\mathcal{L}(\mathbf{w}) = \frac{1}{2}\sum_{\mu=1}^{P}(\mathbf{r}^\mu \cdot \mathbf{w} - y^\mu)^2. \tag{40}$$

Performing gradient descent updates

$$\mathbf{w}_{t+1} = \mathbf{w}_t - \eta\frac{\partial\mathcal{L}}{\partial\mathbf{w}_t} = \mathbf{w}_t - \eta\sum_{\mu=1}^{P}\mathbf{r}^\mu(\mathbf{r}^\mu \cdot \mathbf{w}_t - y^\mu), \tag{41}$$

recovers the delta rule that we discussed in the main text (**Widrow and Hoff, 1960**; **Hertz et al., 1991**). Letting the empirical response matrix $\mathbf{R} = [\mathbf{r}^1, ..., \mathbf{r}^P] \in \mathbb{R}^{N \times P}$ have a SVD $\mathbf{R} = \sum_k \sqrt{\hat{\lambda}_k}\hat{\mathbf{u}}_k\hat{\boldsymbol{\psi}}_k^\top$,

and expanding the weights $\mathbf{w}_t = \sum_k w_{t,k}\hat{\mathbf{u}}_k$ and labels $\mathbf{y} = \sum_k \hat{v}_k\hat{\psi}_k$ in their respective SVD bases, we find

$$w_{t+1,k} = w_{t,k} - \eta\hat{\lambda}_k w_{t,k} + \eta\sqrt{\hat{\lambda}_k}\hat{v}_k \tag{42}$$

For all directions with $\hat{\lambda}_k > 0$, the dynamics converge to the unique fixed point $w_k^* = \frac{\hat{v}_k}{\sqrt{\hat{\lambda}_k}}$, while for all modes with $\hat{\lambda}_k = 0$, the weights remain at $w_k^* = 0$. Thus

$$\mathbf{w}^* = \left[\sum_{k:\hat{\lambda}_k>0}\frac{\hat{\mathbf{u}}_k\hat{\psi}_k^\top}{\sqrt{\hat{\lambda}_k}}\right]\mathbf{y} = \mathbf{R}\left[\sum_{k:\hat{\lambda}_k>0}\frac{\hat{\psi}_k\hat{\psi}_k^\top}{\hat{\lambda}_k}\right]\mathbf{y} = \mathbf{R}\mathbf{K}^+\mathbf{y} \tag{43}$$

where $K^+$ is the Moore-Penrose inverse of the kernel matrix $K_{\mu,\nu} = K(\theta^\mu, \theta^\nu)$. The predictions of the learned function are given by $f = \mathbf{w}^* \cdot \mathbf{r}(\theta)$ which can be expressed as

$$f(\theta) = \mathbf{k}(\theta)^\top\mathbf{K}^+\mathbf{y} \tag{44}$$

The fact that the solution can be written in terms of a linear combination of $\{K(\theta, \theta^\mu)\}_{\mu=1}^P$ is known as the representer theorem (**Schölkopf et al., 2001**; **Rasmussen and Williams, 2005**). A similar analysis for nonlinear readouts where $f(\theta) = g\left(\mathbf{w} \cdot \mathbf{r}(\theta)\right)$ is provided in Appendix Convergence of delta-rule for nonlinear readouts.

## Weight decay and ridge regression

We can introduce a regularization term in our learning problem which penalizes the size of the readout weights. This leads to a modified learning objective of the form

$$\mathcal{L}(\mathbf{w}) = \sum_\mu(\mathbf{r}^\mu \cdot \mathbf{w} - y^\mu)^2 + \lambda\|\mathbf{w}\|^2. \tag{45}$$

Inclusion of this regularization alters the learning rule through *weight decay* $\mathbf{w}_{t+1} = (1 - \eta\lambda)\mathbf{w}_t - \eta\sum_\mu \mathbf{r}^\mu(\mathbf{r}^\mu \cdot \mathbf{w}_t - y^\mu)$, which multiplies the existing weight value by a factor of $1 - \eta\lambda$ before adding the data dependent update. The fixed point of these dynamics is $\mathbf{w} = (\mathbf{R}\mathbf{R}^\top + \lambda\mathbf{I})^{-1}\mathbf{R}\mathbf{y}$. This learning problem and gradient descent dynamics have a closed form solution

$$f(\theta) = \mathbf{r}(\theta) \cdot \mathbf{w}^* = \sum_{\mu=1}^P \alpha^\mu K(\theta, \theta^\mu)\,, \quad \alpha = (\mathbf{K} + \lambda\mathbf{I})^{-1}\mathbf{y}. \tag{46}$$

The generalization benefits of explicit regularization through weight decay is known to be related to the noise statistics in the learning problem (**Canatar et al., 2021**). This is visible in the **Appendix 1—figure 1**, where unlearnable target functions demand nonzero optimal regularization. We simulate weight decay only in **Figure 6C**, where we use $\lambda = 0.01\sum_k \lambda_k$ to improve numerical stability at large $P$.

## Computation of learning curves

Recent work has established analytic results that predict the average case generalization error for kernel regression

$$E_g = \left\langle E_g(\mathcal{D})\right\rangle_\mathcal{D} = \left\langle (f(\theta, \mathcal{D}) - y(\theta))^2\right\rangle_{\theta,\mathcal{D}} \tag{47}$$

where $E_g(\mathcal{D}) = \left\langle (f(\theta, \mathcal{D}) - y(\theta))^2\right\rangle_\theta$ is the generalization error for a certain sample $\mathcal{D}$ of size $P$ and $f(\theta, \mathcal{D})$ is the kernel regression solution for $\mathcal{D}$ (**Bordelon et al., 2020**; **Canatar et al., 2021**). The typical or average case error $E_g$ is obtained by averaging over all possible datasets of size $P$. This average case generalization error is determined solely by the decomposition of the target function $y(\mathbf{x})$ along the eigenbasis of the kernel and the eigenspectrum of the kernel. This diagonalization takes the form

$$\int p(\theta)K(\theta, \theta')\psi_k(\theta)d\theta = \lambda_k\psi_k(\theta') \tag{48}$$

Since the eigenfunctions form a complete set of square integrable functions, we expand both the target function $y(\boldsymbol{\theta})$ and the learned function $f(\boldsymbol{\theta})$ in this basis

$$y(\boldsymbol{\theta}) = \sum_k v_k \psi_k(\boldsymbol{\theta}) \,, \quad f(\boldsymbol{\theta}) = \sum_k \hat{v}_k \psi_k(\boldsymbol{\theta}) \tag{49}$$

Due to the orthonormality of the kernel eigenfunctions $\{\psi_k\}$, the generalization error for any set of coefficients $\hat{v}$ is

$$E_g(\mathbf{w}) = \left\langle (y(\boldsymbol{\theta}) - f(\boldsymbol{\theta}))^2 \right\rangle_{\boldsymbol{\theta}} = \sum_k (\hat{v}_k - v_k)^2 = \|\hat{\mathbf{v}} - \mathbf{v}\|^2 \tag{50}$$

We now introduce training error, or empirical loss, which depends on the disorder in the dataset $\mathcal{D} = \{(\boldsymbol{\theta}^\mu, y^\mu)\}_{\mu=1}^P$

$$H(\hat{\mathbf{v}}, \mathcal{D}) = \sum_\mu (\hat{\mathbf{v}} \cdot \boldsymbol{\psi}(\boldsymbol{\theta}^\mu) - \mathbf{v} \cdot \boldsymbol{\psi}(\boldsymbol{\theta}^\mu))^2 + \lambda \sum_k \frac{\hat{v}_k^2}{\lambda_k} \tag{51}$$

It is straightforward to verify that the optimal $\hat{\mathbf{v}}^*$ which minimizes $H(\hat{\mathbf{v}}, \mathcal{D})$ is the kernel regression solution for kernel with eigenvalues $\{\lambda_k\}$ when $\lambda \to 0$. The optimal weights $\hat{\mathbf{v}}$ can be identified through the first order condition $\nabla H(\hat{\mathbf{v}}, \mathcal{D}) = 0$ which gives

$$\hat{\mathbf{v}}^* = (\boldsymbol{\Psi}\boldsymbol{\Psi}^\top + \lambda \boldsymbol{\Lambda}^{-1})^{-1} \boldsymbol{\Psi}\boldsymbol{\Psi}^\top \mathbf{v} = \mathbf{v} - \lambda(\boldsymbol{\Psi}\boldsymbol{\Psi}^\top + \lambda \boldsymbol{\Lambda}^{-1})^{-1} \boldsymbol{\Lambda}^{-1} \mathbf{v} \tag{52}$$

where $\Psi_{k,\mu} = \psi_k(\boldsymbol{\theta}^\mu)$ are the eigenfunctions evaluated on the training data and $\Lambda_{k,\ell} = \delta_{k,\ell}\lambda_k$ is a a diagonal matrix containing the kernel eigenvalues. The generalization error for this optimal solution is

$$E_g(\mathcal{D}) = \|\hat{\mathbf{v}}^* - \mathbf{v}\|^2 = \mathbf{v}^\top \boldsymbol{\Lambda}^{-1} \mathbf{G}(\mathcal{D})^2 \boldsymbol{\Lambda}^{-1} \mathbf{v} \,, \quad \mathbf{G}(\mathcal{D}) = \left(\frac{1}{\lambda}\boldsymbol{\Psi}\boldsymbol{\Psi}^\top + \boldsymbol{\Lambda}^{-1}\right)^{-1} \tag{53}$$

We note that the dependence on the randomly sampled dataset $\mathcal{D}$ only appears through the matrix $\mathbf{G}(\mathcal{D})$. Thus to compute the *typical* generalization error we need to average over this matrix $\langle \mathbf{G}(\mathcal{D}) \rangle_{\mathcal{D}}$. There are multiple strategies to perform such an average and we will study one here based on a partial differential equation which was introduced in *Sollich, 1998*; *Sollich, 2002* and studied further in *Bordelon et al., 2020*; *Canatar et al., 2021*. In this setting, we denote the average matrix $\mathbf{G}(P) = \langle \mathbf{G}(\mathcal{D}) \rangle_{|\mathcal{D}|=P}$ for a dataset of size $P$. We first will derive a recursion relationship using the Sherman Morrison formula for a rank-1 update to an inverse matrix. We imagine adding a new sampled feature vector $\phi$ to a dataset $\psi$ with size $P$. The average matrix $\mathbf{G}(P+1)$ at $P+1$ samples can be related to $\mathbf{G}(P)$ through the Sherman Morrison rule

$$\begin{aligned}
\mathbf{G}(P+1) &= \left\langle \left(\frac{1}{\lambda}\boldsymbol{\Psi}\boldsymbol{\Psi}^\top + \frac{1}{\lambda}\psi\psi^\top + \boldsymbol{\Lambda}^{-1}\right)^{-1} \right\rangle_{\psi,\mathcal{D}} = \mathbf{G}(P) - \left\langle \frac{\mathbf{G}(\mathcal{D})\psi\psi^\top\mathbf{G}(\mathcal{D})}{\lambda + \psi^\top\mathbf{G}(\mathcal{D})\psi} \right\rangle_{\psi,\mathcal{D}} \\
&\approx \mathbf{G}(P) - \frac{\left\langle \mathbf{G}(\mathcal{D})\langle\psi\psi^\top\rangle_\psi \mathbf{G}(\mathcal{D}) \right\rangle_{\mathcal{D}}}{\lambda + \langle\psi^\top\mathbf{G}(\mathcal{D})\psi\rangle_{\psi,\mathcal{D}}}
\end{aligned} \tag{54}$$

where in the last step we approximated the average of the ratio with the ratio of averages. This operation, is of course, unjustified theoretically, but has been shown to produce accurate learning curves (*Sollich, 2002*; *Bordelon et al., 2020*). Since the chosen basis of kernel eigenfunctions are orthonormal, the average over the new sample is trivial $\langle \psi\psi^\top \rangle_\psi = \mathbf{I}$. We thus arrive at the following recursion relationship for $\mathbf{G}$

$$\mathbf{G}(P+1) = \mathbf{G}(P) - \frac{\langle \mathbf{G}(\mathcal{D})^2 \rangle_{\mathcal{D}}}{\lambda + \mathrm{Tr}\,\mathbf{G}(P)} \tag{55}$$

By introducing an additional source $J$ so that $\mathbf{G}(\mathcal{D}, J)^{-1} = \frac{1}{\lambda}\boldsymbol{\Psi}\boldsymbol{\Psi}^\top + \boldsymbol{\Lambda}^{-1} + J\mathbf{I}$, we can relate $\mathbf{G}(\mathcal{D}, J)$'s first and second moments through differentiation

$$\frac{\partial}{\partial J}\mathbf{G}(P, J) = \frac{\partial}{\partial J}\left\langle \left(\frac{1}{\lambda}\boldsymbol{\Psi}\boldsymbol{\Psi}^\top + J\mathbf{I} + \boldsymbol{\Lambda}^{-1}\right)^{-1} \right\rangle_{\mathcal{D}} = -\left\langle \mathbf{G}(\mathcal{D}, J)^2 \right\rangle_{\mathcal{D}}. \tag{56}$$

Thus the recursion relation simplifies to

$$\mathbf{G}(P+1,J) - \mathbf{G}(P,J) \approx \frac{\partial}{\partial P}\mathbf{G}(P,J) = \frac{1}{\lambda+\text{Tr}\mathbf{G}(P,J)}\frac{\partial}{\partial J}\mathbf{G}(P,J), \tag{57}$$

where we approximated the finite difference in $P$ as a derivative, treating $P$ as a continuous variable. Taking the trace of both sides and defining $\kappa(P,J) = \lambda + \text{Tr}\mathbf{G}(P,J)$ we arrive at the following quasilinear PDE

$$\frac{\partial}{\partial P}\kappa(P,J) = \frac{1}{\kappa(P,J)}\frac{\partial}{\partial J}\kappa(P,J) \tag{58}$$

with the initial condition $\kappa(0,J) = \lambda + \text{Tr}(\mathbf{\Lambda}^{-1} + J\mathbf{I})^{-1}$. Using the method of characteristics, we arrive at the solution $\kappa(P,J) = \lambda + \text{Tr}\left(\mathbf{\Lambda}^{-1} + (v + \frac{P}{\kappa(P,J)})\mathbf{I}\right)^{-1}$. Using this solution to $\kappa$, we can identify the solution to $\mathbf{G}$

$$\mathbf{G}(P,J)_{k,\ell} = \left(\frac{P}{\kappa} + J + \lambda_k^{-1}\right)^{-1}\delta_{k,\ell} = \frac{\kappa\lambda_k}{\lambda_k P + \kappa + J\kappa\lambda_k}\delta_{k,\ell}. \tag{59}$$

The generalization error, therefore can be written as

$$E_g = \mathbf{v}^\top\mathbf{\Lambda}^{-1}\left\langle\mathbf{G}(\mathcal{D})^2\right\rangle_{\mathcal{D}}\mathbf{\Lambda}^{-1}\mathbf{v} = -\frac{\partial}{\partial J}\mathbf{v}^\top\mathbf{\Lambda}^{-1}\mathbf{G}(P,J)\mathbf{\Lambda}^{-1}\mathbf{v} \tag{60}$$

$$= -\sum_k\frac{v_k^2}{\lambda_k^2}\frac{\partial}{\partial J}\left(\frac{P}{\kappa} + J + \lambda_k^{-1}\right)^{-1} = \frac{\kappa^2}{1-\gamma}\sum_k\frac{v_k^2}{(\lambda_k P + \kappa)^2}, \tag{61}$$

where $\gamma = P\sum_k\frac{\lambda_k^2}{(\lambda_k P + \kappa)^2}$, giving the desired result. Note that $\kappa$ depends on $J$ implicitly, which is the source of the $\frac{1}{1-\gamma}$ factor. This result was recently reproduced using techniques from statistical mechanics (*Bordelon et al., 2020*; *Canatar et al., 2021*).

## Spectral bias and code-task alignment

Through implicit differentiation it is straightforward to verify that the ordering of the mode errors $E_k = \frac{\kappa^2}{1-\gamma}(\lambda_k P + \kappa)^{-2}$ matches the ordering of the eigenvalues (*Canatar et al., 2021*). Let $\lambda_k > \lambda_\ell$, then we have

$$\frac{d}{dP}\log\left(\frac{E_k}{E_\ell}\right) = 2\left[\frac{\lambda_\ell}{\lambda_\ell P + \kappa} - \frac{\lambda_k}{\lambda_k P + \kappa}\right] + 2\kappa'(P)\left[\frac{1}{\lambda_\ell P + \kappa} - \frac{1}{\lambda_k P + \kappa}\right]. \tag{62}$$

Since $\lambda_\ell < \lambda_k$, the first bracket must be negative and the second bracket must be positive. Further, it is straightforward to compute that $\kappa'(P) = -\frac{\kappa\gamma}{P(1+\gamma)} < 0$. Therefore $\lambda_k > \lambda_\ell$ implies $\frac{d}{dP}\log\left(\frac{E_k}{E_\ell}\right) < 0$ for all $P$. Since $\log\left(\frac{E_k}{E_\ell}\right) = 0$ at $P = 0$ we therefore have that $\log(E_k/E_\ell) < 0$ for all $P$ and consequently $E_k < E_\ell$. Modes with larger eigenvalues $\lambda_k$ have lower normalized mode errors $E_k$. This observation can be used to prove that target functions acting on the same data distribution with higher cumulative power distributions $C(k)$ for all $k$ will have lower generalization error normalized by total target power, $E_g(P)/E_g(0)$, for all $P$. Proof can be found in *Canatar et al., 2021*.

## Asymptotic power law scaling of learning curves

*Exponential spectral decays:*
First, we will study the setting relevant to the von-Mises kernel where $\lambda_k \sim \beta^k$ and $v_k^2 \sim \alpha^k$ where $\alpha, \beta < 1$. This exponential behavior accounts for differences in bandwidth between kernels which modulates the base $\beta$ of the exponential scaling of $\lambda_k$ with $k$. We will approximate the sum over all mode errors with an integral

$$E_g = \frac{\kappa^2}{1-\gamma}\sum_{k=0}^\infty\frac{v_k^2}{(\lambda_k P + \kappa)^2} \sim \kappa^2\int_0^\infty\frac{\alpha^k}{(\beta^k P + \kappa)^2}dk. \tag{63}$$

If we include a regularization parameter $\lambda$, then $\kappa \sim \lambda$ as $P \to \infty$. Making the change of variables $u = P\beta^k/\lambda$, we transform the above integral into

$$E_g \sim \frac{1}{\ln(1/\beta)} \left(\frac{\lambda}{P}\right)^{\ln(1/\alpha)/\ln(1/\beta)} \int_0^{P/\lambda} \frac{u^{\frac{\ln(1/\alpha)}{\ln(1/\beta)}-1}}{(1+u)^2} du$$

The remaining integral over $u$ is either dominated near $u \approx 0$ if $\frac{\ln(1/\alpha)}{\ln(1/\beta)} < 2$ and behaves as a $P$-independent constant or else is dominated near $u \approx P/\lambda$, in which case the integral scales as $\sim P^{\frac{\ln(1/\alpha)}{\ln(1/\beta)}-2}$. Multiplying these resulting functions with the prefactor, we find the following scaling laws for generalization.

$$E_g \sim \begin{cases} P^{-\frac{\ln(1/\alpha)}{\ln(1/\beta)}} & \frac{\ln(1/\alpha)}{\ln(1/\beta)} < 2 \\ P^{-2} & \frac{\ln(1/\alpha)}{\ln(1/\beta)} > 2 \end{cases} \tag{64}$$

Thus, we obtain a power law scaling of the learning curve $E_g$ which is dominated at large $P$ by $E_g \sim P^{-\min\left(2, \frac{\ln(1/\alpha)}{\ln(1/\beta)}\right)}$. For the von-Mises kernel we can approximate the spectra with $\lambda_k \sim \sigma^{-2k}$ and $v_k^2 \sim \sigma_T^{-2k}$ giving rise to a generalization scaling scaling $E_g \sim P^{-\min\left(2, \frac{\ln \sigma_T}{\ln \sigma}\right)}$.

## Power law spectral decays
The same arguments can be applied for power law kernels $\lambda_k \sim k^{-b}$ and power law targets $v_k^2 \sim k^{-a}$, which is of interest due to its connection to nonlinear rectified neural populations. In this setting, the generalization error is

$$E_g \approx \int_1^\infty \frac{k^{-a}}{(k^{-b}P + \kappa)^2} dk \approx \frac{\kappa^2}{P^2} \int_1^{P^{1/b}} k^{-a+2b} dk + \int_{P^{1/b}}^\infty k^{-a} dk$$
$$= \frac{1}{P^2(1-a+2b)} \left[P^{(1-a)/b+2} - 1\right] + \frac{1}{a-1} P^{(1-a)/b}. \tag{65}$$

We see that there are two possible power law scalings for $E_g$ with the exponents $(a-1)/b$ and 2. At large $P$ this formula will be dominated by the term with minimum exponent so $E_g \sim P^{-\min(a-1,2b)/b}$.

## Laplace kernel generalization
We calculate similar learning curves as we did for the von-Mises kernel but with Laplace kernels to show that our results is not an artifact of the infinite differentiability of the Von Mises kernel. Each of these Laplace kernels has the same asymptotic power law spectrum $\lambda_k \sim o(k^{-2})$, exhibiting a discontinuous first derivative (**Figure 6A**). Despite having the same spectral scaling at large $k$, these kernels can give dramatically different performance in learning tasks, again indicating the influence of the top eigenvalues on generalization at small $P$ (**Figure 6**). Again, the trend for which kernels perform best at low $P$ can be reversed at large $P$. In this case, all generalization errors scale with $E_g \sim P^{-2}$ (**Figure 6B**). More generally, our theory shows that if the task power spectrum and kernel eigenspectrum are both falling as power laws with exponents $a$ and $b$ respectively, then the generalization error asymptotically falls with a power law, $E_g \sim P^{-\min(a-1,2b)/b}$ (Methods) (**Bordelon et al., 2020**). This decay is fastest when $b \geq \frac{a-1}{2}$ for which $E_g \sim P^{-2}$. Therefore, the tail of the kernel's eigenvalue spectrum determines the large sample size behavior of the generalization error for power law kernels. Small sample size limit is still governed by the bulk of the spectrum.

## Learning with multiple output channels
Our theory is not limited to scalar target functions but rather can be easily extended to multiple output functions $y_1, ..., y_C$ from the same data, if for example the task requires computing class membership for $C$ categories. In this setting, each data point has the form $(\boldsymbol{\theta}^\mu, \mathbf{y}^\mu)$ with $\mathbf{y}^\mu \in \mathbb{R}^C$. For these $C$ classes, the generalization error takes the form

$$E_g = \left\langle \|\mathbf{f}(\boldsymbol{\theta}) - \mathbf{y}(\boldsymbol{\theta})\|^2 \right\rangle = \sum_{c=1}^C \left\langle (f_c(\boldsymbol{\theta}) - y_c(\boldsymbol{\theta}))^2 \right\rangle = \sum_k \left[ \sum_c \left\langle y_c(\boldsymbol{\theta}) \phi_k(\boldsymbol{\theta}) \right\rangle^2 \right] E_k. \tag{66}$$

We therefore find that the generalization error in the multi-class setting is the same as the $E_g$ obtained for a single scalar target function with power spectrum $v_k^2 = \sum_c \left\langle y_c(\boldsymbol{\theta}) \phi_k(\boldsymbol{\theta}) \right\rangle^2$ (**Bordelon et al., 2020**; **Canatar et al., 2021**). The relevant cumulative power distribution measures the fraction of total output variance captured by the first $k$ eigenfunctions of the population code

$$C(k) = \frac{\sum_c \sum_{\ell=1}^{k} \langle y_c(\boldsymbol{\theta}) \phi_k(\boldsymbol{\theta}) \rangle^2}{\sum_c \sum_{\ell=1}^{\infty} \langle y_c(\boldsymbol{\theta}) \phi_k(\boldsymbol{\theta}) \rangle^2} \, . \tag{67}$$

## Convergence of Delta-rule for nonlinear readouts

In this section, we consider gradient descent dynamics of an arbitrary convex loss function. For instance, we can consider a binary classification problem where $y \in \{\pm 1\}$ by outputting a prediction of $\hat{y} = \text{sign}(\mathbf{w} \cdot \mathbf{r})$. We could, for example, train a model using the hinge loss $\ell(\mathbf{w} \cdot \mathbf{r}, y) = \max(0, 1 - \mathbf{w} \cdot \mathbf{r}y)$ so that the classifier will converge to a kernel support vector machine (SVM) (*Schölkopf et al., 2002*). The generalization of the classifier would be the error rate of $\hat{y}(\boldsymbol{\theta}) = \text{sign}(\mathbf{w} \cdot \mathbf{r}(\boldsymbol{\theta}))$ compared to the ground truth $y(\boldsymbol{\theta})$.

Let $\mathcal{D} = \{\boldsymbol{\theta}^\mu, y^\mu\}_{\mu=1}^P$ be the dataset with samples $\boldsymbol{\theta}^\mu$ and target values $y^\mu$. We introduce a shorthand $\mathbf{r}^\mu = \mathbf{r}(\boldsymbol{\theta}^\mu)$ for convenience. The loss we aim to minimize is the sum of the losses of each data point in the training set with an additional weight decay parameter

$$H(\mathbf{w}, \mathcal{D}) = \sum_{\mu=1}^P \ell(\mathbf{w} \cdot \mathbf{r}^\mu, y^\mu) + \lambda |\mathbf{w}|^2. \tag{68}$$

For convex $\ell$ and nonzero $\lambda$, the above objective is strongly convex, indicating the existence of a unique minimizer which can be found from simple first order learning rules. For $\lambda > 0$ the initial condition for $\mathbf{w}$ does not influence the final result.

We will now show that the dynamics will converge to a function which only depends on the code $\mathbf{r}(\boldsymbol{\theta})$ through the kernel $K(\boldsymbol{\theta}, \boldsymbol{\theta}')$. To simplify the argument, we consider starting from an initial condition of $\mathbf{w}_{t=0} = \mathbf{0}$ and performing gradient descent updates. Under such an assumption, the weights $\mathbf{w}_t$ will always be in the span of the population vectors on the training set $\{\mathbf{r}^\mu\}_{\mu=1}^P$ since

$$\mathbf{w}_{t+1} = \mathbf{w}_t - \eta \frac{\partial H}{\partial \mathbf{w}}|_{\mathbf{w}_t} = (1 - \eta\lambda)\mathbf{w}_t - \eta \sum_{\mu=1}^P \mathbf{r}^\mu \, \ell'(\mathbf{w} \cdot \mathbf{r}^\mu, y^\mu). \tag{69}$$

The derivative in the final term is taken with respect to the first argument $\ell'(f, y) = -\frac{\partial \ell(f, y)}{\partial f}$. This update is still local and recovers the delta rule that we discussed in the main text for $\ell(\mathbf{w} \cdot \mathbf{r}, y) = \frac{1}{2}(\mathbf{w} \cdot \mathbf{r} - y)^2$ (*Widrow and Hoff, 1960*; *Hertz et al., 1991*). We can express $\mathbf{w}_t$ in terms of the population vectors $\mathbf{w}_t = \sum_{\mu=1}^P \alpha_t^\mu \mathbf{r}^\mu = \mathbf{R}\boldsymbol{\alpha}_t$ so that $\boldsymbol{\alpha}_t \in \mathbb{R}^P$ defines the linear weighting of each sample. The dynamics of these coefficients are

$$\mathbf{R}\boldsymbol{\alpha}_{t+1} = (1 - \eta\lambda)\mathbf{R}\boldsymbol{\alpha}_t - \eta\mathbf{R}\ell'(\mathbf{K}\boldsymbol{\alpha}, \mathbf{y}), \tag{70}$$

where $\mathbf{K} = \mathbf{R}^\top \mathbf{R} \in \mathbb{R}^{P \times P}$ is the kernel Gram matrix evaluated on the training points. We multiply both sides of this equation by $\mathbf{R}^\top$, and define $\boldsymbol{\beta}_t = \mathbf{K}\boldsymbol{\alpha}_t$, which satisfy the following simplified dynamics

$$\boldsymbol{\beta}_{t+1} = (1 - \eta\lambda)\boldsymbol{\beta}_t - \eta\mathbf{K}\ell'(\boldsymbol{\beta}_t, \mathbf{y}) \, , \ \mathbf{w}_t = \mathbf{R}\mathbf{K}^+\boldsymbol{\beta}_t. \tag{71}$$

where $\mathbf{K}^+$ is the pseudo-inverse of $\mathbf{K}$. The nonlinear fixed point condition is $\boldsymbol{\beta} = -\frac{1}{\lambda}\mathbf{K}\ell'(\boldsymbol{\beta}, \mathbf{y})$, which transparently only depends on the kernel $\mathbf{K}$ rather than the full code $\mathbf{R}$. The above equation recovers the correct linear equation $\boldsymbol{\beta} = \mathbf{K}(\mathbf{K} + \lambda\mathbf{I})^{-1}\mathbf{y}$ for square loss. For an arbitrary test point $\boldsymbol{\theta}$, the model makes prediction using $f(\boldsymbol{\theta}) = \mathbf{r}(\boldsymbol{\theta}) \cdot \mathbf{w} = \mathbf{r}(\boldsymbol{\theta}) \cdot [\mathbf{R}\mathbf{K}^+\boldsymbol{\beta}] = \mathbf{k}(\boldsymbol{\theta}) \cdot \boldsymbol{\alpha}$, which also only depends on the kernel $[\mathbf{k}(\boldsymbol{\theta})]_\mu = K(\boldsymbol{\theta}, \boldsymbol{\theta}^\mu)$ on test point $\boldsymbol{\theta}$ and train points $\hat{\boldsymbol{\theta}}_\mu$, as well as the kernel gram matrix $[\mathbf{K}]_{\mu\nu} = K(\boldsymbol{\theta}^\mu, \boldsymbol{\theta}^\nu)$.

To illustrate a specific case with a square error and nonlinear readout, consider output neurons which produce activity $g(\mathbf{w} \cdot \mathbf{r}(\boldsymbol{\theta}))$ for invertible nonlinear function $g$ with non-vanishing gradient, and gradient based learning on $L = \sum_\mu (g(\mathbf{w} \cdot \mathbf{r}^\mu) - y(\boldsymbol{\theta}^\mu))^2$. This gives $\Delta\mathbf{w} \propto \sum_\mu \mathbf{r}^\mu g'(\mathbf{w} \cdot \mathbf{r}^\mu)(y^\mu - g(\mathbf{w} \cdot \mathbf{r}^\mu)) \in \text{span}\{\mathbf{r}^\mu\}_{\mu=1}^P$, which is still a local learning rule. Thus the weights at convergence can be written as $\mathbf{w} = \sum_\mu \alpha^\mu \mathbf{r}^\mu$ and the learned function can be written as $f(\boldsymbol{\theta}) = g\left(\sum_\mu \alpha^\mu K(\boldsymbol{\theta}, \boldsymbol{\theta}^\mu)\right)$ is given by $\boldsymbol{\alpha} = \mathbf{K}^+ g^{-1}(\mathbf{y})$. To see this, first note that $\mathbf{w}_t \in \text{span}\{\mathbf{r}^\mu\}_{\mu=1}^P$ for all $t$ so that $\mathbf{w}^* = \mathbf{R}\boldsymbol{\alpha}^*$. The interpolation condition can be expressed as $g\left(\mathbf{R}^\top \mathbf{w}^*\right) = g(\mathbf{K}\boldsymbol{\alpha}) = \mathbf{y}$, giving the desired result $\boldsymbol{\alpha}^* = \mathbf{K}^+ g^{-1}(\mathbf{y})$. The predictions of the model on a test stimulus $\boldsymbol{\theta}$ are given by $f(\boldsymbol{\theta}) = g\left(\sum_{\mu=1}^P \alpha^{*\mu} K(\boldsymbol{\theta}, \boldsymbol{\theta}^\mu)\right)$. We see that this solution only depends on the kernel (directly and indirectly through $\boldsymbol{\alpha}^*$), rather than the full code.

## Typical case analysis of nonlinear readouts

The analysis of typical case generalization can be extended to nonlinear predictors and loss functions described by (68) which depend on the scalar prediction variable $\mathbf{w} \cdot \mathbf{r}(\boldsymbol{\theta})$ (*Loureiro et al., 2021a*). Thanks Further, if $\mathbf{r}(\boldsymbol{\theta})$ is well approximated as a Gaussian process, then the generalization performance can still be characterized using statistical mechanics methods (*Loureiro et al., 2021a*). Many qualitative features of our results continue to hold, including that the kernel's diagonalization governs training and generalization and that improvements in code task alignment lead to improvements in generalization.

In a later work by *Cui et al., 2022*, SVM and ridge classifiers trained on codes and tasks with power law spectra were analyzed asymptotically, showing power law generalization error decay rates $E_g \sim P^{-\beta}$. These classification learning curves for power law spectra were shown to follow power laws with exponents $\beta$ which are qualitatively similar to the exponents obtained with the square loss which we describe in our section titled Small and Large Sample Size Behaviors of Generalization. Just as in our theory, decay rate exponents $\beta$ are larger for codes which are well aligned to the task and are smaller for codes which are non-aligned.

## Visual scene reconstruction task

### Reconstruction of natural scenes from neural responses

Using the mouse V1 responses to natural scenes, we attempt to reconstruct original images from the neural codes using different numbers of images. The presented natural scenes are taken from ten classes of imagenet which can be downloaded from https://github.com/MouseLand/stringer-pachitariu-et-al-2018b. Let $\boldsymbol{\theta}^{\mu} \in \mathbb{R}^D$ be a $D$-dimensional flattened vector containing the pixel values of the μ-th image and let $\mathbf{r}^{\mu} \in \mathbb{R}^N$ represent the neural response to the μ-th image. The goal in the problem is to learn a collection of weights $\mathbf{W} \in \mathbb{R}^{D \times N}$ which map neural responses $\mathbf{r}^{\mu}$ to images $\boldsymbol{\theta}^{\mu}$

$$\boldsymbol{\theta}^{\mu} \approx \mathbf{W}\mathbf{r}^{\mu}. \tag{72}$$

The generalization error $E_g$ again measures the average error on all points, averaged over all possible datasets $\mathcal{D} = \{(\boldsymbol{\theta}^{\mu}, \mathbf{r}^{\mu})\}_{\mu=1}^{P}$ of size $P$. If the optimal weights for dataset $\mathcal{D}$ is $\mathbf{W}(\mathcal{D})$ then the generalization error is

$$E_g = \left\langle \|\mathbf{W}(\mathcal{D})\mathbf{r}(\boldsymbol{\theta}) - \boldsymbol{\theta}\|^2 \right\rangle_{\boldsymbol{\theta}, \mathcal{D}}. \tag{73}$$

After identifying eigenfunctions $\phi_k(\boldsymbol{\theta})$, we expand the images in this basis $\boldsymbol{\theta} = \sum_k \mathbf{v}_k \psi_k(\boldsymbol{\theta})$ where $\mathbf{v}_k \in \mathbb{R}^D$. The generalization error is therefore $E_g = \sum_k |\mathbf{v}_k|^2 E_k(P)$ and the cumulative power is $C(k) = \frac{\sum_{\ell < k} |\mathbf{v}_\ell|^2}{\sum_{\ell=1}^{\infty} |\mathbf{v}_\ell|^2}$. We perform this reconstruction task on many filtered versions of the natural scenes. To construct a filter, we first compute the Fourier transform of the image. Let $\mathbf{M}(\boldsymbol{\theta}) \in \mathbb{R}^{\sqrt{D} \times \sqrt{D}}$ represent the non-flattened image and let $\hat{\mathbf{M}}(\boldsymbol{\theta}) \in \mathbb{R}^{\sqrt{D} \times \sqrt{D}}$ represent the Fourier transform of the image, computed explicitly as

$$\hat{M}_{k,k'}(\boldsymbol{\theta}) = D^{-1/4} \sum_{n,m} M_{n,m}(\boldsymbol{\theta}) \exp(2\pi i (nk + mk')/\sqrt{D}) \tag{74}$$

To develop the band-pass filter, we calculate $|\mathbf{k}| = \sqrt{k^2 + (k')^2}$ for each of the indices in the matrix. For a band-pass filter with parameters $s_{max}, r$ we simply zero out the entries in $\hat{M}$ which correspond to states with frequencies outside the appropriate band: for any $k, k'$ with $|\mathbf{k}| \notin [\sqrt{s_{max}^2 - r^2}, s_{max}^2]$ then $\hat{M}_{k,k'} \to 0$. We then perform the inverse Fourier transform on $\hat{M}$ to obtain a filtered version of the original image.

## A simple feedforward model of V1

### Linear neurons

We consider a simplified but instructive model of the V1 population code as a linear-nonlinear map from photoreceptor responses through Gabor filters and then nonlinearity (*Adelson and Bergen, 1985*; *Olshausen and Field, 1997*; *Rumyantsev et al., 2020*). Let $\mathbf{x} \in \mathbb{R}^2$ represent the two-dimensional retinotopic position of photoreceptors. The firing rates of the photoreceptor at position $\mathbf{x}$ to a static grating stimulus oriented at angle $\theta$ is

$$h(\mathbf{x}, \theta) = \cos(\mathbf{k}(\theta) \cdot \mathbf{x}) , \ \mathbf{k} = \begin{bmatrix} \cos(\theta) \\ \sin(\theta) \end{bmatrix} \in \mathbb{R}^2 , \ \theta \in [0, 2\pi]. \tag{75}$$

We model each V1 neuron's receptive field as a Gabor filter of the receptor responses $h(\mathbf{x}, \theta)$. The $i$-th V1 neuron has preferred wavevector $\mathbf{k}_i$, generating the following set of weights between photoreceptors and the $i$-th V1 neuron

$$\mathcal{F}(\mathbf{x}, \theta_i) = \frac{\sigma^2}{2\pi} e^{-\frac{\sigma^2}{2}|\mathbf{x}|^2} \cos(\mathbf{k}(\theta_i) \cdot \mathbf{x}). \tag{76}$$

The V1 population code is obtained by filtering the photoreceptor responses. By approximating the resulting sum over all retinal photoreceptors with an integral, we find the response of neuron $i$ to grating stimulus with wavenumber $\mathbf{k}$ is

$$\mathbf{h}(\theta) \cdot \mathcal{F}(\theta_i) = \int \mathcal{F}(\mathbf{x}, \theta_i) h(\mathbf{x}, \theta) d\mathbf{x} = \frac{1}{2} e^{-\frac{1}{2\sigma^2}|\mathbf{k}+\mathbf{k}_i|^2} + \frac{1}{2} e^{-\frac{1}{2\sigma^2}|\mathbf{k}-\mathbf{k}_i|^2}. \tag{77}$$

The response of neuron $i$ is computed through nonlinear rectification of this input current $r_i(\theta) = g(\mathbf{w}(\theta_i) \cdot \mathbf{h}(\theta))$. For a linear neuron $g(z) = z$, the kernel has the following form

$$K(\theta, \theta') = \frac{\cosh(\cos(\theta-\theta')/\sigma^2)}{\cosh(\sigma^{-2})}, \tag{78}$$

where the kernel is normalized to have maximum value of 1. Note that this normalization of the kernel is completely legitimate since it merely rescales each eigenvalue by a constant and does not change the learning curves.

Since the kernel only depends on the difference between angles $\theta - \theta'$, it is said to posess translation invariance. Such translation invariant kernels admit a Mercer decomposition in terms of Fourier modes $K(\theta) = \sum_n \lambda_n \cos(n\theta)$ since the Fourier modes diagonalize shift invariant integral operators on $\mathbb{S}^1$. For the linear neuron, the kernel eigenvalues scale like $\lambda_n \sim \frac{\beta^n}{2^n n!}$, indicating infinite differentiability of the tuning curves. Since $\lambda_n$ decays rapidly with $n$, we find that this Gabor code has an inductive bias that favors low frequency functions of orientation $\theta$.

## Nonlinear simple cells

Introducing nonlinear functions $g(z)$ that map input currents $z$ into the V1 population into firing rates, we can obtain a non-linear kernel $K_g(\theta)$ which has the following definition

$$K_g(\mathbf{k}, \mathbf{k}') = \int p(\mathbf{k}_i) g(\mathcal{F}(\mathbf{k}_i) \cdot \mathbf{h}(\mathbf{k})) g(\mathcal{F}(\mathbf{k}_i) \cdot \mathbf{h}(\mathbf{k}')) d\mathbf{k}_i. \tag{79}$$

In this setting, it is convenient to restrict $\mathbf{k}_i, \mathbf{k}, \mathbf{k}' \in \mathbb{S}^1$ and assume that the preferred wavevectors $\mathbf{k}_i$ are uniformly distributed over the circle. In this case, it suffices to identify a decomposition of the composed function $g(\mathbf{w}_i \cdot \mathbf{h}(\theta))$ in the basis of Chebyshev polynomials $T_n(z)$ which satisfy $T_n(\cos(\theta)) = \cos(n\theta)$

$$
\begin{aligned}
a_n &= \frac{1}{2\pi} \int_0^{2\pi} g\left(e^{-\frac{1}{\sigma^2}} \cosh\left(\frac{1}{\sigma^2}\cos(\theta)\right)\right) \cos(n\theta) d\theta \\
&= \frac{1}{2\pi} \int_{-1}^1 \frac{1}{\sqrt{1-z^2}} g\left(e^{-\sigma^{-2}} \cosh\left(z\sigma^{-2}\right)\right) T_n(z) dz,
\end{aligned}
\tag{80}
$$

which can be computed efficiently with an appropriate quadrature scheme. Once the coefficients $a_n$ are determined, we can compute the kernel by first letting $\theta_i$ to be the angle between $\mathbf{k}$ and $\mathbf{k}_i$ and letting $\theta$ be the angle between $\mathbf{k}$ and $\mathbf{k}'$

$$K_g(\theta) = \int_0^{2\pi} \frac{d\theta_i}{2\pi} \sum_{n,n'} a_n a_{n'} T_n(\cos(\theta_i)) T_{n'}(\cos(\theta_i + \theta)) d\theta_i = \frac{1}{2} \sum_n a_n^2 \cos(n\theta). \tag{81}$$

Thus the kernel eigenvalues are $\lambda_n = \frac{1}{2} a_n^2(\psi)$.

## Asymptotic scaling of spectra

Activation functions that encourage sparsity have slower eigenvalue decays. If the nonlinear activation function has the form $g_{q,a}(z) = \max\{0, z - a\}^q$, then the spectrum decays like $\lambda_n \sim n^{-2q-2}$. A simple

argument justifies this scaling: if the function $g(e^{-\sigma^{-2}}\cosh(\sigma^{-2}z))$ is only $q - 1$ times differentiable then $a_n n^q \sim n^{-1}$ since $\sum_n a_n n^q$ must diverge. Therefore $\lambda_n = a_n^2 \sim n^{-2q-2}$. Note that this scaling is independent of the threshold. Examples of these scalings can be found in *Figure 5—figure supplements 1 and 2*.

## Phase variation, complex cells and invariance

We can consider a slightly more complicated model where Gabors and stimuli have phase shifts

$$h(\mathbf{x}, \theta, \phi) = \cos(\mathbf{k}(\theta) \cdot \mathbf{x} - \phi) , \quad \mathcal{F}(\mathbf{x}, \theta_i, \phi_i) = \tfrac{\sigma^2}{2\pi} e^{-\frac{\sigma^2}{2}|\mathbf{x}|^2} \cos(\mathbf{k}_i \cdot \mathbf{x} - \phi_i). \tag{82}$$

The simple cells are generated by nonlinearity

$$r_i(\theta, \phi) = g(\mathcal{F}(\theta_i, \phi_i) \cdot \mathbf{h}(\theta, \phi)). \tag{83}$$

The input currents into the simple V1 cells can be computed exactly

$$\begin{aligned}
\mathbf{h}(\theta, \phi) \cdot \mathcal{F}(\theta_i, \phi_i) &= \left\langle \cos(\mathbf{k}_i \cdot \mathbf{x} - \phi_i) \cos(\mathbf{k} \cdot \mathbf{x} - \phi) \right\rangle_{\mathbf{x} \sim \mathcal{N}(0, \sigma^2 \mathbf{I})} \cdot \\
&= \tfrac{1}{2} \cos(\phi + \phi_i) e^{-\frac{1}{2\sigma^2}|\mathbf{k}+\mathbf{k}_i|^2} + \tfrac{1}{2} \cos(\phi - \phi_i) e^{-\frac{1}{2\sigma^2}|\mathbf{k}-\mathbf{k}_i|^2}.
\end{aligned} \tag{84}$$

When $|\mathbf{k}| = |\mathbf{k}_i| = 1$, the simple cell tuning curves $r_i = g(\mathbf{w}_i \cdot \mathbf{h})$ only depend on $\cos(\theta - \theta_i)$ and $\phi$, allowing a Fourier decomposition

$$r_i(\theta, \phi) = \sum_n a_n(\phi, \phi_i) \cos(n(\theta - \theta_i)). \tag{85}$$

The simple cell kernel $K_s$, therefore decomposes into Fourier modes over $\theta$

$$K_s(\theta, \theta', \phi, \phi') = \sum_n b_n(\phi, \phi') \cos(n(\theta - \theta')), \tag{86}$$

where $b_n(\phi, \phi') = \left\langle a_n(\phi, \phi_i) a_n(\phi', \phi_i) \right\rangle_{\phi_i}$. It therefore suffices to solve the infinite sequence of integral eigenvalue problems over $\phi$

$$\begin{aligned}
\tfrac{1}{2\pi} \int_0^{2\pi} b_n(\phi, \phi') v_{n,k}(\phi) d\phi &= \lambda_{n,k} v_{n,k}(\phi') \\
\implies K_s(\theta, \theta', \phi, \phi') &= \sum_{n,k} \lambda_{n,k} \cos(n(\theta - \theta')) v_{n,k}(\phi) v_{n,k}(\phi').
\end{aligned} \tag{87}$$

With this choice it is straightforward to verify that the kernel eigenfunctions are $v_{n,k}(\theta, \phi) = e^{in\theta} v_{n,k}(\phi)$ with corresponding eigenvalue $\lambda_{n,k}$. Since $b_n$ is not translation invariant in $\phi - \phi'$, the eigenfunctions $v_{n,k}$ are not necessarily Fourier modes. These eigenvalue problems for $b_n$ must be solved numerically when using arbitrary nonlinearity $g$. The top eigenfunctions of the simple cell kernel depend heavily on the phase of the two grating stimuli $\phi$. Thus, a pure orientation discrimination task which is independent of phase requires a large number of samples to learn with the simple cell population.

## Complex cell populations are phase invariant

V1 also contains complex cells which possess invariance to the phase $\phi$ of the stimulus.

$$\mathcal{F}(\mathbf{x}, \theta_i, \phi_i) = \tfrac{\sigma^2}{2\pi} e^{-\frac{\sigma^2}{2}|\mathbf{x}|^2} \cos(\mathbf{k}(\theta_i) \cdot \mathbf{x} - \phi_i), \tag{88}$$

Again using Gabor filters we model the complex cell responses with a quadratic nonlinearity and sum over two squared filters which are phase shifted by $\pi/2$

$$\begin{aligned}
r_i(\theta, \phi) &= (\mathcal{F}(\theta_i, \phi_i) \cdot \mathbf{h}(\theta, \phi))^2 + (\mathcal{F}(\theta_i, \phi_i - \pi/2) \cdot \mathbf{h}(\theta, \phi))^2 \\
&= \tfrac{1}{4} e^{-\frac{1}{\sigma^2}|\mathbf{k}+\mathbf{k}_i|^2} + \tfrac{1}{4} e^{-\frac{1}{\sigma^2}|\mathbf{k}-\mathbf{k}_i|^2} + \tfrac{1}{2} e^{-\sigma^{-2}} \cos(2\phi_i),
\end{aligned} \tag{89}$$

which we see is independent of the phase $\phi$ of the grating stimulus. Integrating over the set of possible Gabor filters $(\mathbf{k}_i, \phi_i)$ with $|\mathbf{k}| = 1$ again gives the following kernel for the complex cells

$$K_c(\theta) = \tfrac{1}{\cosh(2\beta)} \cosh(2\beta \cos(\theta)). \tag{90}$$

Remarkably, this kernel is independent of the phase $\phi$ of the grating stimulus. Thus, complex cell populations possess good inductive bias for vision tasks where the target function only depends on the orientation of the stimulus rather than it's phase. In reality, V1 is a mixture of simple and complex cells. Let $s \in [0, 1]$ represent the relative proportion of neurons which are simple cells and $(1 - s)$ the relative proportion of complex cells. The kernel for the mixed V1 population is given by a simple convex combination of the simple and complex cell kernels

$$
\begin{aligned}
K_{V1}(\theta, \theta', \phi, \phi') &= \frac{1}{N} \sum_{i=1}^{N} r_i(\theta, \phi) r_i(\theta', \phi') \rightarrow \langle r(\theta, \phi, c) r(\theta', \phi', n) \rangle_{n \sim p_{V1}(n)} \\
&= s \langle r(\theta, \phi, n) r(\theta', \phi', n) \rangle_{n \sim p_s(n)} + (1 - s) \langle r(\theta, \phi, n) r(\theta', \phi', n) \rangle_{n \sim p_c(n)} \\
&= s K_s(\theta, \theta', \phi, \phi') + (1 - s) K_c(\theta, \theta'),
\end{aligned}
\tag{91}
$$

where $n$ denotes neuron type (simple vs complex, tuning etc) and $p_{V1}(n), p_s(n), p_c(n)$ are probability distributions over the V1 neuron identities, the simple cell identities and the complex cell identities respectively. Increasing $s$ increases the phase dependence of the code by giving greater weight to the simple cell population. Decreasing $s$ gives weight to the complex cell population, encouraging phase invariance of readouts.

## Visualization of feedforward Gabor V1 model and induced kernels
Examples of the induced kernels for the Gabor-bank V1 model are provided in *Figure 5*. We show how choice of rectifying nonlinearity $g(z)$ and sparsifying threshold $a$ influence the kernel and their spectra. Learning curves for simple orientation tasks are provided.

## Gabor model spectral bias and fit to V1 data
Motivated by findings in the primary visual cortex (*Hansel and van Vreeswijk, 2002*; *Miller and Troyer, 2002*; *Priebe et al., 2004*; *Priebe and Ferster, 2008*), we studied the spectral bias induced by rectified power-law nonlinearities of the form $g(z) = \max\{0, z - a\}^q$. From theory, such a power-law activation function arises in a spiking neuron when firing is driven by input fluctuations (*Hansel and van Vreeswijk, 2002*; *Miller and Troyer, 2002*). Further, this activation is observed in intracellular recordings over the dynamic range of neurons in primary visual cortex (*Priebe and Ferster, 2008*). For example, in cats, the power, $q$, ranges from 2.7 to 3.9 (*Priebe et al., 2004*). We fit parameters of our model to the Mouse V1 kernel and compared to other parameter sets in *Figure 5—figure supplement 1*. Our best fit value of $q = 1.7$ is lower but on par with the estimates from the cat and reproduces the observed kernel. Computation of the kernel and its eigenvalues (Appendix Nonlinear simple cells) indicates a low frequency bias: the eigenvalues for low frequency modes are higher than those for high frequency modes, indicating a strong inductive bias to learn functions of low frequency in the orientation. Decreasing sparsity (lower $a$) leads to a faster decay in the spectrum (but similar asymptotic scaling at the tail, see *Figure 5—figure supplements 1 and 2*) and a stronger bias towards lower frequency functions (*Figure 5*). The effect of the power of nonlinearity $q$ is more nuanced: increasing power may increase spectra at lower frequencies, but may also lead to a faster decay at the tail (*Figure 5—figure supplements 1 and 2*). In general, an exponent $q$ implies a power-law asymptotic spectral decay $\lambda_k \sim k^{-2q-2}$ as $k \to \infty$ (Appendix Nonlinear simple cells). The behavior at low frequencies may have significant impact for learning with few samples. Overall, our findings show that the spectral bias of a population code can be determined in non-trivial ways by its biophysical parameters, including neural thresholds and nonlinearities.

## Energy model with partially phase-selective cells
The model of the V1 population as a mixture of purely simple and purely complex cells is an idealization which fails to capture the variability in phase selectivity of cells observed in experiment. In this section, we describe a simple model which can interpolate between an invariant code and a code which has high alignment with phase-dependent eigenfunctions. Further, a single scalar parameter $\alpha$ will control how strongly the population is biased towards invariance. We define $r_i(\theta, \phi) = g\left(z_i(\theta, \phi)\right)$ for nonlinear function $g$ and scalar $z$ which is constructed as follows

$$
\begin{aligned}
z_i(\theta, \phi) &= \beta_1 [\mathcal{F}(\theta_i, \phi_i) \cdot \mathbf{h}(\theta, \phi)]_+^2 + \beta_2 [\mathcal{F}(\theta_i, \phi_i + \pi/2) \cdot \mathbf{h}(\theta, \phi)]_+^2 \\
&\quad + \beta_3 [\mathcal{F}(\theta_i, \phi_i + \pi) \cdot \mathbf{h}(\theta, \phi)]_+^2 + \beta_4 [\mathcal{F}(\theta_i, \phi_i + 3\pi/2) \cdot \mathbf{h}(\theta, \phi)]_+^2.
\end{aligned}
\tag{92}
$$

This linear combination is inspired by the construction of simple cells in Dayan & Abbot Chapter 2 (*Dayan and Abbott, 2001*). If all $\beta$ are equal, then this tuning curve is invariant to phase $\phi$. To generate variability in selectivity to phase $\phi$, we will draw $\boldsymbol{\beta}$ from a Dirichlet distrbution on the simplex with concentration parameter $\alpha\mathbf{1}$ so that $p(\boldsymbol{\beta}) \propto \prod_{j=1}^{4} \beta_j^{\alpha-1}$ with $\sum_{j=1}^{4} \beta_j = 1$. In the $\alpha \to \infty$ limit, the probability density concentrates on $\frac{1}{4}\mathbf{1}$, leading to a code comprised entirely of complex cells which are invariant to phase $\phi$. In the $\alpha \to 0$ limit, the density is concentrated around the "edges" of the simplex such as $(1, 0, 0, 0), (0, 1, 0, 0)$, where only one preferred phase is present per neuron. For intermediate values, neurons are partially selective to phase. As before, the selectivity or invariance to phase is manifested in the kernel decomposition and leads to similar learning curves for the three tasks of the main paper (Orientation, Phase, Hybrid). We provide an illustration of tuning curves, F1/F0 distributions, eigenfunctions, and learning curves in *Figure 5—figure supplement 3*.

## Time dependent neural codes

### RNN model and decomposition

In this setting, the population code $\mathbf{r}(\{\boldsymbol{\theta}(t)\}, t)$ is a function of an input stimulus sequence $\boldsymbol{\theta}(t)$ and time $t$. In general the neural code $\mathbf{r}$ at time $t$ can depend on the entire history of the stimulus input $\boldsymbol{\theta}(t')$ for $t' \leq t$, as is the case for recurrent neural networks. We denote dependence of a function $f$ on $\boldsymbol{\theta}(t)$ in this causal manner with the notation $f(\{\boldsymbol{\theta}\}, t)$. In a learning task, a set of readout weights $\mathbf{w}$ are chosen so that a downstream linear readout $f(\{\boldsymbol{\theta}\}, t) = \mathbf{w} \cdot \mathbf{r}(\{\boldsymbol{\theta}\}, t)$ approximates a target sequence $y(\{\boldsymbol{\theta}\}, t)$ which maps input stimulus sequences to output scalar sequences. The quantity of interest is the generalization $E_g$, which in this case is an average over both input sequences and time, $E_g = \left\langle (y(\{\boldsymbol{\theta}\}, t) - f(\{\boldsymbol{\theta}\}, t))^2 \right\rangle_{\boldsymbol{\theta}(t), t}$. The average is computed over a distribution of input stimulus sequences $p(\boldsymbol{\theta}(t))$. To train the readout, $\mathbf{w}$, the network is given a sample of $P$ stimulus sequences $\boldsymbol{\theta}^\mu(t), \mu = 1, ..., P$. For the μ-th training input sequence, the target system $y$ is evaluated at a set of discrete time points $\mathcal{T}_\mu = \{t_1, t_2, ..., t_{|\mathcal{T}_\mu|}\}$ giving a collection of target values $\{y_t^\mu\}_{t \in \mathcal{T}_\mu}$ and a total dataset of size $\mathcal{P} = \sum_{\mu=1}^{P} |\mathcal{T}_\mu|$. The *average case generalization* computes a further average of the generalization error $E_g$ over randomly sampled datasets of size $\mathcal{P}$.

Learning is again achieved through iterated weight updates with delta-rule form, but now have contributions from both sequence index and time $\Delta\mathbf{w} = \eta \sum_\mu \sum_{t \in \mathcal{T}_\mu} \mathbf{r}_t^\mu (y_t^\mu - f_t^\mu)$. As before, optimization of the readout weights is equivalent to kernel regression with a kernel that computes inner products of neural population vectors at different times $t, t'$ for different input sequences $\{\boldsymbol{\theta}\}, \{\boldsymbol{\theta}'\}$. This kernel depends on details of the time varying population code including its recurrent intrinsic dynamics as well as its encoding of the time-varying input stimuli. The optimization problem and delta rule described above converge to the kernel regression solution for kernel gram matrix $K_{t,t'}^{\mu,\mu'} = \frac{1}{N}\mathbf{r}_t^\mu \cdot \mathbf{r}_{t'}^{\mu'}$ (*Dong et al., 2020*; *Yang, 2019*; *Yang, 2020*). The learned function has the form $f(\{\boldsymbol{\theta}\}, t) = \sum_{\mu, t' \in \mathcal{T}_\mu} \alpha_t^\mu K(\{\boldsymbol{\theta}\}, \{\boldsymbol{\theta}\}^\mu, t, t')$, where $\boldsymbol{\alpha} = \mathbf{K}^+\mathbf{y}$ for kernel gram matrix $\mathbf{K} \in \mathbb{R}^{\mathcal{P} \times \mathcal{P}}$ which is computed for the entire set of training sequences, and the vector $\mathbf{y} \in \mathbb{R}^\mathcal{P}$ is the vector containing the desired target outputs for each sequence. Assuming a probability distribution over sequences $\boldsymbol{\theta}(t)$, the kernel can be diagonalized with orthonormal eigenfunctions $\psi_k(\{\boldsymbol{\theta}\}, t)$. Our theory carries over from the static case: kernels whose top eigenfunctions have high alignment with the target dynamical system $y(\{\boldsymbol{\theta}\}, t)$ will achieve the best average case generalization performance.

## Alternative neural codes with same kernel

### Orthogonal transformations are sufficient for linear kernel-preserving transformations

We will now show that for any linear transformation $\tilde{\mathbf{r}} = \mathbf{A}\mathbf{r}$ which preserves the inner product kernel $K(\boldsymbol{\theta}, \boldsymbol{\theta}')$, there exists an orthogonal matrix $\mathbf{Q}$ such that $\tilde{\mathbf{r}} = \mathbf{Q}\mathbf{r}$.

*Proof.*

Let $\tilde{\mathbf{r}}(\boldsymbol{\theta}) = \mathbf{A}\mathbf{r}(\boldsymbol{\theta})$ for all stimuli $\boldsymbol{\theta}$. To preserve the kernel, we must have

$$K(\boldsymbol{\theta}, \boldsymbol{\theta}') = \tilde{\mathbf{r}}(\boldsymbol{\theta}) \cdot \tilde{\mathbf{r}}(\boldsymbol{\theta}') = \mathbf{r}(\boldsymbol{\theta}) \cdot \mathbf{r}(\boldsymbol{\theta}') \implies \mathbf{r}(\boldsymbol{\theta})\mathbf{A}^\top \mathbf{A}\mathbf{r}(\boldsymbol{\theta}') = \mathbf{r}(\boldsymbol{\theta}) \cdot \mathbf{r}(\boldsymbol{\theta}'). \tag{93}$$

Taking projections against each of the orthonormal eigenfunctions $\psi_\ell(\boldsymbol{\theta})$ (see Appendix Singular value decomposition of continuous population responses), we define vectors $\mathbf{u}_k$ as $\sqrt{\lambda_k}\mathbf{u}_k = \left\langle \mathbf{r}(\boldsymbol{\theta})\psi_k(\boldsymbol{\theta}) \right\rangle_{\boldsymbol{\theta}}$, allowing us to express the SVD of the population code $\mathbf{r}(\boldsymbol{\theta}) = \sum_k \sqrt{\lambda_k}\mathbf{u}_k\psi_k(\boldsymbol{\theta})$.

These vectors $\{\mathbf{u}_k\}$ are orthonormal $\mathbf{u}_k \cdot \mathbf{u}_\ell = \delta_{k\ell}$ since, by the definition of the kernel eigenfunctions $\psi_k$,

$$\sqrt{\lambda_k \lambda_\ell} \mathbf{u}_k \cdot \mathbf{u}_\ell = \langle \mathbf{r}(\boldsymbol{\theta}) \cdot \mathbf{r}(\boldsymbol{\theta}') \psi_k(\boldsymbol{\theta}) \psi_\ell(\boldsymbol{\theta}') \rangle_{\boldsymbol{\theta}, \boldsymbol{\theta}'} = \langle \psi_k(\boldsymbol{\theta}) \langle K(\boldsymbol{\theta}, \boldsymbol{\theta}') \psi_\ell(\boldsymbol{\theta}') \rangle_{\boldsymbol{\theta}'} \rangle_{\boldsymbol{\theta}}$$
$$= \lambda_\ell \langle \psi_k(\boldsymbol{\theta}) \psi_\ell(\boldsymbol{\theta}) \rangle_{\boldsymbol{\theta}} = \lambda_k \delta_{k,\ell}. \tag{94}$$

Since $\mathbf{r}(\boldsymbol{\theta})$ and $\tilde{\mathbf{r}}(\boldsymbol{\theta})$ have the same inner product kernel, they must posess the same kernel eigenfunctions $\psi_k$ and kernel eigenvalues $\lambda_k$, which are identified through the eigenvalue problem

$$\int p(\boldsymbol{\theta}) K(\boldsymbol{\theta}, \boldsymbol{\theta}') \psi_k(\boldsymbol{\theta}) d\boldsymbol{\theta} = \lambda_k \psi_k(\boldsymbol{\theta}). \tag{95}$$

We therefore have the following two singular value decompositions for $\mathbf{r}$ and $\tilde{\mathbf{r}}$

$$\mathbf{r}(\boldsymbol{\theta}) = \sum_{k=1}^{N} \sqrt{\lambda_k} \mathbf{u}_k \psi_k(\boldsymbol{\theta}) \, , \, \tilde{\mathbf{r}}(\boldsymbol{\theta}) = \sum_{k=1}^{N} \sqrt{\lambda_k} \tilde{\mathbf{u}}_k \psi_k(\boldsymbol{\theta}). \tag{96}$$

where $\{\mathbf{u}_k\}_{k=1}^{N}$ and $\{\tilde{\mathbf{u}}_k\}_{k=1}^{N}$ are both complete sets of orthonormal vectors (the sums above run over possible zero eigenvalues). Taking the equation $\tilde{\mathbf{r}}(\boldsymbol{\theta}) = \mathbf{A}\mathbf{r}(\boldsymbol{\theta})$, we multiply both sides of the equation by $\psi_k(\boldsymbol{\theta})$ and average over $\boldsymbol{\theta}$ giving

$$\langle \tilde{\mathbf{r}}(\boldsymbol{\theta}) \psi_k(\boldsymbol{\theta}) \rangle = \sqrt{\lambda_k} \tilde{\mathbf{u}}_k = \mathbf{A} \langle \mathbf{r}(\boldsymbol{\theta}) \psi_k(\boldsymbol{\theta}) \rangle_{\boldsymbol{\theta}} = \sqrt{\lambda_k} \mathbf{A} \mathbf{u}_k \tag{97}$$

For an eigenmode $k$ with positive eigenvalue $\lambda_k > 0$, this implies $\tilde{\mathbf{u}}_k = \mathbf{A}\mathbf{u}_k$, while there is no corresponding constraint for the null modes with $\lambda_k = 0$. However, the action of $\mathbf{A}$ on the nullspace of the code has no influence on $\tilde{\mathbf{r}}$ so there is no loss in generality to restrict consideration to transformations $\mathbf{A}$ which satisfy $\tilde{\mathbf{u}}_k = \mathbf{A}\mathbf{u}_k$ for all $k \in [N]$ (rather than just the $\lambda_k > 0$ modes). This choice gives $\mathbf{A} = \sum_{k=1}^{N} \tilde{\mathbf{u}}_k \mathbf{u}_k^{\top}$. Thus, the space of codes $\tilde{\mathbf{r}}(\boldsymbol{\theta})$ with equivalent kernels to $\mathbf{r}(\boldsymbol{\theta}) \cdot \mathbf{r}(\boldsymbol{\theta}')$ generated through linear transformations is equivalent to all possible orthogonal transformations of the original code $\{\mathbf{Q}\mathbf{r}(\boldsymbol{\theta}) : \mathbf{Q}\mathbf{Q}^{\top} = \mathbf{Q}^{\top}\mathbf{Q} = \mathbf{I}\}$. ∎

## Effect of noise on RROS symmetry

The random rotation and optimal shift (RROS) operations introduced in the main text preserve generalization performance under the assumption of a deterministic neural code. However, for noisy codes, the presence of RROS symmetry is dependent on the noise distribution. Below we discuss two commonly analyzed distributions: the Gaussian distribution and the Poisson distribution. For Gaussian noise, the RROS operations preserve the generalization performance and the local Fisher information. However, if noise is constrained to be Poisson then RROS operations do not preserve generalization or Fisher information.

First, we will analyze stimulus dependent Gaussian noise, where generalization performance is preserved under rotations and baseline shifts. Note that if the code at $\theta$ obeyed $\mathbf{r}(\theta) \sim \mathcal{N}(\bar{\mathbf{r}}(\theta), \boldsymbol{\Sigma}_n(\theta))$, then the rotated and shifted code follows $\mathbf{Q}\mathbf{r}(\theta) + \delta \sim \mathcal{N}(\mathbf{Q}\bar{\mathbf{r}}(\theta) + \delta, \mathbf{Q}\boldsymbol{\Sigma}_n(\theta)\mathbf{Q}^{\top})$. This rotated and shifted code $\mathbf{Q}\mathbf{r}(\theta) + \delta$, when centered, will exhibit identical generalization performance as the original code. This is true both for learning from a trial averaged or non-trial averaged code. In the case of Gaussian noise on a centered code, the dataset transforms under a rotation as $\mathcal{D} = \{\mathbf{r}_\mu, y_\mu\} \rightarrow \mathcal{D}' = \{\mathbf{Q}\mathbf{r}_\mu, y_\mu\}$. The optimal weights for a linear model similarly transform as $\mathbf{w}(\mathcal{D}) \rightarrow \mathbf{Q}\mathbf{w}(\mathcal{D})$. Under these transformations the predictor on test point $\theta$ is unchanged since

$$f(\theta) = \mathbf{w} \cdot \mathbf{r}(\theta) \rightarrow \mathbf{w}^{\top} \mathbf{Q}^{\top} \mathbf{Q}\mathbf{r}(\theta) = \mathbf{w} \cdot \mathbf{r}(\theta) \tag{98}$$

Further, the local Fisher information matrix is $\mathbf{I}(\theta) = \frac{\partial \bar{\mathbf{r}}(\theta)^{\top}}{\partial \theta} \boldsymbol{\Sigma}_n^{-1}(\theta) \frac{\partial \bar{\mathbf{r}}(\theta)}{\partial \theta^{\top}} + \frac{1}{2} \text{Tr} \boldsymbol{\Sigma}_n^{-1}(\theta) \frac{\partial \boldsymbol{\Sigma}_n(\theta)}{\partial \theta} \boldsymbol{\Sigma}_n^{-1}(\theta) \frac{\partial \boldsymbol{\Sigma}_n(\theta)}{\partial \theta^{\top}}$ is unchanged under the transformation $\mathbf{r} \rightarrow \mathbf{Q}\mathbf{r} + \delta$. Under this transformation, the covariance simply transforms linearly $\boldsymbol{\Sigma}_n \rightarrow \mathbf{Q}\boldsymbol{\Sigma}_n(\theta)\mathbf{Q}^{\top}$ and the $\mathbf{Q}$ matrices will annihilate under the trace. This shows that, for some noise models, our assumption that rotations and baseline shifts preserve generalization performance will be valid.

However, for Poisson noise, where the variance is tied to the mean firing rate, the RROS operations will not preserve noise structure or information content. The Fisher information at scalar stimulus $\theta$ for a Poisson neuron is $I(\theta) = \frac{\bar{r}'(\theta)^2}{\bar{r}(\theta)}$. A baseline shift $r \rightarrow r + \delta$ to the tuning curve will not change

the numerator since the derivative of the tuning curve is invariant to this transformation, but it will increase the denominator.

## Necessary conditions for optimally sparse codes

Next we argue why optimally sparse codes should be lifetime and population selective. We consider the following optimization problem: find a non-negative neural responses $\mathbf{S} \in \mathbb{R}^{N \times P}$ and baseline vector $\boldsymbol{\delta} \in \mathbb{R}^N$ so that baseline subtracted responses $\mathbf{R} = \mathbf{S} - \boldsymbol{\delta}\mathbf{1}^\top$ realize a desired inner product kernel $\mathbf{K} \in \mathbb{R}^{P \times P}$ and have minimal total firing. This is equivalent to finding the most metabolically efficient code among the space of codes with equivalent inductive bias. Mathematically, we formulate this problem as

$$\min_{\mathbf{S} \in \mathbb{R}^{N \times P}, \boldsymbol{\delta} \in \mathbb{R}^N} \sum_{i\mu} S_{i\mu}, \quad \text{s.t.} \ \left(\mathbf{S} - \boldsymbol{\delta}\mathbf{1}^\top\right)^\top \left(\mathbf{S} - \boldsymbol{\delta}\mathbf{1}^\top\right) = \mathbf{K}, \quad S_{i\mu} \geq 0 \ \forall i \in [N], \mu \in [P]. \quad (99)$$

To enforce the constraints for the definition of the kernel and the non-negativity of the responses, we introduce the following Lagrangian

$$\mathcal{L}(\mathbf{S}, \boldsymbol{\delta}, \mathbf{A}, \mathbf{V}) = \mathbf{1}^\top \mathbf{S} \mathbf{1} - \text{Tr}\left(\left[(\mathbf{S} - \boldsymbol{\delta}\mathbf{1}^\top)^\top(\mathbf{S} - \boldsymbol{\delta}\mathbf{1}^\top) - \mathbf{K}\right]\mathbf{A}\right) - \text{Tr}\mathbf{V}^\top \mathbf{S} \quad (100)$$

where 1 is the vector containing all ones, the Lagrange multiplier matrix $\mathbf{A}$ enforces the definition of the kernel and the KKT multiplier matrix $\mathbf{V}$ enforces the non-negativity constraints for each element of $\mathbf{S}$. The KKT conditions require that any local optimum of the objective would have to satisfy the following equations (*Kuhn and Tucker, 2014*)

$$\begin{aligned}
\frac{\partial \mathcal{L}}{\partial \mathbf{S}} &= \mathbf{1}\mathbf{1}^\top - (\mathbf{S} - \boldsymbol{\delta}\mathbf{1}^\top)\mathbf{A} - \mathbf{V} = \mathbf{0} \\
\frac{\partial \mathcal{L}}{\partial \boldsymbol{\delta}} &= -(\mathbf{S} - \boldsymbol{\delta}\mathbf{1}^\top)\mathbf{A}\mathbf{1} = \mathbf{0} \\
\frac{\partial \mathcal{L}}{\partial \mathbf{A}} &= (\mathbf{S} - \boldsymbol{\delta}\mathbf{1}^\top)^\top(\mathbf{S} - \boldsymbol{\delta}\mathbf{1}^\top) - \mathbf{K} = \mathbf{0} \\
\mathbf{V} \odot \mathbf{S} &= \mathbf{0},
\end{aligned} \quad (101)$$

where $\odot$ denotes the element-wise Hadamard product. Using the complementary slackness condition $\mathbf{S} \odot \mathbf{V} = \mathbf{0}$, and the first optimality condition $\frac{\partial \mathcal{L}}{\partial \mathbf{S}} = \mathbf{0}$, we have

$$\mathbf{S} = \mathbf{S} \odot (\mathbf{S} - \boldsymbol{\delta}\mathbf{1}^\top)\mathbf{A} \quad (102)$$

Therefore, for any neuron-stimulus pair $(i, \mu)$, either $S_{i\mu} = 0$ or $\sum_{\nu \in [P]}(S_{i\nu} - \delta_i)A_{\nu\mu} = 1$. Further, under the condition that K is full rank, we conclude that for any stimulus µ, $\sum_{\nu \in [P]} A_{\mu\nu} = 0$ from the equation $\frac{\partial \mathcal{L}}{\partial \boldsymbol{\delta}} = \mathbf{0}$. Let $\mathcal{I}_i = \{\mu \in [P] : S_{i\mu} > 0\}$ represent the set of stimuli for which neuron $i$ fires. We will call this the *receptive field set* for neuron $i$. Let $\mathbf{B}_{(i)} \in \mathbb{R}^{P \times P}$ have entries

$$[\mathbf{B}_{(i)}]_{\mu\nu} = \begin{cases} [\mathbf{A}_{(i)}^+]_{\mu\nu} & \mu, \nu \in \mathcal{I}_i \\ 0 & \mu \notin \mathcal{I}_i \text{ or } \nu \notin \mathcal{I}_i \end{cases} \quad (103)$$

where the matrix $\mathbf{A}_{(i)}$ is the $|\mathcal{I}_i| \times |\mathcal{I}_i|$ minor of $\mathbf{A}$ obtained by taking all rows and columns with indices $\mu, \nu \in \mathcal{I}_i$, and $\mathbf{A}^+$ denotes pseudo-inverse of $\mathbf{A}$. Then the $i$-th neuron's tuning curve is a function of the index set $\mathcal{I}_i$ the baseline $\delta_i$ and the neuron-independent $P \times P$ matrix $\mathbf{A}$. The non-negativity constraint for neuron $i$'s tuning curve implies that $S_{i\mu} = \sum_{\nu \in \mathcal{I}_i} B_{(i),\mu\nu}[\delta_i \sum_{\gamma \in [P]} A_{\nu\gamma} + 1] > 0$ for all $\mu \in \mathcal{I}_i$. To satisfy the definition of the kernel, we have the following constraint on the matrix $\mathbf{A}$, the index sets $\mathcal{I}_i$ and baselines $\delta_i$

$$\mathbf{K} = \sum_{i=1}^N (\mathbf{s}(\mathcal{I}_i, \delta_i, \mathbf{A}) - \delta_i\mathbf{1})(\mathbf{s}(\mathcal{I}_i, \delta_i, \mathbf{A}) - \delta_i\mathbf{1})^\top \quad (104)$$

This equation implicitly defines the index sets $\mathcal{I}_i$ the baselines $\delta_i$ and the KKT matrix $\mathbf{A}$. We see that, in order to fit an arbitrary kernel, the receptive field sets $\{\mathcal{I}_i\}$ and baselines $\delta_i$ for each neuron must be sufficiently diverse since otherwise only a low rank kernel matrix can be achieved from the optimally sparse code. As a concrete example, suppose that $\mathcal{I}_i = \mathcal{I}$ so that $\mathbf{B}_{(i)} = \mathbf{B}$ and $\delta_i = \delta$ for all $i$. For example, this could occur if each neuron fired for every possible stimulus. In this case, the kernel would be rank one: $\mathbf{K} = N(\mathbf{s}(\mathcal{I}, \delta, \mathbf{A}) - \delta\mathbf{1})(\mathbf{s}(\mathcal{I}, \delta, \mathbf{A}) - \delta\mathbf{1})^\top$. In order to achieve a higher rank

code there must be sufficient diversity of the receptive fields $\mathcal{I}_i$. Thus the only way for optimally sparse codes to realize high rank kernels $\mathbf{K}$ is to have neurons to have different receptive field sets $\mathcal{I}_i$. The necessary optimality conditions thus reveal a preference for sparse neural tuning curves to have high *lifetime sparseness*; to achieve diverse index sets $\mathcal{I}_i$, any given neuron will fire only for a unique subset of the possible stimuli.

## Impact of neural noise and unlearnable targets on learning

While our analysis so far has focused on deterministic population codes, our theory can be extended to neural populations which exhibit variability in responses to identical stimuli. For each stimulus $\boldsymbol{\theta}$, we let the population response $\mathbf{r}(\boldsymbol{\theta})$ be a random vector with mean $\bar{\mathbf{r}}(\boldsymbol{\theta}) = \langle \mathbf{r}(\boldsymbol{\theta}) \rangle_{\mathbf{r}|\boldsymbol{\theta}}$ and covariance $\boldsymbol{\Sigma}_n(\boldsymbol{\theta}) = \left\langle (\mathbf{r}(\boldsymbol{\theta}) - \bar{\mathbf{r}}(\boldsymbol{\theta}))(\mathbf{r}(\boldsymbol{\theta}) - \bar{\mathbf{r}}(\boldsymbol{\theta}))^\top \right\rangle_{\mathbf{r}|\boldsymbol{\theta}}$.

The (deterministic) target function can be decomposed in terms of the mean response as $y(\boldsymbol{\theta}) = \mathbf{w}^* \cdot \bar{\mathbf{r}}(\boldsymbol{\theta})$ (the usual decomposition $y = \mathbf{w}^* \cdot \mathbf{r}(\boldsymbol{\theta})$ gives an unphysical target function which fluctuates with the variability in neural responses). For a given configuration of weights $\mathbf{w}$, the generalization error (which is an average over the joint distribution of $\mathbf{r}, \boldsymbol{\theta}$) is determined only by the signal $\boldsymbol{\Sigma}_s = \left\langle \bar{\mathbf{r}}(\boldsymbol{\theta})\bar{\mathbf{r}}(\boldsymbol{\theta})^\top \right\rangle_{\boldsymbol{\theta}}$ and noise $\boldsymbol{\Sigma}_n = \langle \boldsymbol{\Sigma}_n(\boldsymbol{\theta}) \rangle_{\boldsymbol{\theta}}$ correlation matrices:

$$
\begin{aligned}
E_g(\mathbf{w}) &= \left\langle \left( \mathbf{r}(\boldsymbol{\theta}) \cdot \mathbf{w} - \bar{\mathbf{r}}(\boldsymbol{\theta}) \cdot \mathbf{w}^* \right)^2 \right\rangle_{\mathbf{r},\boldsymbol{\theta}} = \left\langle \left[ (\mathbf{w} - \mathbf{w}^*) \cdot \bar{\mathbf{r}}(\boldsymbol{\theta}) + \mathbf{w} \cdot (\mathbf{r}(\boldsymbol{\theta}) - \bar{\mathbf{r}}(\boldsymbol{\theta})) \right]^2 \right\rangle \\
&= (\mathbf{w} - \mathbf{w}^*)^\top \left\langle \bar{\mathbf{r}}(\boldsymbol{\theta})\bar{\mathbf{r}}(\boldsymbol{\theta})^\top \right\rangle (\mathbf{w} - \mathbf{w}^*) + \mathbf{w}^\top \left\langle (\mathbf{r}(\boldsymbol{\theta}) - \bar{\mathbf{r}}(\boldsymbol{\theta}))(\mathbf{r}(\boldsymbol{\theta}) - \bar{\mathbf{r}}(\boldsymbol{\theta}))^\top \right\rangle \mathbf{w} \\
&= (\mathbf{w} - \mathbf{w}^*)^\top \boldsymbol{\Sigma}_s (\mathbf{w} - \mathbf{w}^*) + \mathbf{w}^\top \boldsymbol{\Sigma}_n \mathbf{w}
\end{aligned}
\tag{105}
$$

where we utilized the fact that $\langle \mathbf{r}(\boldsymbol{\theta}) - \bar{\mathbf{r}}(\boldsymbol{\theta}) \rangle_{\mathbf{r}|\boldsymbol{\theta}} = 0$ to eliminate the cross-term. The two terms in the final expression can be thought of as a bias-variance decomposition over the noise in neural responses. The minimum achievable loss can be obtained by differentiation of the generalization error expression with respect to $\mathbf{w}$, giving $E_g^* = \mathbf{w}^* \boldsymbol{\Sigma}_n (\boldsymbol{\Sigma}_s + \boldsymbol{\Sigma}_n)^{-1} \boldsymbol{\Sigma}_s \mathbf{w}^*$. We note that any noise correlation matrix with noise orthogonal to coding direction $\boldsymbol{\Sigma}_n \mathbf{w}^* = 0$ will give the minimal (zero) asymptotic error. Alignment of the noise $\boldsymbol{\Sigma}_n$ with $\mathbf{w}^*$ gives higher asymptotic error.

In addition to the irreducible error, the presence of neural noise can alter the learning curve at finite $P$. An analytical study of this is difficult, which we leave for future work. We numerically study the effect of neural variability on generalization performance in the orientation discrimination tasks for non-trial-averaged Mouse V1 code in *Appendix 1—figure 1* . We note that the generalization error is worse at each finite value of $P$ when compared to trial averaged (noise free) learning curves. We varied the regularization parameter and did not find an obvious non-zero optimal weight decay $\lambda$, consistent with small noise levels.

Neural noise is not the only phenomenon that can degrade task learning. Codes which are incapable of expressing the target function through linear readouts are also susceptible to overfitting. As explained in *Canatar et al., 2021*, the components of the target function that are inexpressible act as a source of noise on the learning process which can overfit this noise. Such a scenario can occur, for example, when the readout neuron only gets input from a sparse subset of the coding neural population (*Seeman et al., 2018*). We show in *Appendix 1—figure 1C-D* that using subsampled populations of size $N$ can indeed lead to a regime where more data can hurt performance leading to an overfitting error peak, a subsequent non-vanishing asymptotic error, and an optimal weight decay parameter $\lambda$. This phenomenon is known as double descent in machine learning literature (*Belkin et al., 2019*; *Mei and Montanari, 2020*; *Canatar et al., 2021*). At small $N$, these codes are not sufficiently expressive to learn the target function through linear readout. The overfitting peak occurs near the interpolation threshold, the largest value of $P$ where all training sets could be perfectly fit in the $\lambda \to 0$ limit (*Canatar et al., 2021*). At infinite $P$, generalization error asymptotes to the amount of unexplained variance in the target function.

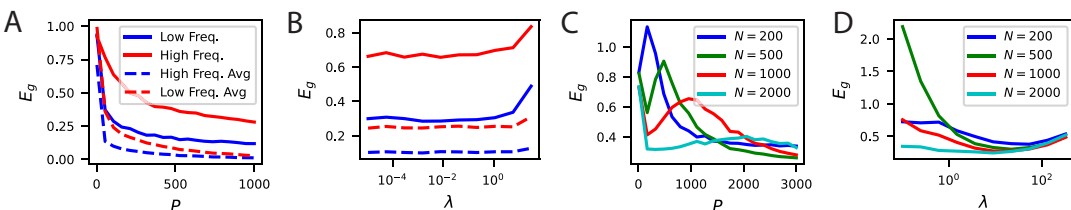

**Appendix 1—figure 1.** Neural noise and subsampled neural codes can lead to overfitting. (A) The learning curves without trial averaging (solid) and with trial averaging (dashed) for the high and low frequency orientation discrimination task. In principle, neural noise could limit asymptotic performance and lead to the existence of an optimal weight decay parameter $\lambda$. (B) Performance at $P = 500$ vs ridge $\lambda$ shows that there is not an optimal weight decay parameter. (C) Generalization of readouts trained on subsets of $N$ V1 neurons exhibit non-monotonic learning curves with an overfitting peak around $P \approx N$. (D) The performance of subsamples of $N$ neurons as a function of the weight decay parameter $\lambda$ at $P = 500$ samples show that, for sufficiently small $N$, there is a non-zero optimal $\lambda$.

