## [Editor Report]

This important study presents a theory of generalization in neural population codes and proposes sample efficiency as a new normative principle. The theory can be used to identify the set of 'easily learnable' stimulus-response mappings from neural data and makes strong behavioral predictions that can be evaluated experimentally. Overall, the new method for elucidating inductive biases of the brain is highly compelling and will be of interest to theoretical and experimental neuroscientists working towards understanding how the cortex works.

---

## [Decision Letter]

**Decision letter after peer review:**

Thank you for submitting your article "Population codes enable learning from few examples by shaping inductive bias" for consideration by *eLife*. Your article has been reviewed by 3 peer reviewers, and the evaluation has been overseen by a Reviewing Editor and Michael Frank as the Senior Editor. The following individuals involved in review of your submission have agreed to reveal their identity: Fabio Anselmi (Reviewer #1); Jeff Beck (Reviewer #2).

Essential revisions:

The consensus among all 3 reviewers is that the manuscript describes an insightful and novel mathematical framework for evaluating the ability of a downstream linear decoder to learn new stimulus-response mappings from relatively few training examples. However, there is also broad consensus among the 3 reviewers that (1) the manuscript is too technical and written in a way that may not be palatable to the *eLife* readership (to paraphrase one of the reviewers "I don't think that many experimentalists will immediately see the sheer utility of having a tool like this in their arsenal" and that 2) some of the assumptions made are overly simplistic and the theory needs to be extended with more realistic neural assumptions regarding noise, readout and kernels.

Wrt (1), before the manuscript can be accepted for publication, it is necessary that the authors expand on possible applications to neural and behavioral experiments. The authors also need to distill the math better, provide more intuitive explanations, and reduce the jargon. Wrt (2) R2 provided a detailed lists of comments with suggestions for more realistic assumptions.

*Reviewer #1 (Recommendations for the authors):*

The authors provide a simple and clear way to understand an aspect of the implicit bias of a neural population code linking it with well-known machine learning methods and concepts such as kernel regression, sample complexity and efficiency.

Although the mathematical results the authors employ are not novel, the way they apply them to the problem of neural coding is novel and interesting to a broad audience.

In particular, the computational neuroscience community can benefit from this work being it is one of the few dealing with the impact of the model implicit bias in explaining real data.

*Reviewer #2 (Recommendations for the authors):*

It is my opinion that the principle utility of this approach lies in its ability to identify the set of 'easily learnable' stimulus-response mappings from neural data which makes strong behavioral predictions that can be easily evaluated. I envision a simple experiment in which empirically obtained kernel functions are used to rank stimulus-response mappings according to their learnability which can then be plotted against measures of performance like the observed learning rate and saturated performance. Because kernel functions are empirically obtained, there is even the potential for meaningful cross-species comparisons. If behaviorally validated, one could also use this approach to label cortical populations by the set of easily learned stimulus-response mappings for that population. This allows for the identification of task-relevant neurons or regions which can be subsequently manipulated to enhance or degrade learning rates.

Of course, any theoretical approach is only as good as the underlying assumptions and so while the primary strength is the simplicity and generality of this approach, the primary weakness is its neglect of some very real and very relevant aspects of neural data in particular and statistical learning in general. In particular, the three principle limitations of this work are tied to its reliance on the assumptions that (1) neurons are noiseless, (2) decoders are linear, and (3) learned weights are unbiased.

(1) Within this framework, a realistic stimulus-dependent noise model can be easily introduced and its effects on the kernel and set of easily learned stimulus-response mappings investigated. So while the kernel would be substantially altered via the addition of a realistic noise model, the applications of the approach outlined above would not be affected. The same cannot be said for the efficient coding application described in this manuscript. There, the authors note that rotations and constant shifts of neural activity do not affect the kernel and thus do not affect the generalization error. This kernel invariance is not present when a non-trivial (i.e. non-isotropic) noise model is added. For example, suppose that neurons are independent and Poisson so that noise scales with the mean of the neural response. In this case, adding a baseline firing rate to a population of unimodal neurons representing orientation necessarily reduces the information content of the population while rotations can affect the fidelity with which certain stimulus values are represented. It is important to note, however, that while this particular efficiency result is not compelling, I believe that it is possible to perform a similar analysis that takes into account realistic noise models and focuses on a broad set of 'biologically plausible' kernels instead of particular invariant ones. For example, one could consider noise covariance structures with differential correlations (Moreno-Bote 2014). Since the magnitude of differential correlations controls the redundancy of the population code this would enable an analysis of the role of redundancy in suppressing (or enhancing) generalization error.

(2) Similarly, the linearity assumption is somewhat restrictive. Global linear decoders of neural activity are known to be highly inefficient and completely fail when decoding orientation in the primary visual cortex in the presence of contrast fluctuations. This is because contrast modulates the amplitude of the neural response and doubling the amplitude means doubling an estimate obtained from a linear decoder even when the underlying orientation has not changed. While the contrast issue could be partially addressed by simply considering normalized neural responses, it is not yet clear how to extend this approach to account for other sources of neural variability and co-variability that cause global linear decoders to fail so badly.

(3) This analysis relies on the assumption that decoder weights learned in the presence of finite data are efficient and unbiased. This assumption is problematic particularly when it comes to inductive bias and generalization error. This is because a standard way to reduce generalization error is to introduce bias into the learned decoder weights through a penalization scheme that privileges decoder weights with small magnitudes. This kind of regularization is particularly important when neurons are noisy. Fortunately, this issue could be addressed by parameterizing changes in the kernel function by the degree and type of regularization potentially leading to a more general result.

Finally, I would like to conclude by explicitly stating that while the limitations imposed by the assumptions listed above temper my enthusiasm in regards to conclusions drawn in this work, I do not believe there is some fundamental problem with the general theoretical framework. Indeed, items 1 and 3 above can be easily addressed through straightforward extensions of the authors approach and I look forward to their implementation. Item 2 is a bit more troublesome, but my intuition tells me that an information-theoretic extension based upon Fisher information may be capable of eliminating all three of these limiting assumptions by exploiting the relationship between FI(\theta) and FI(y=f(\theta)).

Ultimately, all I felt the need to say is in the public part of the review. But I wanted to use this space to insure that it was clear that I feel that this approach has a lot of potential and that I very much hope to see it extended/generalized.

If I were to make any suggestions they would be:

(1) a discussion of the consequences of biologically plausible noise models. I believe that this only requires simply augmenting K by a stimulus dependent noise covariance matrix.

(2) using this addition of a noise model to enable a discussion of the role of redundancy in generalization. This could be accomplished by considering perturbations to the noise covariance matrix that introduce differential correlations (Moreno-Bote 2014) of varying magnitude. More differential correlations means more redundancy and, I suspect, better generalization.

(3) the addition of a paragraph or two outlining some of the other ways that this learnability/generalization measure could be of use to physiologists and behavioral scientists. Training a mouse to do anything more complicated simple orientation detection is challenging to say the least. So having any tool that can be used to identify functions $y=f(\theta)$ that a mouse has a good chance of learning quickly would be highly advantageous. Similarly, by strategically subsampling neurons by tuning properties I suspect it may be possible to identify subpopulations of neurons that are particular important for learning certain functions. This is also cool because it allows physiologists to then target those populations for manipulation.

Of these (3) is probably the most important for this manuscript, because I don't think that many experimentalists will immediately see the sheer utility of having a tool like this in their arsenal.

*Reviewer #3 (Recommendations for the authors):*

The manuscript presents a theory of generalization performance in deterministic population codes, that applies to the case of small numbers of training examples. The main technical result, as far as I understand, is that generalization performance (the expected classification or regression error) of a population code depends exclusively on the 'kernel', i.e. a measure of the pairwise similarity between population activity patterns corresponding to different inputs. The main conceptual results are that, using this theory, one can understand the inductive biases of the code just from analyzing the kernel, particularly the top eigenfunctions; and that sample-efficient learning (low generalization performance with few samples) depends on whether the task is aligned with the population's inductive bias, that is, whether the target function (i.e. the true map from inputs to outputs) is aligned with the top eigenfunctions of the kernel. For instance, in mouse V1 data, they show that the top eigenfunctions correspond to low frequency functions of visual orientation (i.e. functions that map a broad range of similar orientations to similar output value), and that consistent with the theory, the generalization performance for small sample sizes is better for tasks defined by low frequency target functions. In my opinion, perhaps the most significant finding from a neuroscience perspective, is that the conditions for good generalization at low samples are markedly different from those in the large-sample asymptotic regime studies in Stringer et al. 2018 Nature: rather than a trade-off between high-dimensionality and differentiability proposed by Stringer et al., this manuscript shows that in the low-sample regime such codes can be disadvantageous for small sample sizes, that differentiability is not required, that the top eigenvalues matter more than the tail of the spectrum, and what matters is the alignment between the task and the top eigenfunctions. The authors propose sample-efficient learning/generalization as a new principle of neural coding, replacing or complementing efficient coding.

Overall, in my opinion this is a remarkable manuscript, presenting truly innovative theory with somewhat limited but convincing application to neural data. My main concern is that this is highly technical, dense, and long; the mathematical proofs for the theory are buried in the supplement and require knowledge of disparate techniques from statistical physics. Although some of that material on the theory of generalization is covered in previous publications by the authors, it was not clear to me if that is true for all of the technical results or only some.

Fixed population code, learnable linear readout: the authors acknowledge in the very last sentences of the manuscript that this is a limitation, given that neural tuning curves (the population neural code) are adaptable. I imagine extending the theory to both learnable codes and learnable readouts is hard and I understand it's beyond the scope of this paper. But perhaps the authors could motivate and discuss this choice, not just because of its mathematical convenience but also in relation to actual neural systems: when are these assumptions expected to be a good approximation of the real system?

The analysis of V1 data, showing a bias for low-frequency functions of orientation is convincing. But it could help if the authors provided some considerations on the kind of ethological behavioral context where this is relevant, or at least the design of an experimental behavioral task to probe it. Also related, it would be useful to construct and show a counter-example, a synthetic code for which the high-frequency task is easier.

Line 519, data preprocessing: related to the above, is it possible that binning together the V1 responses to gratings with different orientations (a range of 3.6 deg per bin, if I understood correctly) influences the finding of a low-frequency bias?

I found the study of invariances interesting, where the theory provides a normative prediction for the proportion of simple and complex cells. However, I would suggest the authors attempt to bring this analysis a step closer to the actual data: there are no pure simple and complex cells, usually the classification is based on responses to gratings phases (F1/F0) and real neurons take a continuum of values. Could the theory qualitatively predict that distribution?

---

## [Author Response]

Essential revisions:The consensus among all 3 reviewers is that the manuscript describes an insightful and novel mathematical framework for evaluating the ability of a downstream linear decoder to learn new stimulus-response mappings from relatively few training examples. However, there is also broad consensus among the 3 reviewers that (1) the manuscript is too technical and written in a way that may not be palatable to the eLife readership to paraphrase one of the reviewers "I don't think that many experimentalists will immediately see the sheer utility of having a tool like this in their arsenal" and that (2) some of the assumptions made are overly simplistic and the theory needs to be extended with more realistic neural assumptions regarding noise, readout and kernels.Wrt (1), before the manuscript can be accepted for publication, it is necessary that the authors expand on possible applications to neural and behavioral experiments. The authors also need to distill the math better, provide more intuitive explanations, and reduce the jargon. Wrt (2) R2 provided a detailed lists of comments with suggestions for more realistic assumptions.

We thank the editor for these points of advice. We respond to these points in detail below, in our point-by-point rebuttal to the reviewers. Here, we provide a summary of the measures taken to address each of these problems.

– Writing style too technical: We attempted to eliminate mathematical jargon and keep only the crucial mathematical components in the main text. The detailed discussion of the RNN model in the main text was shortened and the reference to Lie groups was removed. We added a paragraph at the beginning of the Results section where we define our mathematical notation explicitly. We also removed unnecessary detail about the V1 Gabor model from the main text. We added more intuition about what a kernel is and how it relates to distances between points in neural space. We also tried adding more intuition about the spectral bias result by giving the simpler expression for the average learned function in Methods Theory of Generalization starting at line 617.

– Contextualize Results for Experimental Audience: To describe how our theory of inductive bias may be useful for experimental neuroscience and could lead to new neural and behavioral experiments, we added a new paragraph at the end of the discussion.

– Assumptions Overly Simplistic: To address our previous model’s failure to model neural noise, we added a Appendix Section Impact of Neural Noise and Unlearnable Targets on Learning, where we discuss effects of noise on the learning curve. We allow for L2 regularization of the readout weights with a ridge penalty *λ* and show that there can be an optimal *λ* to prevent overfitting. We discuss nonlinear readouts in greater detail in App Typical Case Analysis of Nonlinear Readouts, showing that the kernel still governs the learned function for a wider set of learning problems. We discuss recent work, which computes learning curves for nonlinear readouts with a similar technique. Lastly, we provide a more realistic model of V1 tuning curves where neurons are neither completely simple or complex, but have a distribution of F1/F0 values (see App Energy Model with Partially Phase-Selective Cells). We discuss limitations of our work brought up in the reviews in our new discussion. Overall, we view extensions to address these present limitations as promising areas of future research.

Reviewer #1 (Recommendations for the authors):The authors provide a simple and clear way to understand an aspect of the implicit bias of a neural population code linking it with well-known machine learning methods and concepts such as kernel regression, sample complexity and efficiency.Although the mathematical results the authors employ are not novel, the way they apply them to the problem of neural coding is novel and interesting to a broad audience.In particular, the computational neuroscience community can benefit from this work being it is one of the few dealing with the impact of the model implicit bias in explaining real data.

We thank the reviewer for insightful comments and the support. While addressing these comments, we had the chance to clarify some subtle points.

Reviewer #2 (Recommendations for the authors):It is my opinion that the principle utility of this approach lies in its ability to identify the set of 'easily learnable' stimulus-response mappings from neural data which makes strong behavioral predictions that can be easily evaluated. I envision a simple experiment in which empirically obtained kernel functions are used to rank stimulus-response mappings according to their learnability which can then be plotted against measures of performance like the observed learning rate and saturated performance. Because kernel functions are empirically obtained, there is even the potential for meaningful cross-species comparisons. If behaviorally validated, one could also use this approach to label cortical populations by the set of easily learned stimulus-response mappings for that population. This allows for the identification of task-relevant neurons or regions which can be subsequently manipulated to enhance or degrade learning rates.

We thank the reviewer for appreciating the possible applications of our work for future empirical studies on stimulus-response learning and inductive bias in neural systems. We are also excited by these possibilities for neuroscience experiments. We added a this as a discussion item where we suggest some ways our framework could be used in future empirical work. We discuss this addition further below.

Of course, any theoretical approach is only as good as the underlying assumptions and so while the primary strength is the simplicity and generality of this approach, the primary weakness is its neglect of some very real and very relevant aspects of neural data in particular and statistical learning in general. In particular, the three principle limitations of this work are tied to its reliance on the assumptions that (1) neurons are noiseless, (2) decoders are linear, and (3) learned weights are unbiased.

We thank this reviewer for encouraging us to explore additional realism in our model of sample efficient learning and acknowledge the limitations of our current analysis. Below we summarize each of the changes we made in response.

1. We updated the manuscript to allow for biased weights through ridge regression (regression with an additional L2 norm penalty *λ*|**w**|^2^ on the weights) with ridge parameter *λ*. The effect of this parameter on generalization is easy to identify as it merely alters the implicit definition of k to k= λ+k∑kλkλkP+k (see Equation 11).

2. We provide a preliminary analysis of neuron noise in the new Appendix Section Impact of Neural Noise and Unlearnable Targets on Learning. Although we currently do not have a data-averaged theory of generalization (which is involves a very complicated computation), we did attempt empirically to explore the effect of neural variability on task learning. In the new Appendix Figure 1, we show the performance of the non-trial averaged V1 code on the orientation discrimination tasks. While the trial averaged code has much better sample efficiency, the optimal weight decay parameter *λ* for the noisy code is still small.

3. We also studied the effect of training readouts on subsamples of *N* neurons from the total V1 code. For sufficiently small *N*, the target function becomes unexpressible as a linear combination of the *N* selected neurons and the learning rule is likely to overfit. This leads to an optimal weight decay parameter *λ*. We mention how this is related to fitting noisy target functions [37].

4. For the behavior of nonlinear readouts we discuss the results of some recent works which allow for arbitrary nonlinear readout functions in the Discussion and Appendix Convergence of Δ Rule for Nonlinear Readouts. For classification, the power law exponents for the generalization convergence may change, but the phenonenon of code-task alignment is very qualitatively similar [2].

5. We add a Discussion section item about how the presence of neural noise impacts our metabolic efficiency argument. If the neural noise were Gaussian (perhaps with stimulus dependent covariance), then the rotations and baselines would still be legitimate and our argument still obtains. However, for many distributions, such as the Poisson distribution, the variance at a given stimulus is tied to the mean value at that stimulus so simple transformations such as shifts by a baseline or rotation (while preserving the mean response manifold) alter the noise structure and thus the information content of the code. This item is added to the Discussion section.

We will now proceed to provide more detail on each of these topics.

(1) Within this framework, a realistic stimulus-dependent noise model can be easily introduced and its effects on the kernel and set of easily learned stimulus-response mappings investigated. So while the kernel would be substantially altered via the addition of a realistic noise model, the applications of the approach outlined above would not be affected. The same cannot be said for the efficient coding application described in this manuscript. There, the authors note that rotations and constant shifts of neural activity do not affect the kernel and thus do not affect the generalization error. This kernel invariance is not present when a non-trivial (i.e. non-isotropic) noise model is added. For example, suppose that neurons are independent and Poisson so that noise scales with the mean of the neural response. In this case, adding a baseline firing rate to a population of unimodal neurons representing orientation necessarily reduces the information content of the population while rotations can affect the fidelity with which certain stimulus values are represented. It is important to note, however, that while this particular efficiency result is not compelling, I believe that it is possible to perform a similar analysis that takes into account realistic noise models and focuses on a broad set of 'biologically plausible' kernels instead of particular invariant ones. For example, one could consider noise covariance structures with differential correlations (Moreno-Bote 2014). Since the magnitude of differential correlations controls the redundancy of the population code this would enable an analysis of the role of redundancy in suppressing (or enhancing) generalization error.

Impact of Neural Noise on Learning Curve

We agree with the reviewer that any model of learning in a realistic neural system will have to consider the effect of noise (trial-to-trial variability) on the accuracy of learned decoders/predictions. We now mention this explicitly at the beginning of the paper. At the beginning of the Results section on line 93, we added

“These responses define the population code. Throughout this work, we will mostly assume that this population code is determinstic: that identical stimuli generate identical neural responses."

The reviewer is indeed correct that an analysis of generalization can be performed for noisy population codes. Interestingly, even if the noise is stimulus dependent the generalization error can still be expressed in terms of only two correlation matrices, rather than the Fisher information at every value of the stimulus *θ*.

To justify this claim, first let r(*θ*) be stochastic with mean r¯(θ)=⟨r(θ )⟩rθ and covariance Σn(θ)= ⟨(r(θ)−r¯(θ))(r(θ)−r¯(θ))T⟩ For a given configuration of weights w, the generalization error (which is an average over the joint distribution of r*,θ*) is determined only by the signal Σs (w)= ⟨r¯(θ)r¯(θ)T⟩ and (average) noise Σn= ⟨Σn (θ)⟩θ correlation matrices:
(R.9)Eg (w)= ⟨(r(θ)∙w−r¯ (θ)∙w∗)2⟩r,θ= (w− w∗)TΣs (w−w∗)+ wTΣnw

where the (deterministic) target function is given by y(θ)= w∗∙ r¯(θ). The above two expressions can be thought of as a bias-variance decomposition over the noise in neural responses. As before, the challenge is determining the average/typical case behavior of *E_g_* when **w** is learned from the δ rule on *P* samples, this time with noisy responses **r**(*θ_µ_*). The minimum achievable loss can be obtained by differentiation of the population risk expression, giving Eg∗=w∗Σn(Σs+Σn)−1Σsw∗.

While we have not yet computed the full learning curve for arbitrary noise correlation structure **Σ***_n_* (which is quite involved), we hope to pursue this calculation for future work. We want to extend our generalization analysis to arbitrary signal + arbitrary noise correlation structure, including an analysis of the role of differential correlations, but the same computation is made much more difficult by the fact that **Σ***_n_* and **Σ***_s_* are not generally co-diagonalizable.

Further, we empirically tested the effect of neural variability on task learning by training readouts from the V1 code without trial averaging. We studied both the low and high frequency orientation discrimination tasks. The result is provided in R.3 and included in the paper as Appendix Figure 1. We found that the error for the trial-averaged code (dashed) is much lower than the error of the noisy code. However, the error tends to increase with weight decay parameter *λ* without any obvious optimal regularization choice.

We also tested subsampling the neural population, keeping only *N* neurons to model a sparsely connected readout. The result in Figure R.3 (C-D) shows that for small *N*, the learning curves can exhibit overfitting peaks. Further, the generalization error is minimized for an optimal choice of the weight decay parameter.

We address and discuss these points by a new Appendix Section Impact of Neural Noise and Unlearnable Targets on Learning starting on line 1861 together with Figure R.3 (included in the paper as Appendix Figure 1). This section reads:

“While our analysis so far has focused on deterministic population codes, our theory can be extended to neural populations which exhibit variability in responses to identical stimuli. […] At infinite P, generalization error asymptotes to the amount of unexplained variance in the target function.”

We added the following to the Discussion section (line 546)

“Our work focused on how signal correlations influence inductive bias [3, 4]. However, since real neurons do exhibit variability in their responses to identical stimuli, one should consider the effect of neural noise and noise correlations in learning. We provide a preliminary analysis of learning with neural noise in Appendix Impact of Neural Noise and Unlearnable Targets on Learning, where we show that neural noise can lead to irreducible asymptotic error which depends on the geometry of the signal and noise correlations. Further, if the target function is not fully expressible as linear combinations of neural responses, overfitting peaks in the learning curves are possible, but can be mitigated with regularization implemented by a weight decay in the learning rule (see Appendix Figure 1). Future work could extend our analysis to study how signal and noise correlations interact to shape inductive bias and generalization performance in the case where the noise correlation matrices are non-isotropic, including the role of differential correlations [41]. Overall, future work could build on the present analysis to incorporate a greater degree of realism in a theory of inductive bias. ”

Impact of Neural Noise Metabolic Efficiency Results

The reviewer brings up a very good point about how neural noise can prevent decoding accuracy from being preserved under rotations and baseline shifts. To respond to this critique, we will first point out an example (stimulus dependent Gaussian noise), where generalization performance is preserved under rotations and baseline shifts. Note that if the code at *θ* obeyed **r**(*θ*) ∼ N(¯**r**(*θ*)*,***Σ***_n_*(*θ*)), then the rotated and shifted code follows **Qr**(*θ*) + *δ* ∼ N(**Q**¯**r**(*θ*) + *δ,***QΣ***_n_*(*θ*)**Q**^>^). This rotated and shifted code **Qr**(*θ*) + *δ*, when centered, will exhibit identical generalization performance as the original code. This is true both for learning from a trial averaged or non-trial averaged code. In the case of Gaussian noise on a centered code, the dataset transforms under a rotation as

D = {**r***_µ_,y_µ_*} → D^0^ = {*Q***r***_µ_,y_µ_*}. The optimal weights for a linear model similarly transform as **w**(D) → *Q***w**(D). Under these transformations the predictor on test point *θ* is unchanged since(R.10)f(θ)=w⋅r(θ)→w⊤Q⊤Qr(θ)=w⋅r(θ).

Further, the local Fisher information matrixI(Θ)=∂r(θ)T∂θΣn−1(θ)∂r(θ)∂θT+12TrΣn−1(θ)∂Σn(θ)∂θT

is unchanged under the transformation **r** → **Qr** + *δ* since **Σ***_n_*(*θ*) → **QΣ***_n_*(*θ*)**Q**. This shows that, for some noise models, our assumption that rotations and baseline shifts preserve generalization performance will be valid.

However, we agree with the reviewer, that for Poisson noise, where the variance is tied to the mean firing rate, the RROS operations will not preserve noise structure or information content. The Fisher information at scalar stimulus *θ* for a Poisson neuron is I(θ)=r′(θ)2r(θ)

A baseline shift *r* → *r* + *δ* to the tuning curve will not change the numerator since the derivative of the tuning curve is invariant to this transformation, but it will increase the denominator.

We provide a detailed discussion of these points in Appendix Effect of Noise on RROS Symmetry (line 1772) and added a paragraph toward the end of our Discussion section, line 511, which reads:

“As a note of caution, while this analysis holds under the assumption that the neural code is deterministic, real neurons exhibit variability in their responses to repeated stimuli. Such noisy population codes do not, in general, have identical generalization performance under RROS transformations. For example, if each neuron is constrained to produce i.i.d. Poisson noise, then simple shifts of the baseline firing rate reduce the information content of the code. However, if the neural noise is Gaussian (even with stimulus dependent noise covariance), then the generalization error is conserved under RROS operations (App. Effect of Noise on RROS Symmetry). Further studies could focus on revealing the space of codes with equivalent inductive biases under realistic noise models."

(2) Similarly, the linearity assumption is somewhat restrictive. Global linear decoders of neural activity are known to be highly inefficient and completely fail when decoding orientation in the primary visual cortex in the presence of contrast fluctuations. This is because contrast modulates the amplitude of the neural response and doubling the amplitude means doubling an estimate obtained from a linear decoder even when the underlying orientation has not changed. While the contrast issue could be partially addressed by simply considering normalized neural responses, it is not yet clear how to extend this approach to account for other sources of neural variability and co-variability that cause global linear decoders to fail so badly.

We appreciate the reviewer’s critique of global linear decoders. Our intention here is not to design the best decoder, but rather study a decoder’s performance set by the inductive bias of the population code. From this point of view, a linear decoder is a great choice for its analytical tractability. However, we agree that we should at least acknowledge the restrictions that come due to focusing on a linear decoder.

We discuss below some recent works which have extended our methods to nonlinear readouts such as binary classifiers which are invariant to scale transformations **r** → *c*
**r**. Other possible nonlinear functions of **w** · **r** can also be handled within this framework under certain conditions. We have added an acknowledgement of this limitation of our present work and ways it can be extended to the discussion and the new Appendix Typical Case Analysis of Nonlinear Readouts (line 1471). The main text starting on line 541 reads:

“Though our work focused on linear readouts, arbitrary nonlinear readouts which generate convex learning objectives have been recently studied in the high dimensional limit, giving qualitatively similar learning curves which depend on kernel eigenvalues and task model alignment [1, 2] (see App. Typical Case Analysis of Nonlinear Readouts).”

Next, we provide an overview of what is discussed in Appendix Typical Case Analysis of Nonlinear Readouts. The analysis of typical case generalization can be extended to nonlinear predictors and loss functions which depend on the scalar prediction variable *f*(*θ*) = **w** · **r**(*θ*) [1], however the resulting equations are less interpretable. For instance, we can consider binary classification problem where *y* ∈ {±1} by outputting a prediction of *y*ˆ = sign(**w**·**r**) which is invariant to simple rescaling of **r** → *c***r**. We can train a model using the hinge loss *`*(**w** · **r***,y*) = [1 − **w** · **r***y*]_+_ so that the classifier will converge to a kernel support vector machine (SVM) [42]. In general, we can describe a loss function which depends on *f*(*θ*) and the ground truth function *y*(*θ*) as *`*(*f,y*).(R.11)w∗(D)=argminwH(w,D),H(w,D)=∑μ ℓ(w⋅rμ,yμ)+λ∣w∣2

Provided that *l* is convex in **w** · **r**, gradient descent will still converge to the optimum. Further, if **r**(*θ*) is approximately Gaussian, then the generalization performance can still be characterized using statistical mechanics methods [1]. While this set of results is more general, the solution for arbitrary code and task structure has some shortcomings when compared to least squares minimization including the requirement of Monte-carlo sampling to compute test loss and the non-decomposability of the error into separate mode errors.

However, many qualitative features of our results continue to hold, including that the kernel’s diagonalization governs training (Appendix Typical Case Analysis of Nonlinear Readouts) and generalization and that improvements in code task alignment lead to improvements in generalization. In the follow-up work by Cui et al. [2], codes and tasks with power law spectra were analyzed asymptotically, showing power law generalization error decay rates *E_g_* ∼ *P*^−*β*^. The classification learning curves for power law spectra were shown to follow power laws with exponents *β* that are qualitatively similar to the exponents obtained with the square loss which we describe in our section titled Small and Large Sample Size Behaviors of Generalization. Just as in our theory, decay rate exponents *β* are larger for codes which are well aligned to the task and are smaller for codes which are non-aligned.

(3) This analysis relies on the assumption that decoder weights learned in the presence of finite data are efficient and unbiased. This assumption is problematic particularly when it comes to inductive bias and generalization error. This is because a standard way to reduce generalization error is to introduce bias into the learned decoder weights through a penalization scheme that privileges decoder weights with small magnitudes. This kind of regularization is particularly important when neurons are noisy. Fortunately, this issue could be addressed by parameterizing changes in the kernel function by the degree and type of regularization potentially leading to a more general result.

This is a great point. We now discuss inclusion of a ridge parameter *λ* in our calculation which privileges **w** with low *L*2 norm. The *λ* → 0^+^ limit recovers our previous result (see Equations 9 to 11 in Methods, and Appendix Weight Decay and Ridge Regression starting on line 1235).

We explored the effect of neural noise on the learning curve for the V1 population in orientation discrimination (see Figure R.3). The error of the noisy code (solid) is higher than the trial averaged code (dashed). For this code and tasks, the error tends to increase monotonically with regularization strength, indicating a weak effect of noise. However, in (C-D), we study the learning curves for subsamples of *N* neurons from the entire neural population. In this case, overfitting peaks are visible at certain values of *P*. Similarly, there exist non-zero optimal values of weight decay *λ*.

Finally, I would like to conclude by explicitly stating that while the limitations imposed by the assumptions listed above temper my enthusiasm in regards to conclusions drawn in this work, I do not believe there is some fundamental problem with the general theoretical framework. Indeed, items 1 and 3 above can be easily addressed through straightforward extensions of the authors approach and I look forward to their implementation. Item 2 is a bit more troublesome, but my intuition tells me that an information-theoretic extension based upon Fisher information may be capable of eliminating all three of these limiting assumptions by exploiting the relationship between FI(\theta) and FI(y=f(\theta)).

We are also excited by the possibility to analyze sample-efficiency of learning for neural codes with arbitrary signal and noise structure. Though we have only provided a formula for the asymptotic/irreducible error of learning with noisy neurons, we intend to analyze full learning curves in future work. As the reviewer mentions, this is a promising future direction which could even potentially benefit experimentalists in designing learning tasks which can be learned more quickly.

Ultimately, all I felt the need to say is in the public part of the review. But I wanted to use this space to insure that it was clear that I feel that this approach has a lot of potential and that I very much hope to see it extended/generalized.If I were to make any suggestions they would be:(1) a discussion of the consequences of biologically plausible noise models. I believe that this only requires simply augmenting K by a stimulus dependent noise covariance matrix.(2) using this addition of a noise model to enable a discussion of the role of redundancy in generalization. This could be accomplished by considering perturbations to the noise covariance matrix that introduce differential correlations (Moreno-Bote 2014) of varying magnitude. More differential correlations means more redundancy and, I suspect, better generalization.(3) the addition of a paragraph or two outlining some of the other ways that this learnability/generalization measure could be of use to physiologists and behavioral scientists. Training a mouse to do anything more complicated simple orientation detection is challenging to say the least. So having any tool that can be used to identify functions $y=f(\theta)$ that a mouse has a good chance of learning quickly would be highly advantageous. Similarly, by strategically subsampling neurons by tuning properties I suspect it may be possible to identify subpopulations of neurons that are particular important for learning certain functions. This is also cool because it allows physiologists to then target those populations for manipulation.Of these (3) is probably the most important for this manuscript, because I don't think that many experimentalists will immediately see the sheer utility of having a tool like this in their arsenal.

In conclusion, we added the following to address these three points

1. Noise Model: We added the Appendix Section Impact of Neural Noise and Unlearnable Targets on Learning. We discussed this addition in our response above.

2. Structured Noise: We have not yet analyzed the average case learning curves for arbitrary noise correlation structure **Σ***_n_*(*θ*) (since it is much more of an involved calculation), but provide a formula for the irreducible loss. We also discuss our aspiration to pursue an analysis of differential correlations as an important future direction in the discussion.

3. Learnability for Physiologists/Behavioral Scientists: We added the possible applications to experimental neuroscience as a discussion item (see last paragraph) which we quote here:

“Finally, we discuss possible applications of our work to experimental neuroscience. Our theory has potential implications for experimental studies of task learning. First, in cases where the population selective to stimuli can be measured directly, an experimenter could design easy or difficult tasks for an animal to learn from few examples, under a hypothesis that the behavioral output is a linear function of the observed neurons. Second, in cases where it is unclear which neural population contributes to learning, one could utilize our theory to solve the inverse problem of inferring the relevant kernel from observed learning curves on different tasks [43]. From these tasks, the experimenter could compare the inferred kernel to those of different recorded populations. For instance one could compare the kernels from V1, V4, IT in visual ventral stream to the inferred kernel obtained from learning curves on certain visual learning tasks. This could provide new ways to test theories of perceptual learning [7]. Lastly, extensions of our framework could quantify the role of neural variability on task learning and the limitation it imposes on accuracy and sample efficiency.”

Reviewer #3 (Recommendations for the authors):The manuscript presents a theory of generalization performance in deterministic population codes, that applies to the case of small numbers of training examples. The main technical result, as far as I understand, is that generalization performance (the expected classification or regression error) of a population code depends exclusively on the 'kernel', i.e. a measure of the pairwise similarity between population activity patterns corresponding to different inputs. The main conceptual results are that, using this theory, one can understand the inductive biases of the code just from analyzing the kernel, particularly the top eigenfunctions; and that sample-efficient learning (low generalization performance with few samples) depends on whether the task is aligned with the population's inductive bias, that is, whether the target function (i.e. the true map from inputs to outputs) is aligned with the top eigenfunctions of the kernel. For instance, in mouse V1 data, they show that the top eigenfunctions correspond to low frequency functions of visual orientation (i.e. functions that map a broad range of similar orientations to similar output value), and that consistent with the theory, the generalization performance for small sample sizes is better for tasks defined by low frequency target functions. In my opinion, perhaps the most significant finding from a neuroscience perspective, is that the conditions for good generalization at low samples are markedly different from those in the large-sample asymptotic regime studies in Stringer et al. 2018 Nature: rather than a trade-off between high-dimensionality and differentiability proposed by Stringer et al., this manuscript shows that in the low-sample regime such codes can be disadvantageous for small sample sizes, that differentiability is not required, that the top eigenvalues matter more than the tail of the spectrum, and what matters is the alignment between the task and the top eigenfunctions. The authors propose sample-efficient learning/generalization as a new principle of neural coding, replacing or complementing efficient coding.

We thank the reviewer for such a careful reading and impressively accurate summary of our work.

Overall, in my opinion this is a remarkable manuscript, presenting truly innovative theory with somewhat limited but convincing application to neural data. My main concern is that this is highly technical, dense, and long; the mathematical proofs for the theory are buried in the supplement and require knowledge of disparate techniques from statistical physics. Although some of that material on the theory of generalization is covered in previous publications by the authors, it was not clear to me if that is true for all of the technical results or only some.

Thank you for your encouraging words! It is true that our presentation was technically dense. Here is what we did to address this:

– Throughout the paper, we tried eliminating unnecessary jargon and heavy mathematical notation, especially in the V1 model and time-dependent codes sections.

– We tried providing additional intuition and simple derivations in the main text. In the problem setup, we provided a more detailed argument showing that the learned function only depends on the kernel. We also connected kernels to the idea of a distance metric in neural space. We additionally included a simple expression for the average learned function ⟨f(θ,D)⟩D in the Methods Theory of Generalization starting on line 617, showing how the coefficients for each eigenmode are learned at different rates.

– On the question of novelty, the main result on learning curve theory was derived in our prior works [44, 37], but the application of the theory to different neural codes or real neural data had not yet been performed.

Fixed population code, learnable linear readout: the authors acknowledge in the very last sentences of the manuscript that this is a limitation, given that neural tuning curves (the population neural code) are adaptable. I imagine extending the theory to both learnable codes and learnable readouts is hard and I understand it's beyond the scope of this paper. But perhaps the authors could motivate and discuss this choice, not just because of its mathematical convenience but also in relation to actual neural systems: when are these assumptions expected to be a good approximation of the real system?

This is a great point. Below we argue that our theory will apply to learning and generalization dynamics under one of the following two conditions

– Only the readout weights of a certain sensory layer (with code r(θ)) are optimized by a learning rule (such as δ rule in Figure 1). This is the simplest interpretation of our present results. It relies on an assumption that the readout weights converge more quickly than underlying low level sensory representation r(θ) evolves. We would expect this to be the case when learning a new simple task such as a Macaque learning to distinguish two new faces. The IT cortical representation will likely not need to change much in response to the classification of two new stimuli. However, if the animal is learning to classify entirely new types of stimuli over a very long period of training, it may be the case that cortical representations evolve significantly. Research in deep and recurrent networks has revealed operating regimes where the output function can still be written as f(θ) = w · ψ(θ) for a different static feature map ψ(θ) which depends on the architecture of the entire network [23, 24, 45]. For gradient flow, the kernel for this feature map is known as the neural tangent kernel [23]. Deep neural networks operating in this limit have scaling laws which are well predicted by our theory [44, 46, 47, 48]. This operating regime could be relevant for certain learning tasks in biological neural circuits.

While the above cases define the ways that our theory could accurately apply to learning in real sensory systems, there are also regimes of neural network learning where features cannot be modeled as static (in machine learning, this is known as the rich feature learning regime). Ideally a future analysis could extend our results to this setting, but it would be much more challenging. Such feature learning in a deep network is dependent on every layer of processing as well as the specific learning rule one assumes [27, 21]. Since the brain’s solution to the credit assignment problem is still unknown, it is currently unclear which learning rule should be analyzed (back-propagation, feedback alignment, global error broadcasting, gated linear networks, etc). Comparing the typical case performance and learned representations for each of these different learning rules would be a good first step towards understanding the possible inductive biases of networks in the rich regime. Provided a sufficiently refined analysis of these rules, one could perhaps even identify which learning rules are operating in real neural systems based on their measured learning dynamics.

We added the following to the discussion (starting line 520) about when we expect our theory to be accurate

“Our work constitutes a first step towards understanding inductive biases in neuronal circuits. To achieve this, we focused on a linear, δ-rule readout of a static population code. More work is need to study other factors that affect inductive bias. Importantly, sensory neuron tuning curves can adapt during perceptual learning tasks [7, 14, 9, 8] with the strength of adaptation dependent on brain area [10, 11, 12]. However, in many experiments, these changes to tuning in sensory areas are small [8, 9], satisfying the assumptions of our theory. For example monkeys trained on noisy visual motion detection exhibit changes in sensory-motor (LIP) but not sensory areas (MT), consistent with a model of readout from a static sensory population code [13, 15]. However, other perceptual learning tasks and other brain areas can exhibit significant changes in neural tuning [16, 17, 18]. This diversity of results motivates more general analysis of learning in multilayer networks, where the representations in each layer can adapt flexibly to task structure [19, 20, 21, 22]. Alternatively, our current analysis of inductive bias can still be consistent with multilayer learning if the network is sufficiently overparameterized and tuning curves change very little [23, 24, 19]. In this case, network training is equivalent to kernel learning with a kernel that depends on the learning rule and architecture [25]. However, in the regime of neural network training where tuning curves change significantly, more sophisticated analytical tools are needed to predict generalization [26, 27, 21]. Though our work focused on linear readouts, arbitrary nonlinear readouts which generate convex learning objectives have been recently studied in the high dimensional limit, giving qualitatively similar learning curves which depend on kernel eigenvalues and task model alignment [1, 2].”

The analysis of V1 data, showing a bias for low-frequency functions of orientation is convincing. But it could help if the authors provided some considerations on the kind of ethological behavioral context where this is relevant, or at least the design of an experimental behavioral task to probe it. Also related, it would be useful to construct and show a counter-example, a synthetic code for which the high-frequency task is easier.

We thank the reviewer for this comment.

Applications to experimental neuroscience: We added the possible applications to experimental neuroscience as a discussion item (see last paragraph) which we quote here:

“Finally, we discuss possible applications of our work to experimental neuroscience. Our theory has potential implications for experimental studies of task learning. First, in cases where the population selective to stimuli can be measured directly, an experimenter could design easy or difficult tasks for an animal to learn from few examples, under a hypothesis that the behavioral output is a linear function of the observed neurons. Second, in cases where it is unclear which neural population contributes to learning, one could utilize our theory to solve the inverse problem of inferring the relevant kernel from observed learning curves on different tasks [43]. From these tasks, the experimenter could compare the inferred kernel to those of different recorded populations. For instance one could compare the kernels from V1, V4, IT in visual ventral stream to the inferred kernel obtained from learning curves on certain visual learning tasks. This could provide new ways to test theories of perceptual learning [7]. Lastly, extensions of our framework could quantify the role of neural variability on task learning and the limitation it imposes on accuracy and sample efficiency.”

Orientation discrimination task: We agree that this orientation discrimination task isn’t perhaps the most ethologically relevant one, but given that the data consists of responses to oriented gratings, it is a reasonable task to consider. Since neurons are selective to bars at particular orientations, this kind of task is intuitively expected to have high alignment with the V1 code. Also, because the eigenfuntions are close to Fourier modes, the inductive is more interpretable, and, thus, this task serves as a good starter example.

We also analyzed two ethologically more relevant tasks. First was a scene reconstruction task from V1 (Figure 4), and found that low-pass filtered images are easier to reconstruct, consistent with low spatial frequency selectivity of mouse V1 neurons. Second was a more ethologically natural scene categorization task (Figure 3D), but the V1 code did not generalize well, suggesting that later stages of the visual ventral stream may be better suited for categorization tasks. We suspect that other regions in visual stream may be more aligned to these ethologically relevant categorization/classification tasks. For example, primate IT codes may be more aligned with object classification.

Synthetic, counter-example code: This is a fantastic idea! Our Figure 1 actually does this. Figure 1B has two different codes, one for which a low-frequency task is easier and another one for which a high-frequency task is easier. These codes are synthetically created by generating tuning curves from a procedure we describe in Methods Generating Example Codes.

Line 519, data preprocessing: related to the above, is it possible that binning together the V1 responses to gratings with different orientations (a range of 3.6 deg per bin, if I understood correctly) influences the finding of a low-frequency bias?

Thanks for this comment. Based on the reviewer’s suggestion, we computed the eigendecomposition of trial averaged responses for different bin numbers bins ∈ {50*,*100*,*200} which correspond to angular windows of {3.6∘,1.8∘,0.9∘}. The spectrum and eigenfunctions (especially the top modes) are very consistent across different bin sizes. The eigenfunctions become noisier with smaller bins since there are fewer trials to average over.

This figure is included as Figure 3—figure supplement 1 and referred to in the Main text line number 209.

I found the study of invariances interesting, where the theory provides a normative prediction for the proportion of simple and complex cells. However, I would suggest the authors attempt to bring this analysis a step closer to the actual data: there are no pure simple and complex cells, usually the classification is based on responses to gratings phases (F1/F0) and real neurons take a continuum of values. Could the theory qualitatively predict that distribution?

We thank the reviewer for pointing out that our model of simple/complex V1 cells was overly simplified. In reality, as the reviewer mentions there is continuous variability in F1/F0 across the neural population. In response, we modified our energy based model which produces a continuum of F1/F0 values. Further, a single scalar parameter *α* controls how strongly the population is biased towards invariance.

Concretely, we define the response of a cell ri(θ,ϕ)=(zi(θ,ϕ)) for nonlinear function *g* and scalar *z* which is constructed as follows(R.12)zi(θ,φ)=β1[F(θi,φi)⋅h(θ,φ)]+2+β2[F(θi,φi+π/2)⋅h(θ,φ)]+2            +β3[F(θi,φi+π)⋅h(θ,φ)]+2+β4[F(θiφi+3π/2)⋅h(θ,φ)]+2.

This linear combination is inspired by the construction of complex cells in Dayan and Abbot Chapter 2. If only one of the *β*s were 1 and the rest were zero, this would be a perfect simple cell. If all the *β*s were equal, this would a perfect complex cell that is invariant to stimulus phase *φ*. To generate variability in tuning to phase *φ*, we will draw *β* from a Dirichlet distribution on the simplex with concentration parameter *α***1** so that *p*(*β*) ∝ ^Q4^_*j*=1_
*β_j_^α^*^−1^ with ^P4^_*j*=1_
*β_j_* = 1. In the *α* → ∞ limit, the probability density concentrates on <inline-graphic mimetype="image" mime-subtype="png" xlink:href="media/image1.png" />, leading to a code comprised entirely of complex cells which are invariant to phase *φ*. In the *α* → 0 limit, the density is concentrated around the “edges” of the simplex such as (1*,*0*,*0*,*0)*,*(0*,*1*,*0*,*0), etc, giving a population of pure simple cells. For intermediate values, neurons are partially selective to phase. As before, the selectivity or invariance to phase is manifested in the kernel decomposition and leads to similar learning curves for the three tasks of the main paper (Orientation, Phase, Hybrid). We provide an illustration of tuning curves, eigenfunctions, and learning curves in Figure R.5. This model is discussed in detail the new Appendix Energy Model with Partially Phase-Selective Cells (line 1677). The Figure is included as Figure 5—figure supplement 3.

References

Bruno Loureiro, Cédric Gerbelot, Hugo Cui, Sebastian Goldt, Florent Krzakala, Marc Mézard, and Lenka Zdeborová. Capturing the learning curves of generic features maps for realistic data sets with a teacher-student model. *CoRR*, abs/2102.08127, 2021.Hugo Cui, Bruno Loureiro, Florent Krzakala, and Lenka Zdeborová. Error rates for kernel classification under source and capacity conditions, 2022.Bruno Averbeck, Peter Latham, and Alexandre Pouget. Neural correlations, population coding and computation. *Nature Reviews Neuroscience*, 7, 2006.Marlene R Cohen and Adam Kohn. Measuring and interpreting neuronal correlations. *Nature neuroscience*, 14(7):811, 2011.H S Seung and H Sompolinsky. Simple models for reading neuronal population codes. *Proceedings of the National Academy of Sciences*, 90(22):10749–10753, 1993.Uri Cohen, Sueyeon Chung, Daniel Lee, and Haim Sompolinsky. Separability and geometry of object manifolds in deep neural networks. *Nature Communications*, 11(746), 2020.Charles D Gilbert. Early perceptual learning. *Proceedings of the National Academy of Sciences of the United States of America*, 91(4):1195, 1994.Aniek Schoups, Rufin Vogels, Ning Qian, and Guy Orban. Practising orientation identification improves orientation coding in v1 neurons. *Nature*, 412(6846):549–553, 2001.Geoffrey M Ghose, Tianming Yang, and John HR Maunsell. Physiological correlates of perceptual learning in monkey v1 and v2. *Journal of neurophysiology*, 87(4):1867–1888, 2002.Tianming Yang and John HR Maunsell. The effect of perceptual learning on neuronal responses in monkey visual area v4. *Journal of Neuroscience*, 24(7):1617–1626, 2004.Hamed Zivari Adab, Ivo D Popivanov, Wim Vanduffel, and Rufin Vogels. Perceptual learning of simple stimuli modifies stimulus representations in posterior inferior temporal cortex. *Journalof cognitive neuroscience*, 26(10):2187–2200, 2014.

*neuroscience*, 24(10):1441–1451, 2021.

Michael N Shadlen and William T Newsome. Neural basis of a perceptual decision in the parietal cortex (area lip) of the rhesus monkey. *Journal of neurophysiology*, 86(4):1916–1936, 2001.Gregg H Recanzone, Christoph E Schreiner, and Michael M Merzenich. Plasticity in the frequency representation of primary auditory cortex following discrimination training in adult owl monkeys. *Journal of Neuroscience*, 13(1):87–103, 1993.Burkhard Pleger, Ann-Freya Foerster, Patrick Ragert, Hubert R Dinse, Peter Schwenkreis, Jean-Pierre Malin, Volkmar Nicolas, and Martin Tegenthoff. Functional imaging of perceptual learning in human primary and secondary somatosensory cortex. *Neuron*, 40(3):643–653, 2003.Christopher S Furmanski, Denis Schluppeck, and Stephen A Engel. Learning strengthens the response of primary visual cortex to simple patterns. *Current Biology*, 14(7):573–578, 2004.Haozhe Shan and Haim Sompolinsky. A minimum perturbation theory of deep perceptual learning, 2021.Francesca Mastrogiuseppe, Naoki Hiratani, and Peter Latham. Evolution of neural activity in circuits bridging sensory and abstract knowledge. *bioRxiv*, 2022.Blake Bordelon and Cengiz Pehlevan. Self-consistent dynamical field theory of kernel evolution in wide neural networks. *arXiv preprint arXiv:2205.09653*, 2022.Merav Ahissar and Shaul Hochstein. The reverse hierarchy theory of visual perceptual learning. *Trends in cognitive sciences*, 8(10):457–464, 2004.Arthur Jacot, Franck Gabriel, and Clement Hongler. Neural tangent kernel: Convergence and generalization in neural networks. In S. Bengio, H. Wallach, H. Larochelle, K. Grauman, N. Cesa-Bianchi, and R. Garnett, editors, *Advances in Neural Information Processing Systems*, volume 31, pages 8571–8580. Curran Associates, Inc, 2018.Jaehoon Lee, Jascha Sohl-dickstein, Jeffrey Pennington, Roman Novak, Sam Schoenholz, and Yasaman Bahri. Deep neural networks as gaussian processes. In *International Conference on Learning Representations*, 2018.Blake Bordelon and Cengiz Pehlevan. The influence of learning rule on representation dynamics in wide neural networks, 2022.Timo Flesch, Keno Juechems, Tsvetomira Dumbalska, Andrew Saxe, and Christopher Summerfield. Rich and lazy learning of task representations in brains and neural networks. *bioRxiv*, 2021.Greg Yang and Edward J Hu. Tensor programs iv: Feature learning in infinite-width neural networks. In *International Conference on Machine Learning*, pages 11727–11737. PMLR, 2021.D Hansel and C Van Vreeswijk. How noise contributes to contrast invariance of orientation tuning in cat visual cortex. *Journal of Neuroscience*, 22(12):5118–5128, 2002.Kenneth D Miller and Todd W Troyer. Neural noise can explain expansive, power-law nonlinearities in neural response functions. *Journal of neurophysiology*, 87(2):653–659, 2002.Nicholas J Priebe, Ferenc Mechler, Matteo Carandini, and David Ferster. The contribution of spike threshold to the dichotomy of cortical simple and complex cells. *Nature neuroscience*, 7(10):1113–1122, 2004.Nicholas J Priebe and David Ferster. Inhibition, spike threshold, and stimulus selectivity in primary visual cortex. *Neuron*, 57(4):482–497, 2008.32.Song Mei, Theodor Misiakiewicz, and Andrea Montanari. Learning with invariances in random features and kernel models, 2021.33.Zhiyuan Li, Ruosong Wang, Dingli Yu, Simon S. Du, Wei Hu, Ruslan Salakhutdinov, and Sanjeev Arora. Enhanced convolutional neural tangent kernels, 2019.34.Matthew Farrell, Blake Bordelon, Shubhendu Trivedi, and Cengiz Pehlevan. Capacity of groupinvariant linear readouts from equivariant representations: How many objects can be linearly classified under all possible views? *arXiv preprint arXiv:2110.07472*, 2021.35.Lechao Xiao and Jeffrey Pennington. Synergy and symmetry in deep learning: Interactions between the data, model, and inference algorithm, 2022.36.Carsen Stringer, Marius Pachitariu, Nicholas Steinmetz, Matteo Carandini, and Kenneth D.

Harris. High-dimensional geometry of population responses in visual cortex. *Nature*, 571, 6 2018.

37.Abdulkadir Canatar, Blake Bordelon, and Cengiz Pehlevan. Spectral bias and task-model alignment explain generalization in kernel regression and infinitely wide neural networks. *Nature Communications*, in press, 2021.